# System-Aware Unlearning Algorithms:
# Use Lesser, Forget Faster

**Linda Lu** [1]  **Ayush Sekhari** [2]  **Karthik Sridharan** [1]

## Abstract

Machine unlearning addresses the problem of updating a machine learning model/system trained on a dataset $S$ so that the influence of a set of deletion requests $U \subseteq S$ on the unlearned model is minimized. The gold standard definition of unlearning demands that the updated model, after deletion, be nearly identical to the model obtained by retraining. This definition is designed for a worst-case attacker (one who can recover not only the unlearned model but also the remaining data samples, i.e., $S \setminus U$. Such a stringent definition has made developing efficient unlearning algorithms challenging. However, such strong attackers are also unrealistic. In this work, we propose a new definition, *system-aware unlearning*, which aims to provide unlearning guarantees against an attacker that can at best only gain access to the data stored in the system for learning/unlearning requests and not all of $S \setminus U$. With this new definition, we use the simple intuition that if a system can store less to make its learning/unlearning updates, it can be more secure and update more efficiently against a system-aware attacker. Towards that end, we present an exact system-aware unlearning algorithm for linear classification using a selective sampling-based approach, and we generalize the method for classification with general function classes. We theoretically analyze the tradeoffs between deletion capacity, accuracy, memory, and computation time.

## 1. Introduction

Today's large-scale Machine Learning (ML) models are often trained on extensive datasets containing sensitive or personal information. Thus, concerns surrounding privacy and data protection have become increasingly prominent (Yao et al., 2024). These models, due to their high capacity to memorize patterns in the training data, may inadvertently retain and expose information about individual data points (Carlini et al., 2021). This presents significant challenges in the context of privacy regulations such as the European Union's General Data Protection Regulation (2016) (GDPR), California Consumer Privacy Act (2018) (CCPA), and Canada's proposed Consumer Privacy Protection Act, all of which emphasize the "right to be forgotten". As a result, there is a growing need for methods that enable the selective removal of specific training data from models that have already been trained, a process commonly referred to as *machine unlearning* (Cao & Yang, 2015). Beyond privacy, unlearning can be used to combat undesirable model behavior after deployment, such as copyright violations (Dou et al., 2024).

Machine unlearning addresses the need to remove data from a model's knowledge base without the need to retrain the model from scratch each time there is a deletion request, since this can be computationally expensive and often impractical for large-scale systems. The overarching objective is to ensure that, post-unlearning, a model "acts" as if the removed data were never part of the training process. Traditionally, this has been defined through notions of exact (or approximate) unlearning, wherein the model's hypothesis after unlearning should be identical (or probabilistically equivalent) to the model obtained by retraining from scratch on the entire dataset after removing only the deleted points (Sekhari et al., 2021; Ghazi et al., 2023; Guo et al., 2019). While such definitions offer rigorous guarantees even in the most pessimistic scenarios, they often impose stringent requirements, limiting the practical applicability of machine unlearning. This is evidenced by the lack of efficient certified exact or approximate unlearning algorithms beyond the simple case of convex loss functions. Furthermore, Cherapanamjeri et al. (2025) proved that under the traditional definition of unlearning, there exist simple model classes with finite VC dimension, such as linear classifiers, where traditional exact unlearning requires the storage of the entire dataset in order to unlearn. For large datasets, this makes unlearning under the traditional definitions impractical.

[1]Cornell University [2]Boston University. Correspondence to: Linda Lu <lulinda@cs.cornell.edu>.

*Proceedings of the 42nd International Conference on Machine Learning*, Vancouver, Canada. PMLR 267, 2025. Copyright 2025 by the author(s).

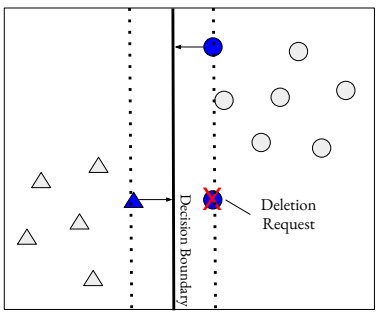 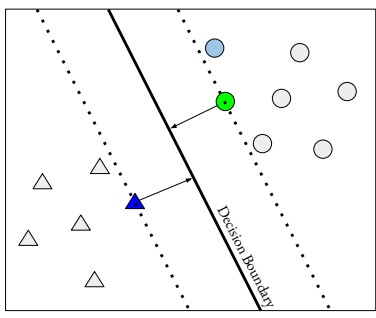 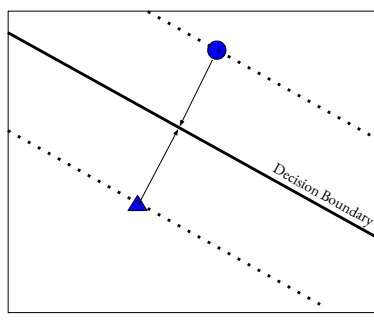

(a) Hard Margin SVM with Deletion     (b) Traditional Unlearning - $A(S \smallsetminus U, \varnothing)$     (c) System-Aware Unlearning - $A(S' \smallsetminus U, \varnothing)$

*Figure 1.* Consider a deletion on a hard margin SVM where the state-of-system $\mathsf{I}_A(S, U)$ is the set of support vectors (in dark blue) and the unlearned model $A(S, U)$. Let $S'$ be the set of support vectors . Thus, we have $A(S, \varnothing) = A(S', \varnothing)$. Under traditional unlearning definitions, when a support vector is deleted, we need to recover the hypothesis from training a new hard margin SVM on the remaining data points $A(S \smallsetminus U, \varnothing)$ (Figure 1(b)). This can lead to new support vectors being selected (ie. the green point in Figure 1(b)), which can drastically change the decision boundary. Under system-aware unlearning, since the support vectors are the only points in the sample that affect the decision boundary, we can treat the remaining support vectors after deletion as a plausible starting sample $S' \smallsetminus U$ and train a new hard margin SVM on the remaining support vectors $A(S' \smallsetminus U, \varnothing)$ (Figure 1(c)), as if the other points in the sample never existed.

At the core unlearning lies a fundamental question: *What does it truly mean to "remove" a data point from a trained model? And more importantly, when we provide privacy guarantees for deleted points against an outside observer or attacker, what information can this attacker reasonably possess?* The current definitions of exact and approximate unlearning take a worst-case perspective here and focus on the output hypothesis being indistinguishable from a retrained model (Sekhari et al., 2021; Ghazi et al., 2023; Guo et al., 2019). However, this approach overlooks a key aspect of the unlearning problem: the observer and their knowledge of the system. In the real world, the feasibility and complexity of unlearning should depend on what the observer can reasonably access, such as model parameters, data retained by the ML system in its memory, data ever encountered by the ML system, etc. For instance, consider a learning algorithm that relies on only a fraction of its training dataset to generate its hypothesis and hence the ML system only stores this data. In such cases, unlearning a data point should intuitively be more straightforward. Even if the entire memory of the system is compromised at some point, only the privacy of the stored points is in jeopardy as long as the learned model does not reveal much about points that the model did not use. Even if the observer has access to larger public datasets that might include parts of the training data, we can expect privacy for data that the system does not use directly for building the model to be preserved. Conversely, if the algorithm utilizes the entire dataset and retains all information in memory, unlearning becomes far more challenging, potentially requiring retraining from scratch, and this is the scenario the current definitions aim to cover.

**Contributions.** Our main contributions are:

- We propose a new, *system-aware formulation of ma-*

*chine unlearning*, which provides unlearning guarantees against an attacker who can observe the entire state of the system after unlearning (including anything the learning system stores or uses internally). If the system stores the entire remaining dataset, then system-aware unlearning definition becomes as stringent as traditional unlearning, but otherwise relaxes it. Thus, we prove that system-aware unlearning generalizes traditional unlearning.

- We present a general framework for system-aware unlearning using sample compression based or core set based algorithms. These algorithms leverage the fact that when an algorithm relies on less information, less information can be exposed to an attacker.

- We present an exact system-aware unlearning algorithm for linear classification using selective sampling for sample compression, thus resulting in *the first exact unlearning algorithm for linear classification requiring memory sublinear in the number of samples*. We establish theoretical bounds on its computation time, memory requirements, deletion capacity, and excess risk.

- We generalize our approach from linear classification to classification with general functions. Thus, providing a novel reduction from (monotonic) selective sampling for general function classes to system-aware unlearning.

The third bullet is particularly interesting because Cherapanamjeri et al. (2025) proved that under traditional unlearning, any exact unlearning algorithm for linear classification must store the entire dataset.

## 2. Setup and Unlearning Definition

Let $\mathcal{X}$ be the space of inputs, $\mathcal{Y}$ be the space of outputs, $\mathcal{D}$ be a distribution over an instance space $\mathcal{Z} = \mathcal{X} \times \mathcal{Y}$, $\mathcal{F} \subseteq \mathcal{X}^{\mathcal{Y}}$ be a model class, and $\ell : \mathcal{Y} \times \mathcal{Y} \to \mathbb{R}$ be a loss function. The

goal of a learning algorithm is to take in a dataset $S \in \mathcal{Z}^*$ over the instance space and output a predictor $\hat{f} \in \mathcal{F}$ which minimizes the excess risk compared to the best predictor $f^* \in \mathcal{F}$ in the model class, where the excess risk is

$$\mathcal{E}(\hat{f}) \coloneqq \mathop{\mathbb{E}}_{(x,y)\sim\mathcal{D}}[\ell(\hat{f}(x),y)] - \min_{f^*\in\mathcal{F}} \mathop{\mathbb{E}}_{(x,y)\sim\mathcal{D}}[\ell(f^*(x),y)].$$

Our goal in machine unlearning is to provide a privacy guarantee for data samples that request to be deleted, while ensuring that the updated hypothesis post-unlearning still has small excess risk. We first present the standard definition of machine unlearning, as stated in Sekhari et al. (2021); Guo et al. (2019), often referred to as *certified machine learning*, which generalizes the commonly used *data deletion guarantee* from Ginart et al. (2019).

**Definition 2.1** (($\varepsilon,\delta$)-unlearning). For a dataset $S \in \mathcal{Z}^*$, and deletions requests $U \subseteq S$, a learning algorithm $A : \mathcal{Z}^* \mapsto \Delta(\mathcal{F})$ and an unlearning algorithm $\bar{A} : \mathcal{Z}^* \times \mathcal{F} \times \mathcal{T} \mapsto \Delta(\mathcal{F})$ is ($\varepsilon,\delta$)-unlearning if for any $F \subseteq \mathcal{F}$,

$$\Pr(\bar{A}(U, A(S), T(S)) \in F)$$
$$\leq e^\varepsilon \cdot \Pr\left(\bar{A}(\varnothing, A(S \smallsetminus U), T(S \smallsetminus U)) \in F\right) + \delta,$$

and

$$\Pr(\bar{A}(\varnothing, A(S \smallsetminus U), T(S \smallsetminus U)) \in F)$$
$$\leq e^\varepsilon \cdot \Pr\left(\bar{A}(U, A(S), T(S)) \in F\right) + \delta,$$

where $T(S)$ denotes any intermediate auxiliary information that is available to $\bar{A}$ for unlearning.

Sekhari et al. (2021) also defined a notion of *deletion capacity*, which controls the number of samples that can be deleted while satisfying the above definition, and simultaneously ensuring good excess risk performance. We defer a discussion of other related work on unlearning definitions and algorithms to Appendix A.

The above definition implicitly assumes that the attacker, in the worst case, has knowledge of $S \smallsetminus U$ and can execute the unlearning algorithm on $S \smallsetminus U$. Although this provides privacy against the most knowledgeable attacker, we argue with a very simple motivating example that the above definition may, unfortunately, be an overkill even in some toy scenarios where we want to unlearn. Consider an algorithm that learns by first randomly sampling a small subset $C \subseteq S$ of size $m$ and then uses $C$ to train a model and discards the rest of the samples in $S \smallsetminus C$. Now, consider an unlearning algorithm that, when given some deletion requests $U$, simply retrains from scratch on $C \smallsetminus U$. This unlearning algorithm is not equivalent to rerunning the algorithm from scratch on $S \smallsetminus U$ which would involve sampling a different subset $C'$ of $m$ samples from $S \smallsetminus U$ and then training a model on $C'$. Since $C'$ contains $m$ samples whereas $C \smallsetminus U$ contains $m - |U|$ samples, the corresponding hypotheses

will not be statistically indistinguishable from each other. Thus, under Definition 2.1, this is not a valid unlearning algorithm. However, this is a valid unlearning algorithm from the perspective of an attacker who only observes the model after unlearning and (in the best case) stored samples $C \smallsetminus U$; here neither of these reveals any information about the deleted samples $U$. Furthermore, an attacker has no ability to gain access to $S \smallsetminus U$ and compare what would have happened had the algorithm been trained on $S \smallsetminus U$.

The crucial thing to note is that Definition 2.1 considers a worst-case scenario where every point encountered by the unlearning algorithm except for the deletion requests, regardless of whether those points were actually used or stored, are known to the attacker. *However, this is unrealistic; samples that were never used for learning or stored in memory can never be leaked to the attacker.* Towards this end, we develop an alternative definition of unlearning. However, we first need to formalize the information that an adversary can compromise from the system post-unlearning.

**Definition 2.2** (State-of-System). Let $\mathcal{I}$ denote some arbitrary set of all possible states. For an unlearning algorithm $A$, we use the mapping $\mathsf{l}_A : \mathcal{Z}^* \times \mathcal{Z}^* \mapsto \Delta(\mathcal{I})$ to denote what is saved in the system by $A$ after unlearning (e.g. the model, any stored samples, auxiliary data statistics, etc.). That is, $\mathsf{l}_A(S,U)$ denotes the state-of-system (what is stored in the system) after learning from sample $S$ and performing the update for unlearning request $U$.

For the system to be useful, it is natural to assume that the state-of-system $\mathsf{l}_A(S,U)$ either contains the unlearned model $A(S,U)$, or more generally, the unlearned model $A(S,U)$ can be computed as a function of $\mathsf{l}_A(S,U)$. If algorithm $A$ requires the storage of samples, intermediate models, or auxiliary information in order to learn or unlearn in the future, then those must also be contained in $\mathsf{l}_A(S,U)$. Whenever clear from the context, we will drop the subscript $A$ from $\mathsf{l}_A$ to simplify the notation. Using the state-of-system $\mathsf{l}_A(S,U)$, we present a system-aware definition of unlearning.

**Definition 2.3** (System-Aware-($\varepsilon,\delta$)-Unlearning). Let $A$ be a (possibly randomized) learning-unlearning algorithm, that first learns on dataset $S \in \mathcal{Z}^*$, then processes a set of deletion requests $U \subseteq S$, and after unlearning, has state-of-system $\mathsf{l}_A(S,U)$. We say that $A$ is a system-aware-($\varepsilon,\delta$)-unlearning algorithm if for all $S$, there exists a $S' \subseteq S$, such that for all $U \subseteq S$, for all measurable sets $F$,

$$\Pr(\mathsf{l}_A(S,U) \in F) \leq e^\varepsilon \cdot \Pr(\mathsf{l}_A(S' \smallsetminus U, \varnothing) \in F) + \delta$$

and

$$\Pr(\mathsf{l}_A(S' \smallsetminus U, \varnothing) \in F) \leq e^\varepsilon \cdot \Pr(\mathsf{l}_A(S,U) \in F) + \delta.$$

If an unlearning algorithm satisfies Definition 2.3 with $\varepsilon = \delta = 0$, then we say that the algorithm is an *exact system-*

*aware unlearning algorithm.* We further remark that this definition assumes that the selection of $U$ is oblivious to the randomness in $A$, but can depend on $S$.

The definition above indicates that for any sample $S$, there exists a subset $S'$ that one can think of as being a good representative of $S$ because the state-of-system when trained on $S$ is nearly identical to that when trained on $S'$. For all unlearning requests $U$, the state-of-system $I_A(S, U)$ after processing the set of deletions is statistically similar to the state-of-system $I_A(S' \smallsetminus U, \varnothing)$ when directly training on $S' \smallsetminus U$. Under system-aware unlearning, we take advantage of the fact that the algorithm designer has control over the information that an attacker could compromise after unlearning, specifically, the fact that the state-of-system is mostly determined by the smaller subset $S'$. Recall that the traditional unlearning definition does not account for this.

Clearly, taking $S' = S$ and $I_A(S, U) = A(S, U)$ (the state-of-system to be exactly the unlearned model), we recover the traditional notion of unlearning from Definition 2.1. This corresponds to the scenario where the entire remaining dataset $S \smallsetminus U$ can be accessed by the adversary. *Thus, system-aware unlearning strictly generalizes the traditional definition of unlearning.* We point out that the traditional definition of unlearning does not require indistinguishability for auxiliary information stored in the system outside of the unlearned model; thus, the traditional definition does not account for system-awareness. If we wanted to view traditional unlearning through the lens of system-awareness, we would consider Definition 2.3 with the additional requirement that $S' = S$.

### 2.1. Why is Considering $S' \smallsetminus U$, Instead of $S \smallsetminus U$, Sufficient to Provide Privacy Guarantees?

Since $S'$ must be fixed ahead of time for all possible deletion requests $U$, $S'$ depends on $S$ but not on $U$. If $S'$ is only a function of $S$ and has no dependence on $U$, then intuitively, $S' \smallsetminus U$ should not leak any more information about $U$ as compared to $S \smallsetminus U$. We formalize this intuition through mutual information. Mutual information is a common way to measure the privacy leakage of an algorithm (Mir, 2013; Cuff & Yu, 2016). Mutual information $\mathsf{MI}(A; B)$ quantifies the amount of information one gains about random variable $A$ by observing random variable $B$.

**Theorem 2.4.** *Let dataset $S$ and set of deletions $U \subseteq S$ come from a stochastic process $\mu$. Then,*

$$\sup_{\mu}(\mathsf{MI}(U; S' \smallsetminus U) - \mathsf{MI}(U; S \smallsetminus U)) \le 0.$$

**Proof.** Since $S' \subseteq S$, notice that $S \smallsetminus U = ((S' \smallsetminus U), ((S \smallsetminus S') \smallsetminus U))$. Consider $\mathsf{MI}(S \smallsetminus U; U)$, applying chain rule:

$\mathsf{MI}(S \smallsetminus U; U)$
$\quad = \mathsf{MI}(S' \smallsetminus U; U) + \mathsf{MI}((S \smallsetminus S') \smallsetminus U; U \mid S' \smallsetminus U)$

$$\ge \mathsf{MI}(S' \smallsetminus U; U),$$

where the last line holds by non-negativity of mutual information. $\qquad\square$

Thus, $S' \smallsetminus U$ leaks no more information about $U$ than $S \smallsetminus U$. For recovering the retraining-from-scratch hypothesis $A(S \smallsetminus U, \varnothing)$ to be a reasonable objective for providing privacy, traditional unlearning definitions implicitly assume that the information between the deleted individuals $U$ and the remaining dataset $S \smallsetminus U$ is small. Theorem 2.4 implies that if the information between $U$ and $S \smallsetminus U$ is small, the information between $U$ and $S' \smallsetminus U$ is also small. We note that all of our technical results, including the relative privacy guarantee from Theorem 2.4, hold without any assumptions on the mutual information between $S$ and $S \smallsetminus U$.

### 2.2. When is the Flexibility of $S' \ne S$ Helpful?

To understand the power of having $S'$ to be different from $S$, consider the following simple scenario: Suppose the learning algorithm $A$ allows for $S$ to be compressed into a small set $S' \subseteq S$ such that $A(S, \varnothing) \approx A(S', \varnothing)$, with state-of-system $I_A(S, \varnothing) = S'$. In this case, under our new definition (Definition 2.3), it is straightforward to handle deletion requests $U$ for which $S' \cap U = \varnothing$, by not making any change to the trained model. To see this, observe that for such a $U$, $A(S' \smallsetminus U, \varnothing) = A(S', \varnothing) \approx A(S, \varnothing)$, and thus we will satisfy Definition 2.3 with $S'$ on the right-hand side. In other words, the individuals outside of $S'$ are already unlearned for free, since the original model output did not rely on them much to begin with. On the other hand, such an algorithm will not satisfy the conditions of the classic unlearning definition in Definition 2.1 for such a $U$ (see Section 4 for more details and a concrete example).

We can formalize algorithms that only rely on a subset of the training dataset as *core set algorithms*.

**Definition 2.5** (Core Set Algorithm). A learning algorithm $A_{\mathrm{cs}} : \mathcal{Z}^* \mapsto \Delta(\mathcal{F})$ is said to be a core set algorithm if there exists a mapping $\mathfrak{C} : \mathcal{Z}^* \mapsto \mathcal{Z}^*$ such that for any $S \in \mathcal{Z}^*$, $\mathfrak{C}(S) \subseteq S$, and we have

$$A_{\mathrm{cs}}(S) = A_{\mathrm{cs}}(\mathfrak{C}(S)).$$

We define $\mathfrak{C}(S) \subseteq S$ to be the *core set* of $S$.

Many sample compression-based learning algorithms for classification tasks, such as SVM or selective sampling, are core set algorithms (Hanneke & Kontorovich, 2021; Floyd & Warmuth, 1995). For hard margin SVMs, the core set is the set of support vectors (Cortes & Vapnik, 1995), and for selective sampling, the core set is the set of queried points (see Section 4 for further details). Typically, the number of samples in $\mathfrak{C}(S)$ is much smaller than $|S|$.

**Additional Examples.** To further highlight the benefit of system-awareness in unlearning, consider the following additional examples:

- *Sample Compression.* Going back to our motivating example, an algorithm that first trains a model on a small compressed set $C \subseteq S$ and then retrains on $C \setminus U$ would be a valid exact unlearning algorithm under system-aware unlearning with $S' = C$.

- *Hard Margin SVM.* $S'$ can be taken to be the set of support vectors. Figure 1 depicts the key differences between traditional unlearning definitions (Definition 2.1) and system-aware unlearning (Definition 2.3) on a hard margin SVM example.[1]

- *Approximate Sample Compression.* Let $f : \mathcal{Z}^* \mapsto \mathcal{F}$ be the regularized ERM with regularization $\lambda$ of a sample and let $C \subseteq S$ be a small compressed set of the sample. An algorithm that outputs $(1-\gamma) \cdot f(C \setminus U) + \gamma \cdot f(S) + \text{Lap}(2\gamma/\lambda\varepsilon)$ is a $\varepsilon$-system-aware unlearning algorithm. This follows from differentially private output perturbation (Chaudhuri et al., 2011; Dwork et al., 2014). We can interpolate between a sample compression scheme and a private model trained on the full dataset.

Further note that for all of the above examples, we only need to make an unlearning update when the set of deletions falls within the small set $S'$. This automatically gives us a large deletion capacity; points outside of $S'$ can always be deleted for free. Furthermore, the expected deletion time is small because computation only needs to be performed for a small number of points at the time of deletion. This ease of unlearning gives us an incentive to learn models that depend on a small number of samples. In Section 3, we exploit the fact that algorithms that rely on fewer samples while training are easier to unlearn.

## 3. System-Aware Unlearning Algorithm via Core-Sets

Various learning algorithms rely on core-sets. In this section, we show how traditional unlearning algorithms can build on core-sets to get system-aware unlearning algorithms. Let $\mathfrak{C}$ be a mapping that satisfies

$$\mathfrak{C}(\mathfrak{C}(S) \setminus U) = \mathfrak{C}(S) \setminus U,$$

for any $S$ and $U \subseteq S$. Furthermore, consider any unlearning algorithm $A_{\text{UN}}$, that induces the state-of-system $I_{A_{\text{UN}}}$, and satisfies the traditional definition of unlearning (Definition 2.3), i.e. we have:

$$\Pr(I_{A_{\text{UN}}}(S, U) \in F) \le e^\varepsilon \cdot \Pr(I_{A_{\text{UN}}}(S \setminus U, \varnothing) \in F) + \delta,$$

and,

$$\Pr(I_{A_{\text{UN}}}(S \setminus U, \varnothing) \in F) \le e^\varepsilon \cdot \Pr(I_{A_{\text{UN}}}(S, U) \in F) + \delta.$$

Consider the following unlearning algorithm, denoted by $A_{\text{CS}}$: Given a dataset $S$ and unlearning requests $U$, return the hypothesis $I_{A_{\text{UN}}}(\mathfrak{C}(S) \setminus U)$. This procedure enjoys the following guarantee:

**Theorem 3.1.** *The algorithm $A_{\text{CS}}$ (defined above) is a $(\varepsilon, \delta)$-system-aware unlearning algorithm with $S' = \mathfrak{C}(S)$, with the state-of-system defined as: $I_A(S, U) = I_{A_{\text{UN}}}(\mathfrak{C}(S), U)$.*

Particularly, $A_{\text{CS}}$ only needs to make unlearning updates for deletions inside the core set, i.e. when $\mathfrak{C}(S) \cap U \ne \varnothing$, otherwise we just return $A_{\text{UN}}(\mathfrak{C}(S), \varnothing)$. Importantly, for most points (which are not in $\mathfrak{C}(S)$), nothing needs to be done during unlearning. Furthermore, only $\mathfrak{C}(S)$ needs to be stored in the system. Thus, using core set algorithms, we can unlearn more efficiently in terms of computation time and memory compared to traditional unlearning.

## 4. Efficient Unlearning for Linear Classification via Selective Sampling

Linear classification is not only a fundamental problem in learning theory, but its use in practice continues to be widespread. For example, in large foundation models and generative models, the last layers of these models are often fine-tuned using linear probing, which trains a linear classifier on representations learned by a deep neural network (Belinkov, 2022; Kornblith et al., 2019). As unlearning gains increasing attention for fine-tuned large-scale ML models, the need for efficient unlearning algorithms for linear classification grows significantly. However, Cherapanamjeri et al. (2025) proved that under traditional unlearning definition, exact unlearning for linear classification requires storing the entire dataset. For today's large datasets, this makes exact unlearning under the traditional definition impractical, even for the simple setting of linear classification. Furthermore, approximate unlearning algorithms, such as the one from Sekhari et al. (2021), reduce to exact unlearning for linear classification. Our key insight in this section is that by using selective sampling as a core set algorithm, we can design an exact system-aware unlearning algorithm that requires memory that only scales sublinearly in the number of samples; recall that this is theoretically impossible under the traditional unlearning definition Cherapanamjeri et al. (2025). We also prove that the expected deletion time is significantly faster than traditional unlearning. Thus, demonstrating that system-aware unlearning algorithms are more efficient than traditional unlearning algorithms.

Selective sampling (Cesa-Bianchi et al., 2009; Dekel et al., 2012; Zhu & Nowak, 2022; Sekhari et al., 2023; Hanneke

---

[1]As seen in Figure 1, depending on the selection of $S'$, system-aware unlearning can lead to very different unlearning objectives compared to traditional unlearning, some of which may be easier to satisfy than the restrictive condition of traditional unlearning.

et al., 2014) is the problem of finding a classifier with small excess risk using the labels of only a few number of points. It has become particularly important as datasets become larger and labeling them becomes more expensive. Typically, selective sampling algorithms query the label of points whose label they are uncertain of and only update the model on points that they query. Thus, selective sampling-based algorithms can be seen as core set algorithms where the core set is the set of points where the label was queried. Under mild assumptions, selective sampling can achieve the optimal excess risk for linear classification with an exponentially small number of samples (Dekel et al., 2012).

**Assumptions.** We consider the problem of binary linear classification. Let $x \in \mathbb{R}^d$ be such that $\|x\| \le 1$ and $y \in \{+1, -1\}$. Furthermore, we assume that there exists a $u \in \mathbb{R}^d$, $\|u\| \le 1$ such that $\mathbb{E}[y_t \mid x_t] = u^\top x_t$. This is the realizability assumption for binary classification and ensures that the Bayes optimal predictor for $y_t$ is $\text{sign}(u^\top x_t)$. In the following sections, let $T := |S|$ and $N_T := |\mathfrak{C}(S)|$.

Our goal in linear classification is to find a hypothesis that performs well under 0-1 loss, i.e. we set $\ell(f(x), y) = \mathbb{1}\{f(x) \ne y\}$. With this goal in mind, we define the excess risk for a hypothesis $w$ as $\mathcal{E}(w) := \mathbb{E}_{(x,y)\sim\mathcal{D}} [\mathbb{1}\{\text{sign}(w^\top x) \ne y\} - \mathbb{1}\{\text{sign}(u^\top x) \ne y\}]$.

We use the selective sampling algorithm BBQSAMPLER from Cesa-Bianchi et al. (2009) to design our unlearning algorithm, given in Algorithm 1. In particular, our algorithm uses BBQSAMPLER to learn a predictor that only depends on a small number of queried points $\mathcal{Q}$. Let the core set $\mathfrak{C}(S)$ be the set of queried points $\mathcal{Q}$ when the BBQSAMPLER executes on $S$. The final predictor of Algorithm 1 after learning returns an ERM over $\mathfrak{C}(S)$ and stores $\mathfrak{C}(S)$. Then when unlearning $U$, Algorithm 1 updates the predictor to be an ERM over $\mathfrak{C}(S) \smallsetminus U$ and removes $U$ from memory. After unlearning, the model output and everything stored in memory only rely on $\mathfrak{C}(S) \smallsetminus U$.

**Theorem 4.1.** *Given the set $S$ and deletion requests $U$, let $\mathfrak{C}(S)$ denote the subset of points for which the labels were queried by BBQSAMPLER in Algorithm 1, and let $A(S, U)$ denote the unlearned model. Then, Algorithm 1 is an exact system-aware unlearning algorithm with $S' = \mathfrak{C}(S)$ and state-of-system $|_A(S, U) = (A(S, U), \mathfrak{C}(S) \smallsetminus U)$. In particular, it satisfies Definition 2.3 with $\varepsilon = \delta = 0$.*

The proof of Theorem 4.1 relies on a key attribute of the BBQSAMPLER—its query condition is *monotonic* with respect to deletion. In particular, a *monotonic* query condition is one such that for all datasets $S$ whose set of queried points is $\mathcal{Q}$, and deletions $U$, the selective sampling algorithm executed on $\mathcal{Q} \smallsetminus U$ queries every point in $\mathcal{Q} \smallsetminus U$.

**Theorem 4.2.** *For any dataset $S$, Algorithm 1 satisfies $\mathfrak{C}(\mathfrak{C}(S) \smallsetminus U) = \mathfrak{C}(S) \smallsetminus U$ for all $U \subseteq S$.*

---

**Algorithm 1** System-Aware Unlearning Algorithm for Linear Classification using Selective Sampling

**Require:**
- Dataset $S$ of size $T$
- Deletion requests $U$
- Deletion capacity $K > 0$
- Sampling parameter $0 \le \kappa \le 1$

1: **Function** BBQSAMPLER$(S, K, \kappa)$
2:     Initialize: $\lambda = K, w_0 = \vec{0}, A_0 = \lambda I, b_0 = \vec{0}, \mathcal{Q} = \varnothing$
3:     **for** each $t = 1, 2, \ldots, T$ **do**
4:         **if** $x_t^\top A_{t-1}^{-1} x_t > T^{-\kappa}$ **then**
5:             Query label $y_t$, and update
6:             $A_t \leftarrow A_{t-1} + x_t x_t^\top, b_t \leftarrow b_{t-1} + y_t x_t$
7:             $w_t \leftarrow A_t^{-1} b_t, \mathcal{Q} \leftarrow \mathcal{Q} \cup \{(x_t, y_t)\}$
8:         **else**
9:             Set $A_t \leftarrow A_{t-1}, b_t \leftarrow b_{t-1}, w_t \leftarrow w_{t-1}$
10:     **return** $\mathcal{Q}, A_T, b_T, w_T$

11: **Function** DELETIONUPDATE$(\mathcal{Q}, A, b, w, U)$
12:     **for** $(x, y) \in U$ such that $(x, y) \in \mathcal{Q}$ **do**
13:         Update $\mathcal{Q} \leftarrow \mathcal{Q} \smallsetminus \{x\}$
14:         Update $A \leftarrow A - xx^\top, b \leftarrow b - yx$ and $w \leftarrow A^{-1}b$
15:     **return** $\mathcal{Q}, X, b, w$

16: // Learn a predictor via selective sampling //
17: $\mathcal{Q}, A, b, w \leftarrow$ BBQSAMPLER$(S, \lambda, \kappa)$

18: // Update the predictor for core set deletions //
19: $\mathcal{Q}, A, b, w \leftarrow$ DELETIONUPDATE$(\mathcal{Q}, A, b, w, U)$
20: **return** $\text{sign}(w^\top x)$

---

The monotonicity of the query condition of the BBQSAMPLER stems from the fact that the query condition is only $x$-dependent and does not depend on the labels $y$ at all. We can interpret the query condition as testing if we have already queried many points that lie in the same direction as $x_t$, because if so, we can be fairly confident of the label of $x_t$. In particular, we have that if $x_t^\top A_t^{-1} x_t > T^{-\kappa}$ for some $t \in [T]$, then $x_t^\top A_{t \smallsetminus x_j}^{-1} x_t > T^{-\kappa}$ for any $j \in [T]$. Intuitively, the monotonicity of the query condition follows from the fact that if a direction was not well queried before deletion, it will not be well queried after deletion. $\mathfrak{C}(S) \smallsetminus U$ only contains queried points, all of which will be re-queried after deletion. Thus, we do not need to re-execute the BBQSAMPLER at the time of unlearning in order to determine the new set of queried points. We can simply remove the effect of $U$ from the predictor, and we only need to make an update for deletion requests in $U$ that are also in $\mathfrak{C}(S)$.

We note that not every selective sampling algorithm has $\mathfrak{C}(\mathfrak{C}(S) \smallsetminus U) = \mathfrak{C}(S) \smallsetminus U$. Various selective sampling algorithms, such as the ones from Dekel et al. (2012) or Sekhari et al. (2023), use a query condition that depends on the labels $y$. Due to the noise in these $y$'s, $y$-dependent query conditions are not monotonic; points that were queried before deletion can become unqueried after deletion. This makes it computationally expensive to compute the core set

after unlearning. We note that since the BBQSAMPLER uses a $y$-independent query condition, the predictor before unlearning is slightly suboptimal in terms of excess risk. However, we are willing to tolerate a small increase in excess risk in order to unlearn efficiently. Additionally, it is unclear how much the excess risk of $y$-dependent selective sampling algorithms would suffer after unlearning.

**Algorithm 1 is not a valid unlearning algorithm under the traditional definition (Definition 2.1).** When a queried point is deleted, an unqueried point could become queried, therefore, $\mathfrak{C}(S \smallsetminus U) \neq \mathfrak{C}(S) \smallsetminus U$. Thus, under traditional unlearning, during DELETIONUPDATE, not only would we have to remove the effect of $U$, but we would also have to add in any unqueried points that would have been queried if $U$ never existed in $S$. It is computationally inefficient to determine which points would have been queried, and it is unnecessary from a privacy perspective. An attacker could never have known that such an unqueried point existed and should have become queried after deletion since it was never used or stored by the original model. This highlights the need for having our proposed definition of unlearning.

**Theorem 4.3.** *The memory required by Algorithm 1 is determined by the number of core set points, which is bounded by $N_T = O(dT^\kappa \log T)$. Furthermore, with probability $1 - \delta$, the excess risk of the final predictor $\operatorname{sign}(w^\top x)$ returned by Algorithm 1 satisfies*

$$\mathcal{E}(w) = O\left(\frac{N_T \log T + \log(1/\delta)}{T - T_{\bar{\varepsilon}} - N_T}\right)$$

*after unlearning up to*

$$K = O\left(\frac{\bar{\varepsilon}^2 \cdot T^\kappa}{d \log T \cdot \log(1/\delta)}\right)$$

*many core set deletions, where $T_\varepsilon = \sum_{t=1}^T \mathbb{1}\{|u^\top x_t| \leq \varepsilon\}$, and $\bar{\varepsilon} > 0$ denotes the minimizing $\varepsilon$ in the regret bound in Theorem D.8.*

We remark that $T_\varepsilon$ represents the number of points where even the Bayes optimal predictor is unsure of the label, which we expect to be small in realistic scenarios. We give a proof sketch of the theorem (full proof in Appendix D).

**Proof Sketch.** The bound on the number of points queried by BBQSAMPLER is well known and can be derived using standard analysis for selective sampling algorithms from Cesa-Bianchi et al. (2009) and Dekel et al. (2012) (see Theorem D.8 for details). The number of queries made by the BBQSAMPLER is exactly the size of the core set.

To bound the excess risk, we first show that the final predictor $\hat{w} = w_T$ from the BBQSAMPLER before unlearning agrees with the Bayes optimal predictor on the classification of all the unqueried points outside the $T_{\bar{\varepsilon}}$ margin points. Let

$\tilde{w}$ be the predictor after $K$ core set deletions. We want to ensure that the signs of $\hat{w}$ and $\tilde{w}$ remain the same for all unqueried points. We do so by first demonstrating that $\hat{w}$ exhibits stability (Bousquet & Elisseeff, 2002; Shalev-Shwartz et al., 2010) on unqueried points. For any unqueried point $x$, $|\hat{w}^\top x - \tilde{w}^\top x| < \sqrt{K \cdot d \log T \cdot \log(1/\delta)} \cdot T^{-\kappa}$. Then we show that $\hat{w}$ has a $\bar{\varepsilon}/2$ margin on the classification of every unqueried point. Putting these together, we show that for up to $K \leq O\left(\frac{\bar{\varepsilon}^2 \cdot T^\kappa}{d \log T \cdot \log(1/\delta)}\right)$ deletions, we can ensure that the signs of $\hat{w}$ and $\tilde{w}$ agree on unqueried points. Thus, after unlearning, we can maintain correct classification on unqueried points with respect to the Bayes optimal predictor.

We cannot make any guarantees on the $N_T$ queried points and the $T_{\bar{\varepsilon}}$ margin points, so we assume full classification error on those points. Finally, using generalization bounds for sample compression (Kakade & Tewari, 2008), we convert the empirical classification loss to an excess risk bound. □

**Memory Required for Unlearning.** The memory required for unlearning is exactly the number of core set points, $O(dT^\kappa \log T)$, and the size of the model, $O(d^2)$, which is significantly less memory than storing the entirety of $S$ of size $T$. Under system-aware unlearning, we obtain the first exact unlearning algorithm for linear classification requiring memory sublinear in the size of the dataset.

**Deletion Capacity and Excess Risk.** Theorem 4.3 bounds the core set deletion capacity. Since $\kappa$ is a free parameter, we can tune it to increase the core set deletion capacity at the cost of increasing the excess risk after deletion. We are trading off deletion capacity at the cost of performance.

### 4.1. Expected Deletion Capacity

Notice that the deletion capacity bound in Theorem 4.1 only applies to core set deletions and does not account for samples in $U$ that are outside $\mathfrak{C}(S)$, and these can be deleted for free. Thus, we have a much larger deletion capacity than what is implied by the bound in Theorem 4.1. Assume that deletions are drawn without replacement from $\mu : \mathcal{X} \to [0, 1]$, a probability weight vector over points in $S$. This implies that the probability of $x$ requesting for deletion, i.e. $\mu(x)$, only depends on $x$ and not on its index within $S$ or on other points in $S$. This assumption is useful for capturing scenarios where the users make requests for deletion solely based on their own data and have no knowledge of where in the sample they appear. Let $K_{\text{TOTAL}}$ be the total number of deletions we can process under $\mu$ before exhausting the core set deletion capacity $K$.

**Theorem 4.4.** *Consider any core-set algorithm A. Let $\pi$ denote denote a uniformly random permutation of the samples in S, and let $\sigma$ be a sequence of deletion requests samples from $\mu$, without replacement. Further, let $K_{\text{CSD}}$ denote the number of core set deletions within the first $K_{\text{TOTAL}}$ deletion*

*requests, then for any $K \geq 1$,*

$$\Pr_{S,\pi,\sigma}(K_{\text{CSD}} > K) \leq \frac{1}{K} \mathbb{E}_S \Bigg[ \sum_{t=1}^{T} \mathbb{E}_\pi [\mathbb{1}\{x_t \in \mathfrak{C}_A(\pi(S))\}] \\ \cdot \sum_{k=1}^{K_{\text{TOTAL}}} \mathbb{E}_\sigma [\mathbb{1}\{x_t = x_{\sigma_k}\}] \Bigg],$$

*where $\mathfrak{C}_A(\pi(S))$ denotes the coreset resulting from running $A$ on the permuted dataset $\pi(S)$. Instantiating the above bound for [Algorithm 1](#), we have*

$$\Pr_{S,\pi,\sigma}(K_{\text{CSD}} > K) \leq \frac{K_{\text{TOTAL}} \cdot T^\kappa}{K} \cdot \mathbb{E}_S[\mathbb{E}_{x \sim \mu}[x^\top \overline{M} x]],$$

*where $\overline{M} := \mathbb{E}_\pi [\frac{1}{T} \sum_{s=1}^{T} A_{s-1}^{-1}]$ and $\kappa \in (0,1)$ is the sampling parameter from [Algorithm 1](#).*

Given the deletion distribution $\mu$, [Theorem 4.4](#) can be used to bound the total number of deletions $K_{\text{TOTAL}}$ that [Algorithm 1](#) can tolerate while ensuring that the probability of exhausting the core set deletion capacity $K$ is small. This is done by bounding $\mathbb{E}_S[\mathbb{E}_{x \sim \mu}[x^\top \overline{M} x]]$, given $\mu$. For a deletion $x$ drawn from $\mu$, $\mathbb{E}_S[\mathbb{E}_{x \sim \mu}[x^\top \overline{M} x]]$ can be interpreted as the expected value of the query condition $x^\top A_t^{-1} x$ when [Algorithm 1](#) encounters $x$ during learning. The bound on the total number of deletions $K_{\text{TOTAL}}$ depends inversely on $\mathbb{E}_S[\mathbb{E}_{x \sim \mu}[x^\top \overline{M} x]]$. When $\mathbb{E}_S[\mathbb{E}_{x \sim \mu}[x^\top \overline{M} x]]$ is small, $x$ is unlikely to be queried, and thus, [Algorithm 1](#) can tolerate a large number of deletions $K_{\text{TOTAL}}$ before exhausting its core set deletion capacity $K$. Furthermore, the query condition decreases as it encounters and queries more points. Thus, $\mathbb{E}_S[\mathbb{E}_{x \sim \mu}[x^\top \overline{M} x]]$ is decreasing as the sample size $T$ increases, and we would expect it to be small for large $T$. $x^\top \overline{M} x$ is maximized when $x$ lies in a direction which does not occur very often. Deletion distributions which place large weight on uncommon directions in $S$ will maximize $\mathbb{E}_S[\mathbb{E}_{x \sim \mu}[x^\top \overline{M} x]]$ and lead to smaller $K_{\text{TOTAL}}$.

**Lemma 4.5.** *Let the deletion distribution $\mu$ be the uniform distribution. In this case, the bound in [Theorem 4.4](#) implies that we can process a total of $K_{\text{TOTAL}} = \frac{c \cdot K \cdot T}{dT^\kappa \log T}$ deletions while ensuring that the probability of exhausting the core set deletion capacity $K$ is at most $c$.*

### 4.2. Expected Deletion Time

We can make a similar argument for the deletion time. At the time of unlearning, we only need to make an update when deleting a core set point. For all other points, there is no computation time for unlearning. Given $K_{\text{TOTAL}}$, which can be derived using [Theorem 4.4](#), we can give an expression for the expected time for deletion.

**Theorem 4.6.** *For a deletion distribution $\mu$, if a core set algorithm $A$ can tolerate up to $K_{\text{TOTAL}}$ deletions before exhausting the core set deletion capacity $K$,*

$$\mathbb{E}[\text{time per deletion}] \leq \frac{K}{K_{\text{TOTAL}}} \times \{\text{time per core set deletion}\}.$$

Updating the predictor of [Algorithm 1](#) for a core set deletion takes $O(d^2)$ time, using the Sherman-Morrison update ([Hager, 1989](#)). Thus, for [Algorithm 1](#), under a uniform deletion distribution, we have $\mathbb{E}[\text{time per deletion}] \leq \frac{d^3 T^\kappa \log T}{T}$, by plugging in $K_{\text{TOTAL}}$ from [Lemma 4.5](#). For large $T$, this is a significant improvement over an exact traditional unlearning algorithm that requires $O(d^2)$ time for each deletion.

**Remark 4.7.** For large $d$, the update time can be replaced by a quantity that depends on the eigenspectrum of the data's Gram matrix. Furthermore, since [Algorithm 1](#) updates an ERM on $\mathfrak{C}(S)$ to an ERM on $\mathfrak{C}(S) \smallsetminus U$, we can further speed up the update time for a core set deletion using gradient descent, which takes $O(d)$ time per update.

**Empirical Evaluation.** In [Appendix B](#), we compare the performance of [Algorithm 1](#) to some common unlearning procedures, from [Bourtoule et al. (2021)](#) and [Sekhari et al. (2021)](#), and exact retraining. Our experiments show that for linear classification, [Algorithm 1](#) can unlearn significantly faster with significantly less memory resources compared to other methods while maintaining comparable accuracy.

## 5. Extension to Efficient Unlearning for Classification with General Function Classes

Analyzing the structure of [Algorithm 1](#), we can begin to identify a general framework for unlearning for classification beyond linear functions. We select a core set using a monotonic selective sampler and perform regression on the core set to obtain our learned model. Then, during unlearning, if a core set point is deleted, we perform regression on the remaining core set points to obtain our unlearned model.

---

**Algorithm 2** System-Aware Unlearning Algorithm for General Classification using Selective Sampling

---

**Require:** • Dataset $S$ of size $T$
       • Deletion requests $U$
       • Function class $\mathcal{F}$
       • GENERALBBQSAMPLER from [Appendix E](#)

1: **Function** DELETIONUPDATE$(\mathcal{Q}, U, \mathcal{F})$
2:    **if** $U \cap \mathcal{Q} \neq \varnothing$ **then**
3:       Update $\mathcal{Q} \leftarrow \mathcal{Q} \smallsetminus U$
4:       $\hat{f} = \text{argmin}_{f \in \mathcal{F}} \sum_{(x_i, y_i) \in \mathcal{Q}} \left( \frac{1+y_i}{2} - f(x_i) \right)^2$
5:       **return** $\mathcal{Q}, \hat{f}$

6: // Learn a predictor via selective sampling //
7: $\mathcal{Q}, \hat{f} \leftarrow$ GENERALBBQSAMPLER$(S, \mathcal{F})$
8: // Update the predictor for core set deletions //
9: $\mathcal{Q}, \hat{f} \leftarrow$ DELETIONUPDATE$(\mathcal{Q}, U, \mathcal{F})$
10: **return** $\text{sign}(\hat{f}(x) - 1/2)$

---

**Assumptions.** We consider the problem of binary classification with a general model class. Let $x \in \mathcal{X}$ and

$y \in \{+1, -1\}$. We are given a class of models $\mathcal{F} = \{f : \mathcal{X} \rightarrow [0,1]\}$, and we assume that there exists a $f^* \in \mathcal{F}$ such that $\mathbb{E}[y_t \mid x_t] = f^*(x_t)$, where the Bayes optimal predictor is $\mathrm{sign}(f^*(x_t) - 1/2)$. We define the excess risk for a hypothesis $\hat{f}$ according to the 0-1 loss as $\mathcal{E}(\hat{f}) := \mathbb{E}_{(x,y)\sim\mathcal{D}}[\mathbb{1}\{\mathrm{sign}(\hat{f}(x) - \frac{1}{2}) \neq y\} - \mathbb{1}\{\mathrm{sign}(f^*(x) - \frac{1}{2}) \neq y\}]$.

It turns out that for a general $\mathcal{F}$, we can construct a version of the BBQSAMPLER, as shown by Gentile et al. (2022). The GENERALBBQSAMPLER is described in Appendix E.

**Theorem 5.1.** *Given the set $S$ and deletion requests $U$, let $\mathfrak{C}(S)$ denote the subset of points for which the labels were queried by* GENERALBBQSAMPLER *in Algorithm 2, and let $A(S, U)$ denote the unlearned model. Then, Algorithm 2 is an exact system-aware unlearning algorithm with $S' = \mathfrak{C}(S)$ and state-of-system $\mathsf{I}_A(S, U) = (A(S, U), \mathfrak{C}(S) \setminus U)$.*

Once again, the proof relies on the monotonicity of the GENERALBBQSAMPLER.

**Theorem 5.2.** *For any dataset $S$, Algorithm 2 has the property that for all $U \subseteq S$, $\mathfrak{C}(\mathfrak{C}(S) \setminus U) = \mathfrak{C}(S) \setminus U$.*

**Memory Required for Unlearning.** The memory required by Algorithm 2 is determined by the query complexity of the GENERALBBQSAMPLER, which depends on an eluder-dimension-like quantity of $\mathcal{F}$. The dimension $\mathfrak{D}(\mathcal{F}, S)$ of model class $\mathcal{F}$ projected onto sample $S$ is defined as:

$$\mathfrak{D}(\mathcal{F}, S) = \sup_\pi \sum_{t=1}^{T} \sup_{f,g \in \mathcal{F}} \frac{(f(x) - g(x))^2}{\sum_{i=1}^{t} (f(x_{\pi(i)}) - g(x_{\pi(i)}))^2 + 1},$$

where $\pi$ is a permutation on $[T]$ (Gentile et al., 2022).

$\mathfrak{D}(\mathcal{F}, S)$ is closely related to the eluder dimension (Russo & Van Roy, 2013) and the disagreement coefficient (Foster et al., 2020), two well-studied active learning complexity measures (see Gentile et al. (2022) for details). The connection between the memory complexity of unlearning and the query complexity of active learning has also been demonstrated in Ghazi et al. (2023); Cherapanamjeri et al. (2025).

**Theorem 5.3.** *Assume that, with probability $1 - \delta/T$, the ERM $\hat{f}$ in Algorithm 2 satisfies the bound $\sum_{t=1}^{T}(f^*(x_t) - \hat{f}(x_t))^2 \leq \mathfrak{R}(T, \delta)$. Then, the memory required by Algorithm 2 is bounded by*

$$N_T = O\left(\min_\varepsilon\left\{T_\varepsilon + \frac{\mathfrak{R}(T, \delta) \cdot \mathfrak{D}(\mathcal{F}, S)}{\varepsilon^2}\right\}\right),$$

To find bounds on the convergence rate $\mathfrak{R}(T, \delta)$ for the ERM, one can look into Yang & Barron (1999); Koltchinskii (2006); Liang et al. (2015). We expect $\mathfrak{R}(T, \delta)$ to be small. Thus, Theorem 5.3 implies that if a function class has small dimension $\mathfrak{D}(\mathcal{F}, S)$ (see Russo & Van Roy (2013); Foster

et al. (2020); Gentile et al. (2022) for examples), we have an efficient algorithm for unlearning. For example, linear functions have $\mathfrak{D}(\mathcal{F}, S) = O(d \log T)$.

**Excess Risk.** In general, bounding the excess risk after deletion for a generic $\mathcal{F}$ is hard. However, if the regression oracle of $\mathcal{F}$ is stable, Algorithm 2 maintains small excess risk after deletion. We formally define stability as follows:

**Definition 5.4** (Uniform Stability; Bousquet & Elisseeff (2002)). *Let $\hat{f}_S \in \mathcal{F}$ be the predictor returned by a learning algorithm $A$ on sample $S \in \mathcal{Z}^n$ and let $\hat{f}_{S \setminus i} \in \mathcal{F}$ be the predictor returned by $A$ on $S \setminus \{x_i\}$. The learning algorithm $A$ satisfies uniform stability with rate $\beta$ if for all $i \in [n]$, for all $z = (x, y) \in \mathcal{Z}$, $|\ell(\hat{f}_S(x), y) - \ell(\hat{f}_{S \setminus i}(x), y)| \leq \beta(n)$.*

**Theorem 5.5.** *If the regression oracle for $\mathcal{F}$ satisfies uniform stability under the squared loss with rate $\beta$, then with probability $1 - \delta$, the excess risk of the final predictor returned by Algorithm 2 satisfies*

$$\mathcal{E}(\hat{f}) = O\left(\frac{1}{T - N_T} \cdot (N_T \log T + \log(1/\delta))\right)$$

*after unlearning up to*

$$K = O\left(\frac{\sqrt{\mathfrak{R}(T, \delta)}}{\sqrt{N_T} \cdot \beta(N_T)}\right)$$

*many core set deletions.*

Typically, we have $\beta(N_T) \approx \frac{1}{N_T}$. Thus, the number of core set deletions that we can tolerate looks like $\sqrt{N_T \cdot \mathfrak{R}(T, \delta)}$. Similarly to the linear case, the proof follows by showing that the predictor after learning agrees with the classification of the Bayes optimal predictor on the unqueried points with some margin and then leveraging uniform stability to ensure that the predictor before and after unlearning continue to agree on the classification of the unqueried points.

## 6. Conclusion

We proposed a new definition for unlearning, called *system-aware unlearning*, that provides unlearning guarantees against an attacker who compromises the system after unlearning. We proved that system-aware unlearning generalizes traditional unlearning definitions and demonstrated that core set algorithms are a natural way to satisfy system-aware unlearning. By using less information, we expose less information to a potential attacker, leading to easier unlearning. To highlight the power of this viewpoint of unlearning, we show that selective sampling can be used to design a more memory and computation-time-efficient exact system-aware unlearning algorithm for classification. Looking forward, it would be interesting to explore how approximate system-aware unlearning ($\varepsilon, \delta \neq 0$) can lead to even faster and more memory-efficient unlearning algorithms.

## Acknowledgements

LL acknowledges support from the Cornell Bowers CIS-Linkedin Fellowship. AS thanks Claudio Gentile for helpful discussions. KS acknowledges support from the Cornell Bowers CIS-Linkedin Grant.

## Impact Statement

This paper presents work whose goal is to advance the field of Machine Learning. There are many potential societal consequences of our work, none which we feel must be specifically highlighted here.

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

# A. Related Work

Beyond the standard definition of machine unlearning from Sekhari et al. (2021); Guo et al. (2019), some other machine unlearning definitions have been proposed. Gupta et al. (2021) generalizes the machine unlearning definition from Sekhari et al. (2021); Guo et al. (2019) to handle adaptive requests. Chourasia & Shah (2023) proposes a data deletion definition under adaptive requesters which does not require indistinguishability from retraining from scratch. They require that the model after deletion be indistinguishable from a randomized mapping $\pi$ on $S$ with the deleted individual $z$ replaced. This definition assumes that the attacker does not have knowledge of the unlearning algorithm itself. If the data deletion requesters are non-adaptive, then $\pi$ can be replaced by the unlearning algorithm $A$, and we recover the standard definition of machine unlearning. However, in general, system-aware unlearning does not generalize this definition. Compared to system-aware unlearning, the data deletion definition from Chourasia & Shah (2023) makes the stronger assumption that the attacker has knowledge of every remaining individual, but the weaker assumption that the attacker does not have knowledge of the unlearning algorithm.

Neel et al. (2021) proposed a distinction between traditional unlearning and "perfect" unlearning. Under perfect unlearning, not only must the observable outputs of the unlearning algorithm be indistinguishable from the retrained-from-scratch model, but the complete internal state of the unlearning algorithm must be indistinguishable from the retrained-from-scratch state. We note that system-aware unlearning when $S' = S$ is exactly equivalent to perfect unlearning. Golatkar et al. (2020) proposed a definition of unlearning that requires the existence of a certificate of forgetting, where the certificate can be any function that does not depend on the deleted individuals, rather than the fixed certificate of retraining-from-scratch. This is akin to requiring the existence of a $S' \subseteq S$ in the definition of system-aware unlearning, rather than fixing $S' = S$. We note that our system-aware algorithm is able to leverage the flexibility in $S'$ to achieve more efficient unlearning, while the algorithms in Golatkar et al. (2020) ultimately do attempt to recover a hypothesis close to retraining-from-scratch; however, we find the connections in the definitions interesting to point out.

Maintaining privacy under system-aware unlearning is closely related to the goal of pan-privacy (Dwork et al., 2010; Amin et al., 2020; Cheu & Ullman, 2020). In the setting of pan-privacy, user data is processed in a streaming fashion, outputs are produced in a sequence, and an adversary may compromise the internal state of the algorithm at any point during the stream. The goal of pan-privacy is provide privacy against an adversary who compromises the internal state at any point in the stream and has access to the preceding outputs.

Beyond unlearning definitions, there has been much work in the development of certified unlearning algorithms. The current literature generally falls into two categories: exact unlearning algorithms which exactly reproduce the model from retraining from scratch on $S \setminus U$ (Ghazi et al., 2023; Cherapanamjeri et al., 2025; Bourtoule et al., 2021; Cao & Yang, 2015; Chowdhury et al., 2024) or approximate unlearning algorithms which use ideas from differential privacy (Dwork et al., 2014) to probabilistically recover a model that is "essentially indistinguishable" from the model produced from retraining from scratch on $S \setminus U$ (Izzo et al., 2021; Sekhari et al., 2021; Chien et al., 2024; Guo et al., 2019). The exact unlearning algorithms are typically memory intensive and require the storage of the entire dataset and multiple models, while the approximate unlearning algorithms tend to be more memory efficient. Certified machine unlearning algorithms meet provable guarantees of unlearning. However, many of the algorithms are limited to the convex setting.

There are a number of (uncertified) unlearning algorithms which have been shown to work well empirically in the nonconvex setting (Goel et al., 2023; Kurmanji et al., 2023; Jang et al., 2022). Furthermore, a number of these empirical methods attempt to unlearn in a "data-free" manner where the remaining individuals are not stored in memory when unlearning (Foster et al., 2023; Bonato et al., 2024). However, recent work has shown that these empirical methods do not unlearn properly and do indeed leak the privacy of the unlearned individuals (Hayes et al., 2024; Pawelczyk et al., 2025).

Furthermore, existing lower bounds prove that there exist simple model classes with finite VC and Littlestone dimension where traditional exact unlearning requires the storage of the entire dataset (Cherapanamjeri et al., 2025). For large datasets, this makes exact unlearning under the traditional definition impractical. Additionally, Cherapanamjeri et al. (2025) proved that even approximate algorithms for certain model classes require the storage of the entire dataset for the hypothesis testing problem after deletion. This provides strong evidence that even approximate learning under the traditional definition requires storing the entire dataset.

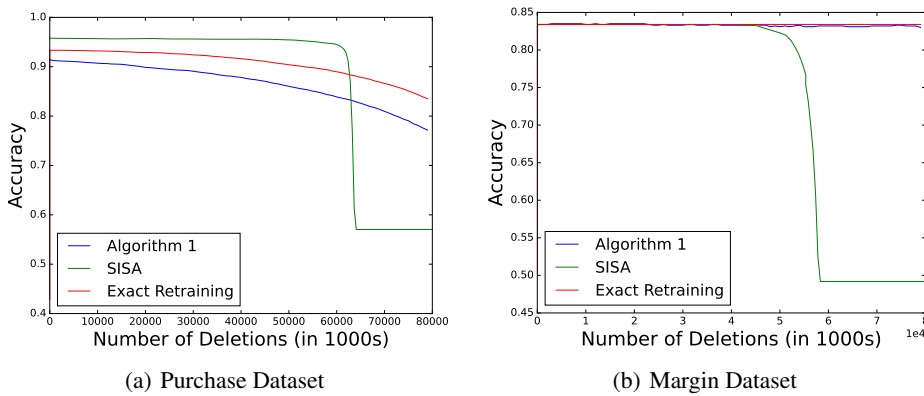

(a) Purchase Dataset

(b) Margin Dataset

*Figure 2.* The test accuracy of each unlearning method over the course 80,000 label dependent deletions.

## B. Experimental Evaluation

Our theoretical results provide guarantees for the worst case deletions. We experimentally verify our theory, and we demonstrate that in practice, Algorithm 1 can maintain small excess error beyond the core set deletion capacities proven in Theorem 4.1. Furthermore, Algorithm 1 is significantly more memory and computation time efficient compared to other unlearning methods for linear classification.

We compare the accuracy, memory usage, training time, and unlearning time of Algorithm 1 to the following unlearning procedures. Algorithm 1 trains a linear model ($y = Ax + b$) on the core set $\mathfrak{C}(S)$.

- *SISA from Bourtoule et al. (2021):* SISA trains models on separate data shards and aggregates them together to produce a final model. During the initial training process, SISA stores intermediate models in order to speed up retraining at the time of unlearning. We train a linear model ($y = w^\top x$) on each shard and aggregate the models using a uniform voting rule.
- *Exact Retraining:* We train a linear model ($y = w^\top x$) on the entire dataset and perform an exact retraining update for each deletion.
- *Unlearning Algorithm from Sekhari et al. (2021):* For linear regression ($y = w^\top x$), this algorithm precisely reduces to exact retraining.

We compare the results on two different datasets.

- *Purchase Dataset:* A binary classification dataset on item purchase data curated by Bourtoule et al. (2021) with 249,215 points in dimension $d = 600$.
- *Margin Dataset:* A synthetic binary classification dataset with 200,000 points in dimension $d = 100$ with a hard margin condition of $\gamma = 0.1$ ($|u^\top x| > 0.1, \forall x$ for some underlying $u \in \mathbb{R}^d$).

For each method, we train an initial linear classifier, and then we process a sequence of 80,000 deletions (around $40\%$ of the dataset) of points all with class label $-1$ (we refer to this as a sequence of *label dependent deletions*). Figure 2 compares the accuracy of the various unlearning methods over the sequence of deletions, and Table 1 and Table 2 compare the computation time and memory usage of the various methods on the two datasets.

*Table 1.* Computation and Memory Usage on the Purchase Dataset

|  | Initial training time (secs) | Accumulated deletion time (secs) | % of data stored in memory |
|---|---|---|---|
| Algorithm 1 | 186.3 | 58.3 | 13.1% |
| SISA | 30.2 | 1174.3 | 100% |
| Exact Retraining | 1828.7 | 579.3 | 100% |

On the Purchase Dataset, first observe that Algorithm 1 can maintain comparable accuracy to exact retraining while dominating exact retraining in initial training time, deletion time, and memory usage. SISA has the best initial accuracy, but

*Table 2.* Computation and Memory Usage on the Margin Dataset

|  | Initial training time (secs) | Accumulated deletion time (secs) | % of data stored in memory |
|---|---|---|---|
| Algorithm 1 | 1.3 | 0.5 | 0.6% |
| SISA | 20.6 | 697.1 | 100% |
| Exact Retraining | 67.8 | 27.1 | 100% |

Algorithm 1 is able to maintain significantly better accuracy under a longer sequence of label dependent deletions compared to SISA. The sequence of label dependent deletions creates a distribution shift in the training data. Algorithm 1 is robust to this shift due to theoretical guarantees, but SISA lacks such theoretical guarantees on its accuracy after unlearning. We also note that SISA is training a more expressive model compared to Algorithm 1 and exact retraining which could be contributing to its improved initial accuracy. Furthermore, Algorithm 1 can maintain comparable accuracy with significantly fewer samples. Algorithm 1 has a longer initial training time, but requires significantly less computation time at the time of deletion compared to SISA. The use of sample compression leads to more efficient unlearning.

When the dataset allows for a more favorable compression scheme, as the Margin Dataset does, the improvements are even more pronounced. Algorithm 1 can match the initial accuracy of SISA and exact retraining, despite using much less data, and Algorithm 1 can maintain significantly better accuracy under a longer sequence of label dependent deletions. Furthermore, Algorithm 1 requires significantly less memory and significantly less computation time, both at the time of training and the time of deletion, compared to SISA and exact retraining due to increased sample compression. When the dataset allows for significant compression, Algorithm 1 dominates SISA and exact retraining in accuracy, memory, and computation time.

## C. Proofs from Section 3

**Theorem 3.1.** *The algorithm $A_{CS}$ (defined above) is a $(\varepsilon, \delta)$-system-aware unlearning algorithm with $S' = \mathfrak{C}(S)$, with the state-of-system defined as: $\mathsf{I}_A(S, U) = \mathsf{I}_{A_{\mathrm{UN}}}(\mathfrak{C}(S), U)$.*

**Proof.** We have $\mathsf{I}_A(S, U) = \mathsf{I}_{A_{\mathrm{UN}}}(\mathfrak{C}(S), U)$.

We also have

$$
\begin{aligned}
\mathsf{I}_A(S' \smallsetminus U, \varnothing) &= \mathsf{I}_A(\mathfrak{C}(S) \smallsetminus U, \varnothing) \\
&= \mathsf{I}_{A_{\mathrm{UN}}}(\mathfrak{C}(\mathfrak{C}(S) \smallsetminus U), \varnothing) \\
&= \mathsf{I}_{A_{\mathrm{UN}}}(\mathfrak{C}(S) \smallsetminus U, \varnothing).
\end{aligned}
$$

By definition, $\mathsf{I}_{A_{\mathrm{UN}}}(\mathfrak{C}(S), U)$ and $\mathsf{I}_{A_{\mathrm{UN}}}(\mathfrak{C}(S) \smallsetminus U, \varnothing)$ are $(\varepsilon, \delta)$-indistinguishable, so $\mathsf{I}_A(S, U)$ and $\mathsf{I}_A(S' \smallsetminus U, \varnothing)$ must be $(\varepsilon, \delta)$-indistinguishable. Thus, $A$ is an $(\varepsilon, \delta)$-system-aware unlearning algorithm. $\square$

## D. Proofs from Section 4

### D.1. Notation

- $[n] = \{1, 2, \ldots, n\}$

- $A_T = \lambda I + \sum_{t=1}^{T} x_t x_t^\top$

- $A_{T \smallsetminus U} = \lambda I + \sum_{t=1}^{T} x_t x_t^\top - \sum_{x_i \in U} x_i x_i^\top$, where $U$ is a set of deletions

- $A_{t \smallsetminus x_j} = \begin{cases} \lambda I + \sum_{t=1}^{T} x_t x_t^\top - x_j x_j^\top & \text{when } j \le t \\ \lambda I + \sum_{t=1}^{T} x_t x_t^\top & \text{otherwise} \end{cases}$,
  for some $t, j \in [T]$

- $A_S = \lambda I + \sum_{x_t \in S} x_t x_t^\top$, where $S$ is a set of points

- $b_T = \sum_{t=1}^{T} y_t x_t$

- $b_T = \sum_{t=1}^{T} y_t x_t - \sum_{x_i \in U} y_i x_i$, where $U$ is a set of deletions

- $w_T = A_T^{-1} b_T$

- $w_{T \smallsetminus U} = A_{T \smallsetminus U}^{-1} b_{T \smallsetminus U}$, where $U$ is a set of deletions

- $\|u\|_X = u^\top X u$, where $u \in \mathbb{R}^d$ and $X \in \mathbb{R}^{d \times d}$

**Theorem 4.1.** *Given the set $S$ and deletion requests $U$, let $\mathfrak{C}(S)$ denote the subset of points for which the labels were queried by* BBQSAMPLER *in Algorithm 1, and let $A(S, U)$ denote the unlearned model. Then, Algorithm 1 is an exact system-aware unlearning algorithm with $S' = \mathfrak{C}(S)$ and state-of-system $\mathsf{I}_A(S, U) = (A(S, U), \mathfrak{C}(S) \smallsetminus U)$. In particular, it satisfies Definition 2.3 with $\varepsilon = \delta = 0$.*

**Proof.** Fix any sample $S$ and set of deletions $U$. Define $S' = \mathfrak{C}(S) = \mathcal{Q}$. Clearly, $S' \subseteq S$. The core set of the BBQSAMPLER is exactly the set of points that it queries. Thus, applying Theorem 4.2, we know $\mathfrak{C}(\mathfrak{C}(S) \smallsetminus U) = \mathfrak{C}(S) \smallsetminus U$. $A(S' \smallsetminus U, \varnothing)$ returns a regularized ERM over $\mathfrak{C}(\mathfrak{C}(S) \smallsetminus U)$, which is exactly $\mathfrak{C}(S) \smallsetminus U$, and stores that ERM and the set $\mathfrak{C}(S) \smallsetminus U$. To process the deletion of $U$, $A(S, U)$ returns a regularized ERM over $\mathfrak{C}(S) \smallsetminus U$ and stores that ERM and the set $\mathfrak{C}(S) \smallsetminus U$. Thus, $\mathsf{I}_A(S, U) = \mathsf{I}_A(S' \smallsetminus U, \varnothing)$ for all $U \subseteq S$. $\square$

**Theorem 4.2.** *For any dataset $S$, Algorithm 1 satisfies $\mathfrak{C}(\mathfrak{C}(S) \smallsetminus U) = \mathfrak{C}(S) \smallsetminus U$ for all $U \subseteq S$.*

**Proof.** Fix any sample $S$, and consider the deletion of a point $x_j \in U$. Consider executing the BBQSAMPLER on the queried points $\mathcal{Q} \setminus \{x_j\}$ compared to executing on $S$. First, observe that the removal of all of the unqueried points has no effect on any of the query conditions, $x_t^\top A_{t \setminus x_j}^{-1} x_t = x_t^\top A_t^{-1} x_t > T^{-\kappa}$. Thus, when $x_j$ is unqueried, all of the points in $\mathcal{Q} \setminus \{x_j\}$ will be queried when executing the BBQSAMPLER on the queried points $\mathcal{Q} \setminus \{x_j\}$

Next, consider the case that the deleted point $x_j$ was queried. Consider a point $x_t$ that was queried at time $t$. We know $x_t^\top A_t^{-1} x_t > T^{-\kappa}$. We note that any points after time $t$ do not affect the query condition at time $t$, so we only focus on deletions of $x_j$ where $j < t$. We have that

$$
\begin{aligned}
x_t^\top A_{t \setminus x_j}^{-1} x_t &= x_t^\top (A_t - x_j x_j^\top)^{-1} x_t \\
&= x_t^\top A_t^{-1} x_t + \left( \frac{x_t^\top A_t^{-1} x_j x_j^\top A_t^{-1} x_t}{1 - x_j^\top A_t^{-1} x_j} \right) \\
&= x_t^\top A_t^{-1} x_t + \frac{(x_t^\top A_t^{-1} x_j)^2}{1 - x_j^\top A_t^{-1} x_j} \\
&\geq x_t^\top A_t^{-1} x_t \\
&\geq T^{-\kappa},
\end{aligned}
$$

where the second to last line follows because the second term is always positive. Thus, $x_t$ remains queried when executing the BBQSAMPLER on $\mathcal{Q} \setminus \{x_j\}$. We can apply the above argument inductively for each $x_j \in U$ to conclude that $\mathfrak{C}(\mathfrak{C}(S) \setminus U) = \mathfrak{C}(S) \setminus U$. $\qquad\square$

**Theorem 4.3.** *The memory required by Algorithm 1 is determined by the number of core set points, which is bounded by $N_T = O(dT^\kappa \log T)$. Furthermore, with probability $1 - \delta$, the excess risk of the final predictor $\mathrm{sign}(w^\top x)$ returned by Algorithm 1 satisfies*

$$
\mathcal{E}(w) = O\left( \frac{N_T \log T + \log(1/\delta)}{T - T_{\bar\varepsilon} - N_T} \right)
$$

*after unlearning up to*

$$
K = O\left( \frac{\bar\varepsilon^2 \cdot T^\kappa}{d \log T \cdot \log(1/\delta)} \right)
$$

*many core set deletions, where $T_\varepsilon = \sum_{t=1}^T \mathbb{1}\{|u^\top x_t| \leq \varepsilon\}$, and $\bar\varepsilon > 0$ denotes the minimizing $\varepsilon$ in the regret bound in Theorem D.8.*

**Proof.** The bound on the number of points queried by the BBQ sampler $|\mathfrak{C}(S)|$ is given by Theorem D.8 using standard analysis for selective sampling algorithms from Cesa-Bianchi et al. (2009); Dekel et al. (2012); Agarwal (2013).

First let's set all of the $T_{\bar\varepsilon}$ margin points aside. Let $w_T$ be the last predictor from the BBQSAMPLER.

First we argue that before deletion, $w_T^\top x$ and $u^\top x$ agree on the sign of all unqueried points $x$ (outside of the $T_{\bar\varepsilon}$ margin points). An unqueried point $x_t$ must have a margin of $\bar\varepsilon$ with respect to $u$, which means $|u^\top x| > \bar\varepsilon$. For an unqueried point $x_t$, we also have

$$
\begin{aligned}
|w_T^\top x_t - u^\top x_t| &= \|w_T - u\|_{A_T} \cdot \|x_t\|_{A_T} \\
&\leq \|w_T - u\|_{A_T} \cdot \|x\|_{A_t} \qquad \text{(using the monotonicity of the query condition)} \\
&\leq \sqrt{d \log T \cdot \log(1/\delta)} \cdot T^{-\kappa} \\
&\qquad \text{(applying Proposition G.1 on the first term and the query condition on the second term)} \\
&\leq \frac{\bar\varepsilon}{2}. \qquad \text{(for sufficiently large T)}
\end{aligned}
$$

Thus, $\mathrm{sign}(w_T^\top x) = \mathrm{sign}(u^\top x)$, so the final predictor after learning $w_T$ and the Bayes optimal predictor $u$ agree on the classification of all of the unqueried points. Furthermore, all of the unqueried points $x$ have a margin of $\frac{\bar\varepsilon}{2}$ with respect to $w_T$.

Thus, in order to ensure that $w_T$ and $w_{T \setminus U}$ after $|U| = K$ deletions continue to agree on the classification of all unqueried points, we need to ensure that $|w_T^\top x - w_{T \setminus U}^\top x| = \Delta < \frac{\bar{\varepsilon}}{2}$. Using the upper bound on $\Delta$ derived using a stability analysis in Theorem D.2 , we get the following deletion capacity on queried points,

$$\Delta \leq 2\sqrt{e(K+1)} \cdot T^{-\kappa/2} \cdot \sqrt{d \log T \cdot \log(1/\delta)} \leq \frac{\bar{\varepsilon}}{2} \qquad \text{(Theorem D.2)}$$

$$e(K+1) \cdot T^{-\kappa} \cdot d \log T \cdot \log(1/\delta) \leq \frac{\bar{\varepsilon}^2}{16}$$

$$K + 1 \leq \frac{\bar{\varepsilon}^2 \cdot T^\kappa}{16e \cdot d \log T \cdot \log(1/\delta)}$$

$$K \leq \frac{\bar{\varepsilon}^2 \cdot T^\kappa}{16e \cdot d \log T \cdot \log(1/\delta)} - 1$$

$$K \leq O\left(\frac{\bar{\varepsilon}^2 \cdot T^\kappa}{d \log T \cdot \log(1/\delta)}\right).$$

For up to $K$ deletions on queried points, $w_T$ and $w_{T \setminus U}$ are guaranteed to agree on the classification of all unqueried points. Thus after unlearning up to $K$ queried points, $w_{T \setminus U}$ and the Bayes optimal predictor $u$ agree on the classification on all of the unqueried points. Note that $w_{T \setminus U}$ is a predictor that only used points in $\mathcal{Q} \setminus U$ during training, and yet, has good performance on points that it never used during training. In particular, we have that

$$\hat{L}_{S \setminus \{\mathcal{Q} \setminus U\}}(w_{T \setminus U}) = \sum_{(x,y) \in S \setminus \{\mathcal{Q} \setminus U\}} \mathbb{1}\{\text{sign}(u^\top x) \neq \text{sign}(w_{T \setminus U}^\top x)\} \leq T_{\bar{\varepsilon}} + K,$$

because outside of the $T_{\bar{\varepsilon}}$ margin points, we showed that $u$ and $w_{T \setminus U}$ agree on the classification of all of the unqueried points. Furthermore, $u$ and $w_{T \setminus U}$ may disagree on the classification of the $K$ deleted queried points.

Through this observation, we can use techniques from generalization for sample compression algorithms (Kakade & Tewari, 2008) to convert the empirical classification loss to an excess risk bound for $w_{T \setminus U}$. First, observe that

$$\mathcal{E}(w_{T \setminus U}) = \mathbb{E}_{(x,y) \sim \mathcal{D}}\big[\mathbb{1}\{\text{sign}(w_{T \setminus U}^\top x) \neq y\} - \mathbb{1}\{\text{sign}(u^\top x) \neq y\}\big]$$

$$= \mathbb{E}_{(x,y) \sim \mathcal{D}}\big[|2|u^\top x| - 1| \cdot \mathbb{1}\{\text{sign}(w_{T \setminus U}^\top x) \neq \text{sign}(u^\top x)\}\big]$$

$$\leq \mathbb{E}_{(x,y) \sim \mathcal{D}}\big[\mathbb{1}\{\text{sign}(w_{T \setminus U}^\top x) \neq \text{sign}(u^\top x)\}\big]$$

Thus, we look to bound the loss of $L(w_{T \setminus U}) = \mathbb{E}_{(x,y) \sim \mathcal{D}}\big[\mathbb{1}\{\text{sign}(w_{T \setminus U}^\top x) \neq \text{sign}(u^\top x)\}\big]$. We are interested in the event that there exists a $\mathcal{Q} \setminus U \subseteq S$, $|\mathcal{Q} \setminus U| = l$ such that $\hat{L}_{S \setminus \{\mathcal{Q} \setminus U\}}(w_{T \setminus U}) \leq T_{\bar{\varepsilon}} + K$ and $L(\hat{f}_{\mathcal{Q} \setminus U}) \geq \varepsilon$.

$$\Pr[\exists\, \mathcal{Q} \setminus U \subseteq S \text{ such that } \hat{L}_{S \setminus \{\mathcal{Q} \setminus U\}}(w_{T \setminus U}) \leq T_{\bar{\varepsilon}} + K \text{ and } L(w_{T \setminus U}) \geq \varepsilon]$$

$$\leq \sum_{l=1}^{T} \Pr[\exists\, \mathcal{Q} \setminus U \subseteq S, |\mathcal{Q} \setminus U| = l \text{ such that } \hat{L}_{S \setminus \{\mathcal{Q} \setminus U\}}(w_{T \setminus U}) \leq T_{\bar{\varepsilon}} + K \text{ and } L(w_{T \setminus U}) \geq \varepsilon]$$

$$\leq \sum_{l=1}^{T} \sum_{\mathcal{Q} \setminus U \subseteq S, |\mathcal{Q} \setminus U| = l} \Pr[\hat{L}_{S \setminus \{\mathcal{Q} \setminus U\}}(w_{T \setminus U}) \leq T_{\bar{\varepsilon}} + K \text{ and } L(w_{T \setminus U}) \geq \varepsilon]$$

$$= \sum_{l=1}^{T} \sum_{\mathcal{Q} \setminus U \subseteq S, |\mathcal{Q} \setminus U| = l} \mathbb{E}\big[\Pr_{S \setminus \{\mathcal{Q} \setminus U\}}[\hat{L}_{S \setminus \{\mathcal{Q} \setminus U\}}(w_{T \setminus U}) \leq T_{\bar{\varepsilon}} + K \text{ and } L(w_{T \setminus U}) \geq \varepsilon \mid \mathcal{Q} \setminus U]\big]$$

Let $|\mathcal{Q}| = N_T$ and let $|U| = K$, where $N_T - K = l$. Now for any fixed $\mathcal{Q} \setminus U$, the above probability is just the probability of having a true risk greater than $\varepsilon$ and an empirical risk at most $T_{\bar{\varepsilon}} + K$ on a test set of size $T - N_T + K$. Now for any random variable $z \in [0, 1]$, if $\mathbb{E}[z] \geq \varepsilon$ then $\Pr[z = 0] \leq 1 - \varepsilon$. Thus, for a given $\mathcal{Q} \setminus U$,

$$\Pr_{S \setminus \{\mathcal{Q} \setminus U\}}[\hat{L}_{S \setminus \{\mathcal{Q} \setminus U\}}(w_{T \setminus U}) \leq T_{\bar{\varepsilon}} + K \text{ and } L(w_{T \setminus U}) \geq \varepsilon] \leq (1 - \varepsilon)^{T - N_T - T_{\bar{\varepsilon}}}$$

Plugging this in above, we have

$$\Pr[\exists\, \mathcal{Q} \setminus U \subseteq S, |\mathcal{Q} \setminus U| = l \text{ such that } \hat{L}_{S \setminus \{\mathcal{Q} \setminus U\}}(w_{T \setminus U}) \leq T_{\bar{\varepsilon}} + K \text{ and } L(w_{T \setminus U}) \geq \varepsilon]$$

$$\leq \sum_{l=1}^{T} \sum_{\mathcal{Q} \setminus D \subseteq S, |\mathcal{Q} \setminus D| = l} (1-\varepsilon)^{T-N_T-T_{\bar{\varepsilon}}}$$

$$\leq \sum_{l=1}^{T} T^l \cdot (1-\varepsilon)^{T-N_T-T_{\bar{\varepsilon}}}$$

$$= \sum_{l=1}^{T} T^{N_T-K} \cdot (1-\varepsilon)^{T-N_T-T_{\bar{\varepsilon}}}$$

$$= \sum_{l=1}^{T} T^{N_T-K} \cdot (1-\varepsilon)^{T-N_T-T_{\bar{\varepsilon}}}$$

$$\leq \sum_{l=1}^{T} T^{N_T} \cdot e^{-\varepsilon(T-N_T-T_{\bar{\varepsilon}})}$$

$$= T^{N_T+1} \cdot e^{-\varepsilon(T-N_T-T_{\bar{\varepsilon}})}$$

We want this probability to be at most $\delta$. Setting $\varepsilon$ appropriately, we have

$$\varepsilon = \frac{1}{T - N_T - T_{\bar{\varepsilon}}} \cdot ((N_T + 1) \log T + \log(1/\delta))$$

Thus, with probability at least $1 - \delta$,

$$L(\hat{f}_{\mathcal{Q} \setminus D}) \leq \frac{1}{T - N_T - T_{\bar{\varepsilon}}} \cdot ((N_T + 1) \log T + \log(1/\delta))$$

This implies that with probability at least $1 - \delta$,

$$\mathcal{E}(w_{T \setminus U}) \leq \frac{1}{T - N_T - T_{\bar{\varepsilon}}} \cdot ((N_T + 1) \log T + \log(1/\delta)).$$

$\square$

**Theorem 4.4.** *Consider any core-set algorithm $A$. Let $\pi$ denote denote a uniformly random permutation of the samples in $S$, and let $\sigma$ be a sequence of deletion requests samples from $\mu$, without replacement. Further, let $K_{\mathrm{CSD}}$ denote the number of core set deletions within the first $K_{\mathrm{TOTAL}}$ deletion requests, then for any $K \geq 1$,*

$$\Pr_{S,\pi,\sigma}(K_{\mathrm{CSD}} > K) \leq \frac{1}{K} \mathbb{E}_S \Bigg[ \sum_{t=1}^{T} \mathbb{E}_\pi [\mathbb{1}\{x_t \in \mathfrak{C}_A(\pi(S))\}] \\ \cdot \sum_{k=1}^{K_{\mathrm{TOTAL}}} \mathbb{E}_\sigma [\mathbb{1}\{x_t = x_{\sigma_k}\}] \Bigg],$$

*where $\mathfrak{C}_A(\pi(S))$ denotes the coreset resulting from running $A$ on the permuted dataset $\pi(S)$. Instantiating the above bound for [Algorithm 1](#), we have*

$$\Pr_{S,\pi,\sigma}(K_{\mathrm{CSD}} > K) \leq \frac{K_{\mathrm{TOTAL}} \cdot T^\kappa}{K} \cdot \mathbb{E}_S[\mathbb{E}_{x \sim \mu}[x^\top \overline{M} x]],$$

*where $\overline{M} := \mathbb{E}_\pi [\frac{1}{T} \sum_{s=1}^{T} A_{s-1}^{-1}]$ and $\kappa \in (0, 1)$ is the sampling parameter from [Algorithm 1](#).*

**Proof.** We begin by considering

$$\Pr_{S,\pi,\sigma}(K_{\mathrm{CSD}} > K) \leq \frac{1}{K} \mathbb{E}[K_{\mathrm{CSD}}] \qquad \text{(Markov's Inequality)}$$

$$= \frac{1}{K} \mathbb{E}_{S,\pi,\sigma} \Bigg[ \sum_{t=1}^{T} \mathbb{1}\{x_t \in C_\pi\} \cdot \sum_{k=1}^{K_{\mathrm{TOTAL}}} \mathbb{1}\{x_t = x_{\sigma_k}\} \Bigg]$$

$$\text{($C_\pi$ is the resulting core set after executing on $\pi(S)$)}$$

$$= \frac{1}{K} \mathbb{E}_S \left[ \sum_{t=1}^{T} \mathbb{E}_\pi \left[ \mathbb{1}\{x_t \in C_\pi\} \right] \cdot \sum_{k=1}^{K_{\text{TOTAL}}} \mathbb{E}_\sigma \left[ \mathbb{1}\{x_t = x_{\sigma_k}\} \right] \right]$$

$$= \frac{1}{K} \mathbb{E}_{S,\sigma} \left[ \sum_{t=1}^{T} \mathbb{E}_\pi \left[ \mathbb{1}\{x_t \in C_\pi\} \right] \cdot \sum_{k=1}^{K_{\text{TOTAL}}} \mathbb{1}\{x_t = x_{\sigma_k}\} \right]$$

$$= \frac{1}{K} \mathbb{E}_{S,\sigma} \left[ \sum_{k=1}^{K_{\text{TOTAL}}} \mathbb{E}_\pi \left[ \mathbb{1}\{x_{\sigma_k} \in C_\pi\} \right] \right].$$

This proves the first half of the theorem.

Next define $\nu(x) = \mathbb{E}_\pi[\mathbb{1}\{x \in C_\pi\}]$. Consider the case when the deletion distribution $\mu$ satisfies $\mu(x) > \mu(x') \implies \nu(x) \geq \nu(x')$. This is exactly the worst case in terms of deletion capacity: points that have a high probability of being included in the core set are exactly the points that have a high probability of being deleted.

In this case, we can apply Theorem G.2 to get

$$\Pr_{S,\pi,\sigma}(X > K) \leq \frac{1}{K} \mathbb{E}_{S,\sigma} \left[ \sum_{k=1}^{K_{\text{TOTAL}}} \mathbb{E}_\pi \left[ \mathbb{1}\{x_{\sigma_k} \in C_\pi\} \right] \right]$$

$$\leq \frac{1}{K} \mathbb{E}_S \left[ \mathbb{E}_{x \sim \mu} \sum_{k=1}^{K_{\text{TOTAL}}} \left[ \mathbb{E}_\pi \left[ \mathbb{1}\{x \in C_\pi\} \right] \right] \right] \qquad \text{(where } x \text{ is sampled without replacement from } W\text{)}$$

$$\leq \frac{K_{\text{TOTAL}}}{K} \mathbb{E}_S \left[ \mathbb{E}_{x \sim \mu} \left[ \mathbb{E}_\pi \left[ \mathbb{1}\{x \in C_\pi\} \right] \right] \right]$$

$$\leq \frac{K_{\text{TOTAL}}}{K} \mathbb{E}_S \left[ \mathbb{E}_{x \sim \mu} \left[ \frac{T^\kappa}{T} \sum_{s=1}^{T} x^\top \mathbb{E}_\pi[A_{s-1}^{-1}] x \right] \right] \qquad \text{(plugging in upper bound for } \mathbb{E}_\pi[\mathbb{1}\{x \in C_\pi\}]\text{)}$$

$$\leq \frac{K_{\text{TOTAL}} \cdot T^\kappa}{K \cdot T} \mathbb{E}_S \left[ \mathbb{E}_{x \sim \mu} \left[ \sum_{s=1}^{T} x^\top \mathbb{E}_\pi[A_{s-1}^{-1}] x \right] \right]$$

$$\leq \frac{K_{\text{TOTAL}} \cdot T^\kappa}{K} \mathbb{E}_S \left[ \mathbb{E}_{x \sim \mu} \left[ x^\top \mathbb{E}_\pi \left[ \frac{1}{T} \sum_{s=1}^{T} A_{s-1}^{-1} \right] x \right] \right]$$

$$\leq \frac{K_{\text{TOTAL}} \cdot T^\kappa}{K} \mathbb{E}_S[\mathbb{E}_{x \sim \mu}[x^\top \overline{M} x]],$$

where $\overline{M} = \mathbb{E}_\pi[\frac{1}{T} \sum_{s=1}^{T} A_{s-1}^{-1}]$ for a given sample $S$. $\qquad \square$

**Lemma D.1.** *Let the deletion distribution $\mu$ be the uniform distribution. In this case, the bound in Theorem 4.4 implies that we can process a total of $K_{\text{TOTAL}} = \frac{c \cdot K \cdot T}{dT^\kappa \log T}$ deletions while ensuring that the probability of exhausting the core set deletion capacity $K$ is at most $c$.*

**Proof.** First, we consider

$$\mathbb{E}_S[\mathbb{E}_{x \sim \text{unif}}[x^\top \overline{M} x]] = \mathbb{E}_S \left[ \frac{1}{T} \sum_{t=1}^{T} x_t^\top \overline{M} x_t \right]$$

$$\leq \frac{d \log T}{T}. \qquad (\sum_{t=1}^{T} x_t A_{t-1}^{-1} x_t \leq d \log T)$$

Plugging this into Theorem 4.4 and solving for $K_{\text{TOTAL}}$ completes the proof of the lemma. $\qquad \square$

## D.2. Auxiliary Results

**Theorem D.2.** *Let $w_T$ be the final predictor after running the* BBQSAMPLER *from Algorithm 1 with $\lambda = K$. Let $U$ be a sequence of deletions of length $K$. Let $w_{T \setminus U}$ be the predictor after the sequence of $U$ deletions have been applied. Let $x$ be an unqueried point. Then we have*

$$\Delta = w_{T \setminus U}^\top x - w_T^\top x - \leq 2\sqrt{e(K+1)} \cdot T^{-\kappa/2} \cdot \sqrt{d \log T \cdot \log(1/\delta)}$$

$$= O\left(\sqrt{K} \cdot T^{-\kappa/2} \cdot \sqrt{d \log T \cdot \log(1/\delta)}\right),$$

*with probability at least $1 - \delta$.*

**Proof.** Let $U_i$ be the set of the first $i$ deletions. Then we have

$$
\begin{aligned}
\Delta &= w_{T \setminus U}^\top x - w_T^\top x \\
&= \sum_{i=1}^{K} (w_{T \setminus U_i}^\top x - w_{T \setminus U_{i-1}}^\top x) \\
&= \sum_{i=1}^{K} \frac{2\sqrt{e(K+1)}}{K} \cdot T^{-\kappa/2} \cdot \sqrt{d \log T \cdot \log(1/\delta)} \qquad \text{(applying Theorem D.3)} \\
&\leq \frac{2K\sqrt{e(K+1)}}{K} \cdot T^{-\kappa/2} \cdot \sqrt{d \log T \cdot \log(1/\delta)} \\
&\leq 2\sqrt{e(K+1)} \cdot T^{-\kappa/2} \cdot \sqrt{d \log T \cdot \log(1/\delta)}.
\end{aligned}
$$

$\square$

**Theorem D.3.** *Let $\lambda = K$ be the regularization parameter. Consider a set $U$ of deletions where $|U| < K$. Let $w_{T \setminus U}$ be the predictor after the set of $U$ deletions have been applied and let $w_{T \setminus (U \cup x_i)}$ be the predictor after the set of $U$ deletions have been applied along with an additional deletion of $x_i$. Let $x$ be an unqueried point. Then we have*

$$\Delta = w_{T \setminus (U \cup x_i)}^\top x - w_{T \setminus U}^\top x \leq \frac{2\sqrt{e(K+1)}}{K} \cdot T^{-\kappa/2} \cdot \sqrt{d \log T \cdot \log(1/\delta)},$$

*for $\lambda = K$, with probability at least $1 - \delta$.*

**Proof.** Let $A_{T \setminus U} = A_T - \sum_{j \in U} x_j x_j^\top$ and $b_{T \setminus U} = b_T - \sum_{j \in U} y_j x_j$. Then we have

$$
\begin{aligned}
\Delta &= w_{T \setminus (D \cup x_i)}^\top x - w_{T \setminus U}^\top x \\
&= (b_{T \setminus U} - y_i x_i)^\top (A_{T \setminus U} - x_i x_i^\top)^{-1} x - b_{T \setminus U}^\top A_{T \setminus U}^{-1} x \\
&= b_{T \setminus U}^\top (A_{T \setminus U} - x_i x_i^\top)^{-1} x - y_i x_i^\top (A_{T \setminus U} - x_i x_i^\top)^{-1} x - b_{T \setminus U}^\top A_{T \setminus U}^{-1} x \\
&= b_{T \setminus U}^\top A_{T \setminus U}^{-1} x + \left(\frac{b_{T \setminus U}^\top A_{T \setminus U}^{-1} x_i x_i^\top A_{T \setminus U}^{-1} x}{1 - x_i^\top A_{T \setminus U}^{-1} x_i}\right) - y_i x_i^\top A_{T \setminus U}^{-1} x - y_i \left(\frac{x_i^\top A_{T \setminus U}^{-1} x_i x_i^\top A_{T \setminus U}^{-1} x}{1 - x_i^\top A_{T \setminus U}^{-1} x_i}\right) - b_{T \setminus U}^\top A_{T \setminus U}^{-1} x \\
&\qquad\qquad\qquad\qquad\qquad\qquad\qquad\qquad\qquad\qquad\qquad\qquad\qquad\qquad\qquad\qquad\qquad \text{(Sherman-Morrison)} \\
&= \left(\frac{b_{T \setminus U}^\top A_{T \setminus U}^{-1} x_i x_i^\top A_{T \setminus U}^{-1} x}{1 - x_i^\top A_{T \setminus U}^{-1} x_i}\right) - y_i x_i^\top A_{T \setminus U}^{-1} x - y_i \left(\frac{x_i^\top A_{T \setminus U}^{-1} x_i x_i^\top A_{T \setminus U}^{-1} x}{1 - x_i^\top A_{T \setminus U}^{-1} x_i}\right) \\
&= \left(\frac{w_{T \setminus U}^\top x_i x_i^\top A_{T \setminus U}^{-1} x}{1 - x_i^\top A_{T \setminus U}^{-1} x_i}\right) - y_i x_i^\top A_{T \setminus U}^{-1} x - y_i \left(\frac{x_i^\top A_{T \setminus U}^{-1} x_i x_i^\top A_{T \setminus U}^{-1} x}{1 - x_i^\top A_{T \setminus U}^{-1} x_i}\right) \\
&= \left(\frac{w_{T \setminus U}^\top x_i x_i^\top A_{T \setminus U}^{-1} x}{1 - \frac{1}{\lambda+1}}\right) - y_i x_i^\top A_{T \setminus U}^{-1} x - y_i \left(\frac{x_i^\top A_{T \setminus U}^{-1} x}{(\lambda+1)(1 - \frac{1}{\lambda+1})}\right) \qquad (x_i^\top A_{T \setminus U}^{-1} x_i \leq \frac{1}{\lambda+1} \text{ from Lemma D.6}) \\
&= \frac{1}{1 - \frac{1}{\lambda+1}} \cdot w_{T \setminus U}^\top x_i \cdot x_i^\top A_{T \setminus U}^{-1} x - y_i x_i^\top A_{T \setminus U}^{-1} x - \frac{1}{\lambda} y_i x_i^\top A_{T \setminus U}^{-1} x \\
&= \frac{1}{1 - \frac{1}{\lambda+1}} \cdot w_{T \setminus U}^\top x_i \cdot x_i^\top A_{T \setminus U}^{-1} x - \left(1 + \frac{1}{\lambda}\right) y_i x_i^\top A_{T \setminus U}^{-1} x \\
&= \left(1 + \frac{1}{\lambda}\right) x_i^\top A_{T \setminus U}^{-1} x \cdot (w_{T \setminus U}^\top x_i - y_i) \\
&\leq \left(1 + \frac{1}{\lambda}\right) (w_{T \setminus U}^\top x_i - y_i) \cdot \sqrt{x_i^\top A_{T \setminus U}^{-1} x_i \cdot x^\top A_{T \setminus U}^{-1} x} \qquad \text{(applying Lemma D.7)} \\
&\leq \left(\frac{\lambda+1}{\lambda}\right) (w_{T \setminus U}^\top x_i - y_i) \cdot \sqrt{\frac{e}{\lambda+1} \cdot T^{-\kappa}} \qquad \text{(applying Lemma D.6 and Corollary D.5)}
\end{aligned}
$$

$$= \frac{\sqrt{e(\lambda+1)}}{\lambda} \cdot T^{-\kappa/2}(w_{T \smallsetminus U}^\top x_i - y_i)$$

$$= \frac{\sqrt{e(\lambda+1)}}{\lambda} \cdot T^{-\kappa/2}(w_{T \smallsetminus U}^\top x_i - u^\top x_i + \zeta_i)$$

$$= \frac{\sqrt{e(\lambda+1)}}{\lambda} \cdot T^{-\kappa}(w_{T \smallsetminus U}^\top x_i - u^\top x_i) + \frac{\sqrt{e(\lambda+1)}}{\lambda} \cdot \zeta_i \cdot T^{-\kappa/2}$$

$$\leq \frac{\sqrt{e(\lambda+1)}}{\lambda} \cdot T^{-\kappa/2}\|w_{T \smallsetminus U} - u\|_{A_{T \smallsetminus U}}\|x_i\|_{A_{T \smallsetminus U}^{-1}} + \frac{\sqrt{e(\lambda+1)}}{\lambda} \cdot \zeta_i \cdot T^{-\kappa/2}$$

$$\leq \frac{\sqrt{e(\lambda+1)}}{\lambda} \cdot T^{-\kappa/2}\|w_{T \smallsetminus U} - u\|_{A_{T \smallsetminus U}}\|x_i\|_{A_{T \smallsetminus U}^{-1}} + \frac{\sqrt{e(\lambda+1)}}{\lambda} \cdot T^{-\kappa/2} \qquad (\zeta_i < 1)$$

$$\leq \frac{\sqrt{e(\lambda+1)}}{\lambda} \cdot T^{-\kappa/2} \cdot \sqrt{d\log(T-K) \cdot \log(1/\delta)} + \frac{\sqrt{e(\lambda+1)}}{\lambda} \cdot T^{-\kappa/2} \qquad \text{(Proposition G.1)}$$

$$\leq \frac{2\sqrt{e(\lambda+1)}}{\lambda} \cdot T^{-\kappa/2} \cdot \sqrt{d\log T \cdot \log(1/\delta)}.$$

$\square$

**Lemma D.4.** *Let $\lambda$ be the regularization parameter. Let $U$ be a set of deletions such that $|U| = K$. Let $A_{T \smallsetminus U}$ denote $A_T - \sum_{x_j \in U} x_j x_j^\top$. Then we have*

$$x^\top A_{T \smallsetminus U}^{-1} x \leq \sum_{i=0}^{K} \frac{\binom{K}{i}}{\lambda^i} \cdot T^{-\kappa}.$$

**Proof.** We prove the claim using induction. First, assume that $x^\top A_{T \smallsetminus U}^{-1} x \leq \sum_{i=0}^{K} \frac{\binom{K}{i}}{\lambda^i} \cdot T^{-\kappa}$ (induction hypothesis). Consider an additional deletion and the effect on the query condition, $x^\top (A_{T \smallsetminus U} - x_i x_i)^{-1} x$. We have

$$x^\top (A_{T \smallsetminus U} - x_i x_i^\top)^{-1} x = x^\top A_{T \smallsetminus U}^{-1} x + \left( \frac{x^\top A_{T \smallsetminus U}^{-1} x_i x_i^\top A_{T \smallsetminus U}^{-1} x}{1 - x_i^\top A_{T \smallsetminus U}^{-1} x_i} \right)$$

$$\leq x^\top A_{T \smallsetminus U}^{-1} x + \left( \frac{(x_i^\top A_{T \smallsetminus U}^{-1} x)^2}{(1 - \frac{1}{\lambda+1})} \right) \qquad (x_i^\top A_{T \smallsetminus (D \smallsetminus x_i)}^{-1} x_j \leq \frac{1}{\lambda+1} \text{ from Lemma D.6})$$

$$\leq \sum_{i=0}^{K} \frac{\binom{K}{i}}{\lambda^i} \cdot T^{-\kappa} + \left( \frac{(x_i^\top A_{T \smallsetminus U}^{-1} x)^2}{(1 - \frac{1}{\lambda+1})} \right) \qquad (x^\top A_{T \smallsetminus U}^{-1} x \leq \sum_{i=0}^{K} \frac{\binom{K}{i}}{\lambda^i} \cdot T^{-\kappa} \text{ from induction hypothesis})$$

$$\leq \sum_{i=0}^{K} \frac{\binom{K}{i}}{\lambda^i} \cdot T^{-\kappa} + \left( \frac{x_i^\top A_{T \smallsetminus U}^{-1} x_i \cdot x_i^\top A_{T \smallsetminus U}^{-1} x}{(1 - \frac{1}{\lambda+1})} \right) \qquad \text{(Lemma D.7)}$$

$$\leq \sum_{i=0}^{K} \frac{\binom{K}{i}}{\lambda^i} \cdot T^{-\kappa} + \left( \frac{\sum_{i=0}^{K} \frac{\binom{K}{i}}{\lambda^i} \cdot T^{-\kappa}}{(\lambda+1)(1 - \frac{1}{\lambda+1})} \right) \qquad \text{(using induction hypothesis and Lemma D.6)}$$

$$\leq \sum_{i=0}^{K} \frac{\binom{K}{i}}{\lambda^i} \cdot T^{-\kappa} + \sum_{i=0}^{K} \frac{\binom{K}{i}}{\lambda^{i+1}} \cdot T^{-\kappa}$$

$$\leq \sum_{i=0}^{K} \frac{\binom{K}{i}}{\lambda^i} \cdot T^{-\kappa} + \sum_{i=1}^{K+1} \frac{\binom{K}{i-1}}{\lambda^i} \cdot T^{-\kappa}$$

$$\leq \frac{\binom{K}{0}}{\lambda^0} \cdot T^{-\kappa} + \sum_{i=1}^{K} \frac{\binom{K}{i}}{\lambda^i} \cdot T^{-\kappa} + \sum_{i=1}^{K} \frac{\binom{K}{i-1}}{\lambda^i} \cdot T^{-\kappa} + \frac{\binom{K}{K}}{\lambda^{K+1}} \cdot T^{-\kappa}$$

$$\leq \frac{\binom{K}{0}}{\lambda^0} \cdot T^{-\kappa} + \sum_{i=1}^{K} \frac{\binom{K+1}{i}}{\lambda^i} \cdot T^{-\kappa} + \frac{\binom{K}{K}}{\lambda^{K+1}} \cdot T^{-\kappa} \qquad \text{(Pascal's Identity)}$$

$$\leq \frac{\binom{K}{0}}{\lambda^0} \cdot T^{-\kappa} + \sum_{i=1}^{K} \frac{\binom{K+1}{i}}{\lambda^i} \cdot T^{-\kappa} + \frac{\binom{K+1}{K+1}}{\lambda^{K+1}} \cdot T^{-\kappa}$$

$$\leq \sum_{i=0}^{K+1} \frac{\binom{K+1}{i}}{\lambda^i} \cdot T^{-\kappa}.$$

$\square$

**Corollary D.5.** *Let* $\lambda = K$ *be the regularization parameter. Let* $U$ *be a set of deletions such that* $|U| < K$, *then* $x^\top A_{T \setminus U}^{-1} x \leq e \cdot T^{-\kappa}$

**Proof.** From Lemma D.4, we have

$$x^\top A_{T \setminus U}^{-1} x \leq \sum_{i=0}^{|U|} \frac{\binom{|U|}{i}}{\lambda^i} \cdot T^{-\kappa}$$

$$\leq \sum_{i=0}^{K} \frac{\binom{K}{i}}{\lambda^i} \cdot T^{-\kappa}$$

$$\leq \sum_{i=0}^{K} \frac{\binom{K}{i}}{K^i} \cdot T^{-\kappa}$$

$$= \left(1 + \frac{1}{K}\right)^K \cdot T^{-\kappa}$$

$$\leq e \cdot T^{-\kappa}.$$

$\square$

**Lemma D.6.** $x_i^\top A_S^{-1} x_i \leq \frac{1}{\lambda+1}$, *for any set of* $S$ *points such that* $x_i \in S$, *where* $A_S = I + \sum_{x_t \in S} x_t x_t^\top$.

**Proof.** We want to consider the $x_i$ that maximizes $x_i^\top A_S^{-1} x_i$. Let $A_{S \setminus i} = I + \sum_{x_t \in S \setminus \{x_i\}} x_t x_t^\top$. Then we want to maximize the following,

$$x_i^\top A_S^{-1} x_i = x_i^\top (A_{S \setminus i} + x_i x_i^\top)^{-1} x_i$$

$$= x_i^\top A_{S \setminus i}^{-1} x_i - \frac{x_i^\top A_{S \setminus i}^{-1} x_i x_i^\top A_{S \setminus i}^{-1} x_i}{1 + x_i^\top A_{S \setminus i}^{-1} x_i}.$$

Let $a = x_i^\top A_{S \setminus i}^{-1} x_i$. Then we have

$$x_i^\top A_S^{-1} x_i = a - \frac{a^2}{1+a} = \frac{a}{1+a} = 1 - \frac{1}{1+a}.$$

We want to maximize the above expression where $0 \leq a \leq \frac{1}{\lambda}$ (since $0 \leq x_i^\top A_{S \setminus i}^{-1} x_i \leq \frac{1}{\lambda}$). The expression is maximized when $a = \frac{1}{\lambda}$. Thus, $x_i^\top A_{T-1}^{-1} x_i \leq \frac{1}{\lambda(1+\frac{1}{\lambda})} = \frac{1}{\lambda+1}$. $\square$

**Lemma D.7.** $(x_i^\top A_S^{-1} x)^2 \leq x_i^\top A_S^{-1} x_i \cdot x^\top A_S^{-1} x$, *for any set of* $S$ *points such that* $x_i \in S$, *where* $A_S = \lambda I + \sum_{x_t \in S} x_t x_t^\top$.

**Proof.** We can decompose the terms as $A_S^{-1} = \sum_{i=1}^{d} \lambda_i u_i u_i^\top$, $x_i = \sum_{i=1}^{d} \alpha_i u_i$, and $x = \sum_{i=1}^{d} \beta_i u_i$. Using these decompositions, we compute the following two terms,

$$(x_i^\top A_T^{-1} x)^2 = \left(\left(\sum_{i=1}^{d} \alpha_i u_i^\top\right)\left(\sum_{i=1}^{d} \lambda_i u_i u_i^\top\right)\left(\sum_{i=1}^{d} \beta_i u_i\right)\right)^2$$

$$= \left(\sum_{i=1}^{d} \lambda_i \alpha_i \beta_i\right)^2,$$

and

$$x_i^\top A_T^{-1} x_i \cdot x^\top A_T^{-1} x = \left( \sum_{i=1}^d \alpha_i u_i^\top \right) \left( \sum_{i=1}^d \lambda_i u_i u_i^\top \right) \left( \sum_{i=1}^d \alpha_i u_i \right) \left( \sum_{i=1}^d \beta_i u_i^\top \right) \left( \sum_{i=1}^d \lambda_i u_i u_i^\top \right) \left( \sum_{i=1}^d \beta_i u_i \right)$$

$$= \left( \sum_{i=1}^d \lambda_i \alpha_i^2 \right) \left( \sum_{i=1}^d \lambda_i \beta_i^2 \right).$$

From Jensen's inequality, we know that $(\sum_{i=1}^d p_i x_i)^2 \le \sum_{i=1}^d p_i x_i^2$ where $p_i > 0$ for all $i$ and $\sum_{i=1}^d p_i = 1$ since $f(x) = x^2$ is convex. Let $p_i = \lambda_i \alpha_i^2 / (\sum_{j=1}^d \lambda_j \alpha_j^2)$ (note that all $\lambda_i$'s $> 0$) and let $x_i = \beta_i / \alpha_i$. This gives us

$$\frac{\left( \sum_{i=1}^d \lambda_i \alpha_i \beta_i \right)^2}{\left( \sum_{i=1}^d \lambda_i \alpha_i^2 \right)^2} \le \frac{\sum_{i=1}^d \lambda_i \alpha_i^2 \cdot \frac{\beta_i^2}{\alpha_i^2}}{\sum_{i=1}^d \lambda_i \alpha_i^2}$$

$$\left( \sum_{i=1}^d \lambda_i \alpha_i \beta_i \right)^2 \le \left( \sum_{i=1}^d \lambda_i \alpha_i^2 \right) \left( \sum_{i=1}^d \lambda_i \beta_i^2 \right)$$

This directly implies that $(x_i^\top A_T^{-1} x)^2 \le x^\top A_T^{-1} x \cdot x_i^\top A_T^{-1} x_i$. $\qquad\square$

**Theorem D.8.** *Let $\lambda = K \le T$ be the regularization parameter and $0 < \kappa < 1$ be the sampling parameter of the* BBQSAMPLER. *Then we have the following regret and query complexity bounds on the* BBQSAMPLER,

$$R_T = \min_\varepsilon \varepsilon T_\varepsilon + O\left( \frac{1}{\varepsilon} \left( K + d \log T + \log \frac{T}{\delta} \right) + \frac{1}{\varepsilon^{2/\kappa}} \right),$$
$$N_T = O(dT^\kappa \log T).$$

**Proof.** Adapted from the analysis in Dekel et al. (2012) and Cesa-Bianchi et al. (2009).

Let $\Delta_t = u^\top x_t$ and $\hat{\Delta}_t = w_t^\top x$. We decompose the regret as follows,

$$R_T \le \varepsilon T_\varepsilon + \sum_{t=1}^T \bar{Z}_t \mathbb{1}\{\Delta_t \hat{\Delta}_t < 0, \Delta_t^2 > \varepsilon^2\} + \sum_{t=1}^T Z_t \mathbf{1}\{\Delta_t \hat{\Delta}_t < 0, \Delta_t^2 > \varepsilon^2\}|\Delta_t|$$

$$= \varepsilon T_\varepsilon + U_\varepsilon + Q_\varepsilon. \qquad \text{(regret decomposition from Dekel et al. (2012) Lemma 3)}$$

We define an additional term

$$\hat{\Delta}_t' = \begin{cases} \text{sign}(\hat{\Delta}_t) & \text{if } |\hat{\Delta}_t| > 1 \\ \hat{\Delta}_t & \text{otherwise} \end{cases}.$$

Then we have

$$Q_\varepsilon \le \frac{1}{\varepsilon} \sum_{t=1}^T Z_t \mathbb{1}\{\hat{\Delta}_t \Delta_t < 0\} \Delta_t^2$$

$$= \frac{1}{\varepsilon} \sum_{t=1}^T Z_t \mathbb{1}\{\hat{\Delta}_t' \Delta_t < 0\} \Delta_t^2 \qquad\qquad (\hat{\Delta}_t \text{ and } \hat{\Delta}_t' \text{ have the same sign})$$

$$\le \frac{1}{\varepsilon} \sum_{t=1}^T Z_t (\Delta_t - \hat{\Delta}_t)^2 \qquad\qquad (\hat{\Delta}_t' \Delta_t < 0 \text{ implies } \Delta_t^2 \le (\Delta_t - \hat{\Delta}_t')^2)$$

$$\le \frac{2}{\varepsilon} \left( \sum_{t=1}^T Z_t ((\Delta_t - y)^2 - (\hat{\Delta}_t - y)^2) + 144 \log \frac{T}{\delta} \right) \qquad\qquad \text{(Dekel et al. (2012) Lemma 23 (i))}$$

$$\le \frac{4}{\varepsilon} \left( \sum_{t=1}^T Z_t \left( d_{t-1}(w^*, w_{t-1}) - d_t(w^*, w_t) + 2 \log \frac{|A_t|}{|A_{t-1}|} \right) + 144 \log \frac{T}{\delta} \right)$$

(Dekel et al. (2012) Lemma 25 (iv) where $d_t(w^*, w) = \frac{1}{2}(w^* - w)^\top A_t (w^* - w)$)

$$\leq \frac{4}{\varepsilon}\left(d_0(w^*, w_0) + \log|A_T| + 144\log\frac{T}{\delta}\right)$$

$$\leq \frac{2}{\varepsilon}\left(\lambda + d\log(\lambda + N_T) + 144\log\frac{T}{\delta}\right) \qquad \text{(Dekel et al. (2012) Lemma 24 (iii))}$$

$$= O\left(\frac{1}{\varepsilon}\left(\lambda + d\log T + \log\frac{T}{\delta}\right)\right).$$

Let $r_t = x_t^\top A_t^{-1} x_t$. Then we have

$$U_\varepsilon \leq \sum_{t=1}^{T} \bar{Z}_t \, \mathbb{1}\{|\hat{\Delta}_t - \Delta_t| > \varepsilon\}$$

$$\leq (2 + e)\sum_{t=1}^{T} \bar{Z}_t \exp\left(-\frac{\varepsilon^2}{8r_t}\right) \qquad \text{(following Cesa-Bianchi et al. (2009) Theorem 1)}$$

$$= (2 + e)\sum_{t=1}^{T} \bar{Z}_t \exp\left(-\frac{\varepsilon^2 T^\kappa}{8}\right) \qquad \text{(when } \bar{Z}_t = 1, r_t < T^{-\kappa} \text{ by the query condition)}$$

$$\leq (2 + e)\sum_{t=1}^{T} \bar{Z}_t \exp\left(-\frac{\varepsilon^2 t^\kappa}{8}\right)$$

$$\leq (2 + e)\lceil 1/\kappa\rceil!\left(\frac{8}{\varepsilon^2}\right)^{1/\kappa} \qquad \text{(following Cesa-Bianchi et al. (2009) Theorem 1)}$$

$$\leq O\left(\frac{1}{\varepsilon^{2/\kappa}}\right).$$

Putting the above terms together completes the proof of regret.

Now for the number of queries. Let $r_t = x_t^\top A_t^{-1} x_t$. Consider the following sum,

$$\sum_{t=1}^{T} Z_t r_t \leq \sum_{t=1}^{T} Z_t \cdot \log\frac{|A_t|}{|A_{t-1}|} \qquad \text{(Lemma 24 from Dekel et al. (2012) where } |\cdot| \text{ is the determinant)}$$

$$= \log\frac{|A_T|}{|A_0|}$$

$$\leq \log|A_T|$$

$$\leq d\log(\lambda + N_T)$$

$$\leq d\log(T).$$

We use the above sum to bound the number of queries,

$$N_T = \sum_{r_t > T^{-\kappa}} 1$$

$$\leq \sum_{r_t > T^{-\kappa}} \frac{r_t}{T^{-\kappa}}$$

$$\leq T^\kappa \sum_{r_t > T^{-\kappa}} r_t$$

$$\leq O(dT^\kappa \log(T)). \qquad \text{(using the sum above)}$$

$\square$

## E. Missing Details of GENERALBBQSAMPLER

Define

$$D^2(x; \langle x_1, \dots, x_{t-1}\rangle) = \sup_{f,g \in \mathcal{F}} \frac{(f(x) - g(x))^2}{\sum_{i=1}^{t}(f(x_i) - g(x_i))^2 + 1}.$$

---

**Algorithm 3** GENERALBBQSAMPLER (Slightly Modified Version of Algorithm 2 from Gentile et al. (2022))

---

**Require:** • Dataset $S$ of size $T$
      • Confidence level $\delta \in (0, 1]$

1: **Initialize:** $\mathcal{P}_0 = S$ and $\mathcal{R}_0 = S$
2: **for** $\ell = 1, 2, \dots$ **do**
3:     Initialize within stage $\ell$: $\varepsilon_\ell = 2^{-\ell}/\sqrt{\mathfrak{R}(T, \delta)}$, $t = 0$, $\mathcal{Q}_\ell = \varnothing$
4:
5:     **while** $\mathcal{P}_{\ell-1} \setminus \mathcal{Q}_\ell \neq \varnothing$ and $\max_{x \in \mathcal{P}_{\ell-1} \setminus \mathcal{Q}_\ell} D(x, \mathcal{Q}_\ell) > \varepsilon_\ell$ **do**
6:         $t = t + 1$
7:         Pick $x_{\ell,t} \in \mathrm{argmax}_{x \in \mathcal{P}_{\ell-1} \setminus \mathcal{Q}_\ell} D(x, \mathcal{Q}_\ell)$
8:         Update $\mathcal{Q}_\ell = \mathcal{Q}_\ell \cup \{x_{\ell,t}\}$
9:
10:     Set $T_\ell = t$, the number of queries made in stage $\ell$
11:     **if** $\mathcal{Q}_\ell \neq \varnothing$ **then**
12:         Query the labels $y_{\ell,1}, \dots, y_{\ell,T_\ell}$ associated with the unlabeled data in $\mathcal{Q}_\ell$ and compute

$$\hat{f}_{\mathcal{Q}_\ell} = \mathrm{argmin}_{f \in \mathcal{F}} \sum_{t=1}^{T_\ell} \left(\frac{1 + y_{\ell,t}}{2} - f(x_{\ell,t})\right)^2$$

13:         Set $\mathcal{C}_\ell = \{x \in \mathcal{P}_{\ell-1} \setminus \mathcal{Q}_\ell : |\hat{f}_{\mathcal{Q}_\ell}(x) - 1/2| > 3 \cdot 2^{-\ell}\}$
14:     **else** $\hat{f}_\ell = 1/2, \mathcal{C}_\ell = \varnothing$
15:
16:     Set $\mathcal{P}_\ell = \mathcal{P}_{\ell-1} \setminus \mathcal{Q}_\ell$ and $\mathcal{R}_\ell = \mathcal{R}_{\ell-1} \setminus (\mathcal{C}_\ell \cup \mathcal{Q}_\ell)$
17:     **if** $\mathfrak{D}(\mathcal{F}, \mathcal{P}) \cdot \mathfrak{R}(T, \delta)/2^{-\ell+1} > 2^{-\ell+1}|\mathcal{R}_\ell|$ **then**
18:         Set $L = \ell$
19:         **exit for loop**
20:
21: Set $\mathcal{Q} = \bigcup_{\ell=1}^{L} \mathcal{Q}_\ell$
22: Compute

$$\hat{f}_{\mathcal{Q}} = \mathrm{argmin}_{f \in \mathcal{F}} \sum_{(x_i, y_i) \in \mathcal{Q}} \left(\frac{1 + y_i}{2} - f(x_i)\right)^2$$

    **return** $\hat{f}_{\mathcal{Q}}, \mathcal{Q}$

---

## F. Proofs from Section 5

### F.1. Notation

- $[n] = \{1, 2, \dots, n\}$

- $\hat{f}_S$ - ERM computed over set $S$

- $\hat{f}_{S \setminus i}$ - ERM computed over set $S \setminus \{x_i\}$

- $\mathcal{Q}_\ell(t)$ - the set of points queried in stage $\ell$ up to time $t$

- $\varepsilon_\ell = 2^{-\ell}/\sqrt{\mathfrak{R}(T, \delta)}$

- $D^2(x; \langle x_1, \ldots, x_{t-1} \rangle) = \sup_{f,g \in \mathcal{F}} \frac{(f(x) - g(x))^2}{\sum_{i=1}^{t}(f(x_i) - g(x_i))^2 + 1}$

**Theorem 5.1.** *Given the set $S$ and deletion requests $U$, let $\mathfrak{C}(S)$ denote the subset of points for which the labels were queried by* GENERALBBQSAMPLER *in Algorithm 2, and let $A(S, U)$ denote the unlearned model. Then, Algorithm 2 is an exact system-aware unlearning algorithm with $S' = \mathfrak{C}(S)$ and state-of-system $\mathsf{I}_A(S, U) = (A(S, U), \mathfrak{C}(S) \smallsetminus U)$.*

**Proof.** Fix any sample $S$ and set of deletions $U$. Define $S' = \mathfrak{C}(S) = \mathcal{Q}$. Clearly, $S' \subseteq S$. The core set of the GENERALB-BQSAMPLER is exactly the set of points that it queries. Thus, applying Theorem 5.2, we know $\mathfrak{C}(\mathfrak{C}(S) \smallsetminus U) = \mathfrak{C}(S) \smallsetminus U$. $A(S' \smallsetminus U, \varnothing)$ returns an ERM over $\mathfrak{C}(\mathfrak{C}(S) \smallsetminus U)$, which is exactly $\mathfrak{C}(S) \smallsetminus U$, and stores that ERM and the set $\mathfrak{C}(S) \smallsetminus U$. To process the deletion of $U$, $A(S, U)$ returns an ERM over $\mathfrak{C}(S) \smallsetminus U$ and stores that ERM and the set $\mathfrak{C}(S) \smallsetminus U$. Thus, $\mathsf{I}_A(S, U) = \mathsf{I}_A(S' \smallsetminus U, \varnothing)$ for all $U \subseteq S$. $\qquad\square$

**Theorem 5.2.** *For any dataset $S$, Algorithm 2 has the property that for all $U \subseteq S$, $\mathfrak{C}(\mathfrak{C}(S) \smallsetminus U) = \mathfrak{C}(S) \smallsetminus U$.*

**Proof.** Consider the deletion of any $x_j \in S$. Let $\mathcal{Q} = \bigcup_{\ell=1}^{L} \mathcal{Q}_\ell$ be the set of queried points after executing the GENERALBBQSAMPLER on $S$, and let $\mathcal{Q}' = \bigcup_{\ell=1}^{L} \mathcal{Q}'_\ell$ be the set of queried points after executing the GENERALBBQSAMPLER on $\mathcal{Q} \smallsetminus \{x_j\}$. Note that $\mathfrak{C}(S) = \mathcal{Q}$ and $\mathfrak{C}(\mathfrak{C}(S) \smallsetminus \{x_j\}) = \mathcal{Q}'$. We want to show that $\mathcal{Q}' = \mathcal{Q} \smallsetminus \{x_j\}$.

First, note that the deletion of any unqueried point has no effect on the query condition of any other point, so the set of queried points when executing the GENERALBBQSAMPLER on $\mathcal{Q}$ is exactly $\mathcal{Q}$. If $x_j \notin \mathcal{Q}$, then clearly $\mathcal{Q}' = \mathcal{Q} \smallsetminus \{x_j\}$. Thus, we focus on the case where $x_j \in \mathcal{Q}$.

Next, note that the query condition of points queried before $x_j$ only depends on points in $\mathcal{Q} \smallsetminus \{x_j\}$. Thus, all points queried before $x_j$ will still be queried.

Let $\ell^*$ be the stage and $t^*$ be the time at which $x_j$ was originally queried. Let $\mathcal{Q}_\ell(t')$ be the set of points queried in stage $\ell$ before time $t'$ in the original run of the GENERALBBQSAMPLER. For each point queried at time $t' > t^*$, we know $D^2(x, \mathcal{Q}_{\ell^*}(t')) > \varepsilon_{\ell^*}^2$. Now consider

$$
\begin{aligned}
D^2(x, \mathcal{Q}_{\ell^*}(t') \smallsetminus \{x_j\}) &= \sup_{f,g \in \mathcal{F}} \frac{(f(x) - g(x))^2}{\sum_{\mathcal{Q}_{\ell^*}(t') \smallsetminus \{x_j\}}(f(x_i) - g(x_i))^2 + 1} \\
&\geq \sup_{f,g \in \mathcal{F}} \frac{(f(x) - g(x))^2}{\sum_{\mathcal{Q}_{\ell^*}(t')}(f(x_i) - g(x_i))^2 + 1} \\
&> \varepsilon_\ell^2.
\end{aligned}
$$

Furthermore, observe that

$$
\operatorname*{argmax}_{x \in (\mathcal{Q}_{\ell^*-1} \smallsetminus \{x_j\}) \smallsetminus \mathcal{Q}_{\ell^*}(t')} D^2(x, \mathcal{Q}'_{\ell^*} \smallsetminus \{x_j\}) = \operatorname*{argmax}_{x \in \mathcal{P}_{\ell^*-1} \smallsetminus \mathcal{Q}_{\ell^*}(t')} D^2(x, \mathcal{Q}_{\ell^*}(t')),
$$

where the first term represents what would be queried at time $t'$ when executing on $\mathcal{Q} \smallsetminus \{x_j\}$ and the second term represents what would be queried at time $t'$ when executing on $\mathcal{Q}$. Thus, we will query the same set of points in both executions after time $t$ in stage $\ell$. Thus, at the end of stage $\ell$, when executing on dataset $S$ with $x_j$ deleted, we have queried every point in $\mathcal{Q}_\ell \smallsetminus \{x_j\}$.

At the end of stage $\ell$ in the execution on $\mathcal{Q}$, the set of points remaining in the pool is exactly $\bigcup_{\ell=\ell^*+1}^{L} \mathcal{Q}_\ell$. Each of the query conditions in future stages after stage $\ell^*$ only depends on the points in $\bigcup_{\ell=\ell^*+1}^{L} \mathcal{Q}_\ell$. These query conditions are unaffected, and all points in $\bigcup_{\ell=\ell^*+1}^{L} \mathcal{Q}_\ell$ will be queried.

Thus, the set of queried points after executing the GENERALBBQSAMPLER on $\mathcal{Q} \smallsetminus \{x_j\}$ is exactly $\mathcal{Q} \smallsetminus \{x_j\}$. We can apply the above argument inductively for each $x_j \in U$ to conclude that $\mathfrak{C}(\mathfrak{C}(S) \smallsetminus U) = \mathfrak{C}(S) \smallsetminus U$. $\qquad\square$

**Theorem 5.3.** *Assume that, with probability $1 - \delta/T$, the ERM $\hat{f}$ in Algorithm 2 satisfies the bound $\sum_{t=1}^{T}(f^*(x_t) - \hat{f}(x_t))^2 \leq \mathfrak{R}(T, \delta)$. Then, the memory required by Algorithm 2 is bounded by*

$$
N_T = O\left( \min_{\varepsilon} \left\{ T_\varepsilon + \frac{\mathfrak{R}(T, \delta) \cdot \mathfrak{D}(\mathcal{F}, S)}{\varepsilon^2} \right\} \right),
$$

**Proof.** For points in $\mathcal{R}_L = \mathcal{P} \smallsetminus (\cup_{\ell=1}^{L} \mathcal{Q}_\ell \cup \cup_{\ell=1}^{L} \mathcal{C}_\ell)$, we know that $|\hat{f}_{\mathcal{Q}_L}(x) - 1/2| \leq 3 \cdot 2^{-L}$ by the design of Algorithm 3. We also know that for all $x \in \mathcal{R}_L$, $|f^*(x) - \hat{f}_{\mathcal{Q}_\ell}(x)| \leq 2^{-L}$ from Lemma F.1. Thus, for all $x \in \mathcal{R}_L$, $|f^*(x) - 1/2| \leq 4 \cdot 2^{-L} = 2^{-L+2}$.

From the design of Algorithm 3, we also know that $|\mathcal{R}_L| = |\mathcal{P} \smallsetminus (\cup_{\ell=1}^{L} \mathcal{Q}_\ell \cup \cup_{\ell=1}^{L} \mathcal{C}_\ell)| \leq 4^{L-1} \cdot \mathfrak{D}(\mathcal{F}, S)$.

Let

$$T_\varepsilon = \sum_{t=1}^{T} \mathbb{1}\{|\text{sign}(f^*(x_t)) - 1/2| \leq \varepsilon\}$$

When $2^{-L+2} \leq \varepsilon$, we can upper bound $|\mathcal{R}_L|$ by $T_\varepsilon$. Thus, we have

$$4^{L-1} \cdot \mathfrak{D}(\mathcal{F}, S) \cdot \mathfrak{R}(T, \delta) \leq T_\varepsilon$$

Otherwise, when $2^{-L+2} \geq \varepsilon$, we know that

$$|\mathcal{R}_L| \leq 4^{L-1} \cdot \mathfrak{D}(\mathcal{F}, S) \cdot \mathfrak{R}(T, \delta) \qquad \text{(Theorem G.3)}$$

$$\leq \frac{4}{\varepsilon^2} \cdot \mathfrak{D}(\mathcal{F}, S) \cdot \mathfrak{R}(T, \delta) \qquad (2^L \leq \tfrac{4}{\varepsilon})$$

Thus, we have that

$$4^{L-1} \cdot \mathfrak{D}(\mathcal{F}, S) \cdot \mathfrak{R}(T, \delta) \leq \min_{\varepsilon} \left\{ T_\varepsilon + \frac{4}{\varepsilon^2} \cdot \mathfrak{R}(T, \delta) \cdot \mathfrak{D}(\mathcal{F}, S) \right\}.$$

From Theorem G.3, we know that

$$N_T \leq 4^{L+1} \cdot \mathfrak{R}(T, \delta) \cdot \mathfrak{D}(\mathcal{F}, S) \qquad \text{(Theorem G.3)}$$

$$\leq O\left( \min_{\varepsilon} \left\{ T_\varepsilon + \frac{64}{\varepsilon^2} \cdot \mathfrak{R}(T, \delta) \cdot \mathfrak{D}(\mathcal{F}, S) \right\} \right).$$

$\square$

**Theorem 5.5.** *If the regression oracle for $\mathcal{F}$ satisfies uniform stability under the squared loss with rate $\beta$, then with probability $1 - \delta$, the excess risk of the final predictor returned by Algorithm 2 satisfies*

$$\mathcal{E}(\hat{f}) = O\left( \frac{1}{T - N_T} \cdot (N_T \log T + \log(1/\delta)) \right)$$

*after unlearning up to*

$$K = O\left( \frac{\sqrt{\mathfrak{R}(T, \delta)}}{\sqrt{N_T} \cdot \beta(N_T)} \right)$$

*many core set deletions.*

**Proof.** From Lemma F.3, we know that the regression oracle satisfies the following, such that for all $S \in \mathcal{Z}^n$, for all $i \in [n]$, for all $\{x_1, \ldots, x_n\}$

$$\sum_{t=1}^{n} (\hat{f}_{S \smallsetminus i}(x_t) - \hat{f}_S(x_t))^2 \leq n \cdot \beta(n)^2.$$

Let $\mathcal{Q}$ be the set of queried points and let $N_T$ be the number of queried points. Let $D$ be a set of $K$ deletions. Let $D_i$ be the set of the first $i$ deletions. Then, we have

$$\sum_{\mathcal{Q}_\ell} (\hat{f}_{\mathcal{Q} \smallsetminus D_{i+1}}(x_t) - \hat{f}_{\mathcal{Q} \smallsetminus D_i}(x_t))^2 \leq N_T \cdot \beta(N_T)^2.$$

In stage $\ell$ of Algorithm 3, we know that for all $x \in \mathcal{C}_\ell$ (the set of unqueried points for which we are confident on the label), we have

$$\sup_{f,g \in \mathcal{F}} (f(x) - g(x))^2 \le \varepsilon_\ell^2 \left( \sum_{Q_\ell} (f(x_i) - g(x_i))^2 + 1 \right),$$

due to the query condition of stage $\ell$.

Thus,

$$
\begin{aligned}
\sum_{t=1}^{N_T} (\hat{f}_{\mathcal{Q} \smallsetminus D}(x_t) - \hat{f}_{\mathcal{Q}}(x_t))^2 &\le \sum_{t=1}^{N_T} \left( \sum_{i=1}^{K} (\hat{f}_{\mathcal{Q} \smallsetminus D_i}(x_t) - \hat{f}_{\mathcal{Q} \smallsetminus D_{i-1}}(x_t)) \right)^2 \\
&\le K^2 \sum_{t=1}^{N_T} \left( \frac{1}{K} \sum_{i=1}^{K} (\hat{f}_{\mathcal{Q} \smallsetminus D_i}(x_t) - \hat{f}_{\mathcal{Q} \smallsetminus D_{i-1}}(x_t)) \right)^2 \\
&\le K^2 \sum_{t=1}^{N_T} \frac{1}{K} \sum_{i=1}^{m} (\hat{f}_{\mathcal{Q} \smallsetminus D_i}(x_t) - \hat{f}_{\mathcal{Q} \smallsetminus D_{i-1}}(x_t))^2 && \text{(Jensen's Inequality)} \\
&= K \sum_{t=1}^{N_T} \sum_{i=1}^{K} (\hat{f}_{\mathcal{Q} \smallsetminus D_i}(x_t) - \hat{f}_{\mathcal{Q} \smallsetminus D_{i-1}}(x_t))^2 \\
&\le K \sum_{i=1}^{K} \sum_{t=1}^{N_T} (\hat{f}_{\smallsetminus D_i}(x_t) - \hat{f}_{\smallsetminus D_{i-1}}(x_t))^2 \\
&\le K \sum_{i=1}^{K} N_T \cdot \beta(N_T)^2 && \text{(applying Lemma F.3)} \\
&= K^2 \cdot N_T \cdot \beta(N_T)^2
\end{aligned}
$$

Plugging this in, we have

$$
\begin{aligned}
\sup_{f,g \in \mathcal{F}} (f(x) - g(x))^2 &\le \varepsilon_\ell^2 \left( \sum_{Q_\ell} (f(x_i) - g(x_i))^2 + 1 \right) \\
&\le \varepsilon_\ell^2 \cdot K^2 \cdot N_T \cdot \beta(N_T)^2
\end{aligned}
$$

for all $x \in \mathcal{C}_\ell$.

The above implies that for any $x \in \mathcal{C}_\ell$,

$$
(\hat{f}_{\mathcal{Q} \smallsetminus D}(x) - \hat{f}_{\mathcal{Q}}(x))^2 \le \varepsilon_\ell^2 \cdot K^2 \cdot N_T \cdot \beta(N_T)^2
$$
$$
|\hat{f}_{\smallsetminus D}(x_t) - \hat{f}(x_t))| \le \varepsilon_\ell \cdot K \cdot \sqrt{N_T} \cdot \beta(N_T)
$$

Furthermore, from Lemma F.2, we know that for all stages $\ell$, for all $x \in \mathcal{C}_\ell$,

$$
|\hat{f}_{\mathcal{Q}}(x) - 1/2| > 2^{-\ell}.
$$

and $\text{sign}(\hat{f}_{\mathcal{Q}}(x) - 1/2) = \text{sign}(f^*(x) - 1/2)$. Thus, $\hat{f}_{\mathcal{Q}}$ classifies all of the points in $\bigcup_{\ell=1}^{L} \mathcal{C}_\ell$ correctly with some margin. After deletion, $\hat{f}_{\mathcal{Q} \smallsetminus D}$ and $\hat{f}_{\mathcal{Q}}$ agree on the sign of $x$ when

$$
\begin{aligned}
\varepsilon_\ell \cdot K \cdot \sqrt{N_T} \cdot \beta(N_T) &\le 2^{-\ell} \\
2^{-\ell} \cdot \frac{1}{\sqrt{\mathfrak{R}(T, \delta)}} \cdot K \cdot \sqrt{N_T} \cdot \beta(N_T) &\le 2^{-\ell} && (\varepsilon_\ell = 2^{-\ell}/\sqrt{\mathfrak{R}(T,\delta)}) \\
\frac{1}{\sqrt{\mathfrak{R}(T, \delta)}} \cdot K \cdot \sqrt{N_T} \cdot \beta(N_T) &\le 1
\end{aligned}
$$

$$K \le \frac{\sqrt{\mathfrak{R}(T,\delta)}}{\sqrt{N_T} \cdot \beta(N_T)}.$$

Thus, for up to

$$K \le \frac{\sqrt{\mathfrak{R}(T,\delta)}}{\sqrt{N_T} \cdot \beta(N_T)}$$

deletions, $\text{sign}(\hat{f}_{\mathcal{Q} \smallsetminus D}(x) - 1/2) = \text{sign}(\hat{f}_{\mathcal{Q}}(x) - 1/2) = \text{sign}(f^*(x) - 1/2)$ on all of the points in $\cup_{\ell=1}^{L} \mathcal{C}_\ell$.

For points in $\mathcal{R}_L = \mathcal{P} \smallsetminus (\cup_{\ell=1}^{L} \mathcal{Q}_\ell \cup \cup_{\ell=1}^{L} \mathcal{C}_\ell)$, we know that $|\hat{f}_{\mathcal{Q}_L}(x) - 1/2| \le 3 \cdot 2^{-L}$ by the design of [Algorithm 3](). We also know that for all $x \in \mathcal{R}_L$, $|f^*(x) - \hat{f}_{\mathcal{Q}_\ell}(x)| \le 2^{-L}$ from [Lemma F.1](). Thus, for all $x \in \mathcal{R}_L$, $|f^*(x) - 1/2| \le 4 \cdot 2^{-L} = 2^{-L+2}$.

From the design of [Algorithm 3](), we also know that $|\mathcal{R}_L| = |\mathcal{P} \smallsetminus (\cup_{\ell=1}^{L} \mathcal{Q}_\ell \cup \cup_{\ell=1}^{L} \mathcal{C}_\ell)| \le 4^{L-1} \cdot \mathfrak{D}(\mathcal{F}, S) \cdot \mathfrak{R}(T,\delta)$.

$\mathcal{R}_L$ is the set of points which we did not query but we are unsure of the label. We know from [Theorem 5.3]() that

$$|\mathcal{R}_L|, N_T = O\left( \min_{\varepsilon} \left\{ T_\varepsilon + \frac{1}{\varepsilon^2} \cdot \mathfrak{R}(T,\delta) \cdot \mathfrak{D}(\mathcal{F}, S) \right\} \right).$$

As shown above, for all other unqueried points, $\hat{f}_{\mathcal{Q} \smallsetminus D}$ agrees with the classification of the Bayes optimal classifier, despite not using these points during training. In particular, we have that

$$\hat{L}_{S \smallsetminus \{\mathcal{Q} \smallsetminus D\}}(\hat{f}_{\mathcal{Q} \smallsetminus D}) = \sum_{S \smallsetminus \{\mathcal{Q} \smallsetminus D\}} \mathbb{1}\{\text{sign}(\hat{f}_{\mathcal{Q} \smallsetminus D}(x) - 1/2) \ne \text{sign}(f^*(x) - 1/2)\} \le N_T + K,$$

because $\hat{f}_{\mathcal{Q} \smallsetminus D}$ and $f^*(x)$ may disagree on the classification of the $K$ deleted queried points and the points in $N_T$ points in $\mathcal{R}_L$; they must agree on all other unqueried points.

Similarly to the linear case, we can use techniques from generalization for sample compression algorithms ([Kakade & Tewari, 2008]()) to convert the empirical classification loss to an excess risk bound for $\hat{f}_{\mathcal{Q} \smallsetminus D}$. First observe that

$$\begin{aligned}
\mathcal{E}(\hat{h}) &= \mathbb{E}_{(x,y) \sim \mathcal{D}}[\mathbb{1}\{\text{sign}(\hat{f}_{\mathcal{Q} \smallsetminus D}(x) - 1/2) \ne y\} - \mathbb{1}\{\text{sign}(f^*(x) - 1/2) \ne y\}] \\
&= \mathbb{E}_{(x,y) \sim \mathcal{D}}[|2|f^*(x) - 1/2| - 1| \cdot \mathbb{1}\{\text{sign}(\hat{f}_{\mathcal{Q} \smallsetminus D}(x) - 1/2) \ne \text{sign}(f^*(x) - 1/2)\}] \\
&\le \mathbb{E}_{(x,y) \sim \mathcal{D}}[\mathbb{1}\{\text{sign}(\hat{f}_{\mathcal{Q} \smallsetminus D}(x) - 1/2) \ne \text{sign}(f^*(x) - 1/2)\}]
\end{aligned}$$

We look to bound the loss of $L(\hat{f}_{\mathcal{Q} \smallsetminus D}) = \mathbb{E}_{(x,y) \sim \mathcal{D}}[\mathbb{1}\{\text{sign}(\hat{f}_{\mathcal{Q} \smallsetminus D}(x) - 1/2) \ne \text{sign}(f^*(x) - 1/2)\}]$. We are interested in the event that there exists a $\mathcal{Q} \smallsetminus D \subseteq S, |\mathcal{Q} \smallsetminus D| = l$ such that $\hat{L}_{S \smallsetminus \{\mathcal{Q} \smallsetminus D\}}(\hat{f}_{\mathcal{Q} \smallsetminus D}) \le N_T + K$ and $L(\hat{f}_{\mathcal{Q} \smallsetminus D}) \ge \varepsilon$

$$\begin{aligned}
&\Pr[\exists \, \mathcal{Q} \smallsetminus D \subseteq S \text{ such that } \hat{L}_{S \smallsetminus \{\mathcal{Q} \smallsetminus D\}}(\hat{f}_{\mathcal{Q} \smallsetminus D}) \le N_T + K \text{ and } L(\hat{f}_{\mathcal{Q} \smallsetminus D}) \ge \varepsilon] \\
&\le \sum_{l=1}^{T} \Pr[\exists \, \mathcal{Q} \smallsetminus D \subseteq S, |\mathcal{Q} \smallsetminus D| = l \text{ such that } \hat{L}_{S \smallsetminus \{\mathcal{Q} \smallsetminus D\}}(\hat{f}_{\mathcal{Q} \smallsetminus D}) \le N_T + K \text{ and } L(\hat{f}_{\mathcal{Q} \smallsetminus D}) \ge \varepsilon] \\
&\le \sum_{l=1}^{T} \sum_{\mathcal{Q} \smallsetminus D \subseteq S, |\mathcal{Q} \smallsetminus D| = l} \Pr[\hat{L}_{S \smallsetminus \{\mathcal{Q} \smallsetminus D\}}(\hat{f}_{\mathcal{Q} \smallsetminus D}) \le N_T + K \text{ and } L(\hat{f}_{\mathcal{Q} \smallsetminus D}) \ge \varepsilon] \\
&= \sum_{l=1}^{T} \sum_{\mathcal{Q} \smallsetminus D \subseteq S, |\mathcal{Q} \smallsetminus D| = l} \mathbb{E}\left[\Pr_{S \smallsetminus \{\mathcal{Q} \smallsetminus D\}}[\hat{L}_{S \smallsetminus \{\mathcal{Q} \smallsetminus D\}}(\hat{f}_{\mathcal{Q} \smallsetminus D}) \le N_T + K \text{ and } L(\hat{f}_{\mathcal{Q} \smallsetminus D}) \ge \varepsilon \mid \mathcal{Q} \smallsetminus D]\right]
\end{aligned}$$

Let $|\mathcal{Q}| = N_T$ and let $|D| = K$, where $N_T - K = l$. Now for any fixed $\mathcal{Q} \smallsetminus D$, the above probability is just the probability of having a true risk greater than $\varepsilon$ and an empirical risk at most $N_T + K$ on a test set of size $T - N_T + K$. Now for any random variable $z \in [0, 1]$, if $\mathbb{E}[z] \ge \varepsilon$ then $\Pr[z = 0] \le 1 - \varepsilon$. Thus, for a given $\mathcal{Q} \smallsetminus D$,

$$\Pr_{S \smallsetminus \{\mathcal{Q} \smallsetminus D\}}[\hat{L}_{S \smallsetminus \{\mathcal{Q} \smallsetminus D\}}(\hat{f}_{\mathcal{Q} \smallsetminus D}) \le N_T + K \text{ and } L(\hat{f}_{\mathcal{Q} \smallsetminus D}) \ge \varepsilon] \le (1 - \varepsilon)^{T - 2N_T}$$

Plugging this in above, we have

$$\Pr[\exists\, \mathcal{Q} \smallsetminus D \subseteq S,\ |\mathcal{Q} \smallsetminus D| = l \text{ such that } \hat{L}_{S \smallsetminus \{\mathcal{Q} \smallsetminus D\}}(\hat{f}_{\mathcal{Q} \smallsetminus D}) \le N_T + K \text{ and } L(\hat{f}_{\mathcal{Q} \smallsetminus D}) \ge \varepsilon]$$

$$\le \sum_{l=1}^{T} \sum_{\mathcal{Q} \smallsetminus D \subseteq S, |\mathcal{Q} \smallsetminus D| = l} (1 - \varepsilon)^{T - 2N_T}$$

$$\le \sum_{l=1}^{T} T^l \cdot (1 - \varepsilon)^{T - 2N_T}$$

$$= \sum_{l=1}^{T} T^{N_T - K} \cdot (1 - \varepsilon)^{T - 2N_T}$$

$$= \sum_{l=1}^{T} T^{N_T - K} \cdot (1 - \varepsilon)^{T - 2N_T}$$

$$\le \sum_{l=1}^{T} T^{N_T} \cdot e^{-\varepsilon(T - 2N_T)}$$

$$= T^{N_T + 1} \cdot e^{-\varepsilon(T - 2N_T)}$$

We want this probability to be at most $\delta$. Setting $\varepsilon$ appropriately, we have

$$\varepsilon = \frac{1}{T - 2N_T} \cdot ((N_T + 1) \log T + \log(1/\delta)).$$

Thus, with probability at least $1 - \delta$,

$$L(\hat{f}_{\mathcal{Q} \smallsetminus D}) \le \frac{1}{T - 2N_T} \cdot ((N_T + 1) \log T + \log(1/\delta)).$$

This implies that with probability at least $1 - \delta$,

$$\mathcal{E}(\hat{h}) \le \frac{1}{T - 2N_T} \cdot ((N_T + 1) \log T + \log(1/\delta)).$$

$\square$

## F.2. Auxiliary Results

**Lemma F.1.** *With probability at least $1 - \delta$, for all stages $\ell$, for all $x \in \mathcal{P}_{\ell-1} \smallsetminus \mathcal{Q}_\ell$,*

$$|f^*(x) - \hat{f}_{\mathcal{Q}}(x)| \le 2^{-\ell} \text{ and } |f^*(x) - \hat{f}_{\mathcal{Q}_\ell}(x)| \le 2^{-\ell}$$

**Proof.** The following proof is adapted from Gentile et al. (2022, Lemma 16).

For all $\ell$, for all $x \in \mathcal{P}_{\ell-1} \smallsetminus \mathcal{Q}_\ell$,

$$
\begin{aligned}
(f^*(x) &- \hat{f}_{\mathcal{Q}}(x))^2 \\
&= \frac{(f^*(x) - \hat{f}_{\mathcal{Q}}(x))^2}{\sum_{x_t \in \mathcal{Q}} (f^*(x_t) - \hat{f}_{\mathcal{Q}}(x_t))^2 + 1} \left( \sum_{x_t \in \mathcal{Q}} (f^*(x_t) - \hat{f}_{\mathcal{Q}}(x_t))^2 + 1 \right) \\
&= \frac{(f^*(x) - \hat{f}_{\mathcal{Q}}(x))^2}{\sum_{x_t \in \mathcal{Q}_\ell} (f^*(x_t) - \hat{f}_{\mathcal{Q}}(x_t))^2 + 1} \left( \sum_{x_t \in \mathcal{Q}} (f^*(x_t) - \hat{f}_{\mathcal{Q}}(x_t))^2 + 1 \right) \\
&\le \sup_{f,g \in \mathcal{F}} \frac{(f(x) - g(x))^2}{\sum_{x_t \in \mathcal{Q}_\ell} (f(x_t) - g(x_t))^2 + 1} \left( \sum_{x_t \in \mathcal{Q}} (f^*(x_t) - \hat{f}_{\mathcal{Q}}(x_t))^2 + 1 \right) \\
&= D^2(x; \mathcal{Q}_\ell) \left( \sum_{x_t \in \mathcal{Q}} (f^*(x_t) - \hat{f}_{\mathcal{Q}}(x_t))^2 + 1 \right)
\end{aligned}
$$

$$\leq \varepsilon_\ell^2 \cdot \mathfrak{R}(\delta, T)$$
$$= (2^{-\ell})^2$$

where the second to last line holds with probability $1 - \delta/T$.

We also have for all $\ell$, for all $x \in \mathcal{P}_{\ell-1} \smallsetminus \mathcal{Q}_\ell$,

$$(f^*(x) - \hat{f}_{\mathcal{Q}_\ell}(x))^2$$

$$= \frac{(f^*(x) - \hat{f}_{\mathcal{Q}_\ell}(x))^2}{\sum_{x_t \in \mathcal{Q}_\ell}(f^*(x_t) - \hat{f}_{\mathcal{Q}_\ell}(x_t))^2 + 1} \left( \sum_{x_t \in \mathcal{Q}_\ell}(f^*(x_t) - \hat{f}_{\mathcal{Q}_\ell}(x_t))^2 + 1 \right)$$

$$\leq \sup_{f,g \in \mathcal{F}} \frac{(f(x) - g(x))^2}{\sum_{x_t \in \mathcal{Q}_\ell}(f(x_t) - g(x_t))^2 + 1} \left( \sum_{x_t \in \mathcal{Q}_\ell}(f^*(x_t) - \hat{f}_{\mathcal{Q}_\ell}(x_t))^2 + 1 \right)$$

$$= D^2(x; \mathcal{Q}_\ell) \left( \sum_{x_t \in \mathcal{Q}_\ell}(f^*(x_t) - \hat{f}_{\mathcal{Q}_\ell}(x_t))^2 + 1 \right)$$

$$\leq \varepsilon_\ell^2 \cdot \mathfrak{R}(\delta, T)$$

$$= (2^{-\ell})^2$$

where the second to last line holds with probability $1 - \delta/T$. Summing over the rest over all stages $\ell < T$ with a union bound yields the proof. $\qquad\square$

**Lemma F.2.** *For every $\ell$, for every $x \in \mathcal{C}_\ell$, $\operatorname{sign}(\hat{f}_{\mathcal{Q}_\ell}(x)) = \operatorname{sign}(\hat{f}_{\mathcal{Q}}(x)) = \operatorname{sign}(f^*(x))$ and $|\hat{f}_{\mathcal{Q}}(x) - 1/2| > 2^\ell$.*

**Proof.** Adapted from Lemma 17 from Gentile et al. (2022).

For every stage $\ell$, every $x \in \mathcal{C}_\ell$, we know that $|\hat{f}_{\mathcal{Q}_\ell}(x) - 1/2| > 3 \cdot 2^\ell$ by the design of Algorithm 3. Putting this together with Lemma F.1, since $\mathcal{C}_\ell \subseteq \mathcal{P}_{\ell-1} \smallsetminus \mathcal{Q}_\ell$, we must have for every $\ell$, for every $x \in \mathcal{C}_\ell$, $\operatorname{sign}(\hat{f}_{\mathcal{Q}_\ell}(x)) = \operatorname{sign}(\hat{f}_{\mathcal{Q}}(x)) = \operatorname{sign}(f^*(x))$. Furthermore, we have that $|f^*(x) - 1/2| > 2 \cdot 2^\ell$ and $|\hat{f}_{\mathcal{Q}}(x) - 1/2| > 2^\ell$. $\qquad\square$

**Lemma F.3.** *Let $\hat{f}_S \in \mathcal{F}$ be the predictor returned by a regression oracle on sample $S$ and let $\hat{f}_{S\smallsetminus i} \in \mathcal{F}$ be the predictor returned by a regression oracle on sample $S \smallsetminus \{x_i\}$. If the regression oracle satisfies uniform stability under the squared loss, $\ell(\hat{y}, y) = (\hat{y} - y)^2$, then for all $S = \mathcal{Z}^n$, for all $i \in [n]$, , for all $\{x_1, \ldots, x_n\}$,*

$$\sum_{t=1}^n (\hat{f}_{S\smallsetminus i}(x_t) - \hat{f}_S(x_t))^2 \leq n \cdot \beta(n)^2.$$

**Proof.** Since the regression oracle satisfies uniform stability under the squared loss, $\ell(\hat{y}, y) = (\hat{y} - y)^2$, we have for all $S \in \mathcal{Z}^n$, for all $i \in [n]$, for all $z = (x, y) \in \mathcal{Z}$,

$$|(\hat{f}_S(x) - y)^2 - (\hat{f}_{S\smallsetminus i}(x) - y)^2| \leq \beta(n).$$

Expanding the left hand side, we have

$$|(\hat{f}_S(x) - y)^2 - (\hat{f}_{S\smallsetminus i}(x) - y)^2| = |\hat{f}_S(x)^2 - 2y\hat{f}_S(x) - y^2 - \hat{f}_{S\smallsetminus i}(x)^2 + 2y\hat{f}_{S\smallsetminus i}(x) + y^2|$$

$$= |\hat{f}_S(x)^2 - \hat{f}_{S\smallsetminus i}(x)^2 + 2y(\hat{f}_{S\smallsetminus i}(x) - \hat{f}_S(x))|$$

$$= |\hat{f}_S(x)^2 - \hat{f}_{S\smallsetminus i}(x)^2| + 2|\hat{f}_{S\smallsetminus i}(x) - \hat{f}_S(x)||$$

$$\text{(holds for all } y \text{, so set } y = \{-1, +1\} \text{ accordingly)}$$

$$\geq 2|\hat{f}_{S\smallsetminus i}(x) - \hat{f}_S(x)|$$

From uniform stability, we can conclude

$$|\hat{f}_{S\smallsetminus i}(x) - \hat{f}_S(x)| \leq \beta(n).$$

Thus,

$$\sum_{t=1}^{n} (\hat{f}_{S\smallsetminus i}(x_t) - \hat{f}_S(x_t))^2 \le n \cdot \beta(n)^2.$$

$\square$

## G. Helpful Theorems

**Proposition G.1** (Agarwal (2013), Proposition 1). *With probability at least $1 - \delta$, for all $t \in [T]$,*

$$\|w_t - u\|_{A_t} \le O\left(\sqrt{d \log T \cdot \log(1/\delta)}\right).$$

**Theorem G.2** (Ben-Hamou et al. (2018), Theorem 1). *Let $X$ be the cumulative value of sequence of length $n \le N$ drawn from $\Omega$ without replacement,*

$$X = \nu(\mathbf{I}_1) + \cdots + \nu(\mathbf{I}_n),$$

*and let $Y$ be the cumulative value of sequence of length $n \le N$ drawn from $\Omega$ with replacement,*

$$Y = \nu(\mathbf{J}_1) + \cdots + \nu(\mathbf{J}_n).$$

*If the value function $\nu$ and the weight vector $W$ follow the property that*

$$\omega(i) > \omega(j) \implies \nu(i) \ge \nu(j),$$

*then*

$$\mathbb{E}[X] \le \mathbb{E}[Y].$$

**Theorem G.3** (Gentile et al. (2022), Theorem 19). *For any pool realization $\mathcal{P}$, the label complexity $N_T$ of Algorithm 3 operating on a pool $\mathcal{P}$ of size $T$ is bounded deterministically as*

$$N_T \le 4^{L+1} \cdot \mathfrak{R}(T, \delta) \cdot \mathfrak{D}(\mathcal{F}, \mathcal{P})$$

