# OpenReview forum: "System-Aware Unlearning Algorithms: Use Lesser, Forget Faster"
_ICML.cc/2025/Conference — ICML 2025 poster_

### Official Review · Reviewer_BX2n · 2025-03-13

**Overall Recommendation:** 2

**Summary:**

This paper introduces a system-aware unlearning framework, which is a new definition of machine unlearning that relaxes the unlearning definition by assuming a weaker attacker that has no access to the full training data, i.e., the definition only requires the unlearned model to be indistinguishable from a model trained on a subset of the data ("core set" called in the paper).

The paper mainly focuses on linear classification, and the proposed method employs the selective sampling algorithm BBQSampler to select the core set from the training data to conduct unlearning.

**Claims And Evidence:**

The claims in the paper are not convincing to me.

The paper claims that the new definition provides privacy guarantees by ensuring that the unlearned model using the core set reveals no more information than the traditional unlearned model. However, privacy leakage does not happen due to the definition of unlearning, it could arise from the networks or algorithms themselves, such as model inversion attacks. Moreover, with the traditional definition, data-free machine unlearning methods [1-4] were proposed and would provide more protection than the proposed method using partial data.

[1] Foster, J., Schoepf, S., and Brintrup, A. Fast machine unlearning without retraining through selective synaptic dampening. In Proceedings of the AAAI Conference on Artificial Intelligence, pp. 12043–12051, 2024.

[2] Tarun, A. K., Chundawat, V. S., Mandal, M., and Kankanhalli, M. Fast yet effective machine unlearning. IEEE Transactions on Neural Networks and Learning Systems, pp. 1–10, 2023.

[3] Bonato, J., Cotogni, M., and Sabetta, L. Is retain set all you need in machine unlearning? restoring performance of unlearned models with out-of-distribution images. In European Conference on Computer Vision, pp. 1–19. Springer, 2025.

[4] Chundawat, V. S., Tarun, A. K., Mandal, M., and Kankanhalli, M. Zero-shot machine unlearning. IEEE Transactions on Information Forensics and Security, pp. 2345– 2354, 2023b.

**Essential References Not Discussed:**

[1-4], for instance. The authors are encouraged to investigate and check recent machine unlearning papers.

**Experimental Designs Or Analyses:**

The paper mainly focuses on linear classification. However, current machine unlearning methods mostly examine with deep neural networks such as ResNet, ViT and even large-scale vision-language model CLIP [1-5].

[5] Poppi, Samuele, et al. "Safe-CLIP: Removing NSFW concepts from vision-and-language models." European Conference on Computer Vision. Cham: Springer Nature Switzerland, 2024.

**Methods And Evaluation Criteria:**

The proposed method using partial training data for unlearning makes sense, but the authors are encouraged to investigate relevant works, such as data-free methods [1-4], for better understanding the development of machine unlearning methods.
To me, the paper just proposes a method with only access to partial training data, and use selective sampling algorithm to identify the core set. If the paper only focus on exact unlearning with partial training data, what is the advantage of the proposed method to approximate (data-free) unlearning methods? For different deletion requests, the proposed method need to find a core set for the requests, which is impractical in the end.

**Other Comments Or Suggestions:**

Please see the above sections.

**Other Strengths And Weaknesses:**

The core idea of using sample compression for unlearning is interesting but is restricted to linear classification. The argument that it enhances privacy is not convincing (See above).

**Questions For Authors:**

- In Theorem 2.4, does $S \setminus U=( (S^{'} \setminus U), ( (S \setminus S^{'}) \setminus U ) )$ mean $S \setminus U=(S^{'} \setminus U) \cup ( (S \setminus S^{'}) \setminus U ) $?

- What does "measurable sets F" mean in Definition 2.3?

**Relation To Broader Scientific Literature:**

The key contribution of this paper is to use the selective sampling algorithm BBQSampler to select the core set from the training data to conduct unlearning for linear classification.

**Theoretical Claims:**

I only checked those till Section 4. Please see Claims And Evidence*.

---

> ### Author Rebuttal · Authors · 2025-03-31
>
> > CE 1 - *The claims in the paper are not convincing to me. The paper claims that the new definition ...*
>
> The authors disagree with the claim that data-free machine unlearning methods would provide more privacy protection than the proposed method. Data-free machine unlearning methods still attempt to recover a model approximately equivalent to retraining a model from scratch on the remaining dataset S \ U, which is the traditional definition of unlearning. Furthermore, [1, 2, 3, 4] do not provide any rigorous guarantee that such an unlearning goal is actually being achieved. In fact, [5] has shown that SSD [1] fails to properly unlearn. Recent work [5, 6] has highlighted that many empirical methods for unlearning fail to properly unlearn and do indeed leak information about deleted individuals. This shows a dire need for algorithms that meet theoretical guarantees of unlearning.
>
> Additionally, we present a new definition of unlearning, system-aware unlearning, which provides privacy guarantees that are provably stronger than traditional unlearning. Then, we provide an algorithm that provably satisfies system-aware unlearning, and we provably demonstrate that system-aware unlearning algorithms for linear classification are more memory and computation efficient than traditional unlearning algorithms.
>
> > MAEC 1 - *The proposed method using partial training data for unlearning makes sense, but the ...*
>
> The method in our paper does not only have access to partial training data. The system-aware algorithm initially has access to the entire dataset and then uses selective sampling to select the most important points to form a core set in order to facilitate faster and more efficient unlearning in the future.
>
> Furthermore, it is important to note that the core set is computed once during the initial learning phase and fixed before any deletion requests arrive and then never computed again. As various deletion requests arrive, Algorithm 1 performs an unlearning update and does not recompute a new core set for every new set of requests. This method is provably more efficient than traditional unlearning algorithms.
>
> > EDOA 1 - *The paper mainly focuses on linear classification. However, current machine unlearning ...*
>
> Even under the traditional unlearning definition, it is still unclear how to perform theoretically rigorous and efficient unlearning in simple models like regression. We focus on algorithms with theoretically rigorous unlearning guarantees.
>
> > QFA 1 - *In Theorem 2.4, does $S \setminus U = ((S’ \setminus U), ((S \setminus S’) \setminus U))$ mean $S \ U =(S’ \setminus U) \cap ((S \setminus S’) \setminus U))$?*
>
> In Theorem 2.4, we treat $S \setminus U$ as a vector that can be tensorized into two parts, $((S’ \setminus U), ((S \setminus S’) \setminus U))$, so that we can apply the chain rule of mutual information. We will clarify this in the final version of the paper.
>
> > QFA 2 - *What does "measurable sets F" mean in Definition 2.3?*
>
> $\mathcal{F}$ is the $\sigma$-algebra over the outcome space of possible system-states. Measurable sets $F$ can be thought of as all possible events (subsets of outcomes) over the $\sigma$-algebra $\mathcal{F}$. We note that this notation is standard for all indistinguishability-style definitions.
>
> Please let us know if you have any additional questions or concerns.
>
> [1] Foster, J., Schoepf, S., and Brintrup, A. Fast machine unlearning without retraining through selective synaptic dampening. In Proceedings of the AAAI Conference on Artificial Intelligence, pp. 12043–12051, 2024.
>
> [2] Tarun, A. K., Chundawat, V. S., Mandal, M., and Kankanhalli, M. Fast yet effective machine unlearning. IEEE Transactions on Neural Networks and Learning Systems, pp. 1–10, 2023.
>
> [3] Bonato, J., Cotogni, M., and Sabetta, L. Is retain set all you need in machine unlearning? restoring performance of unlearned models with out-of-distribution images. In European Conference on Computer Vision, pp. 1–19. Springer, 2025.
>
> [4] Chundawat, V. S., Tarun, A. K., Mandal, M., and Kankanhalli, M. Zero-shot machine unlearning. IEEE Transactions on Information Forensics and Security, pp. 2345– 2354, 2023b.
>
> [5] Machine Unlearning Fails to Remove Data Poisoning Attacks. Martin Pawelczyk, Jimmy Z. Di, Yiwei Lu, Gautam Kamath, Ayush Sekhari, Seth Neel. ICLR 2025.
>
> [6] Inexact Unlearning Needs More Careful Evaluations to Avoid a False Sense of Privacy. Jamie Hayes, Ilia Shumailov, Eleni Triantafillou, Amr Khalifa, Nicolas Papernot. SatML 2025.

---

> > ### Comment · Reviewer_BX2n · 2025-04-06
> >
> > Thank the authors for the response. The response addressed some of my concerns, so I raised my score to 2.
> >
> > However, even replace it with the System-Aware definition, for data-free methods, that said, S’ \ U = $\emptyset$, the State-of-System $I_A(S, U)$ would not save any remaining data in the system, therefore providing more privacy protection than the proposed method.
> > The paper also claims that, S’ \ U can not reveal additional information about U than S \ U; but this does not mean that it provides a rigorous privacy guarantee. So, only S’ \ U may still leak information about deleted individuals. For example, if U share the same distribution as S’ \ U, then the risk of leakage from S’ \ U could still be high since the selected S' is a good representative of S.
> >
> > I am currently kind of confused about the definition, it seems to only introduce the state of the system (what is saved in the system by the unlearning algorithm after unlearning) to the DP-like unlearning definition; so from the perspective of an attacker who only observes the model after unlearning and stored samples. If so, we just need to design unlearning algorithms with few or even without access to training data, and this means that, the extent of privacy protection is not affected by the definition, instead, it is affected by the unlearning algorithms. Please let me know if I misunderstood something here.

---

> > > ### Author Response · Authors · 2025-04-08
> > >
> > > > *However, even replace it with the System-Aware definition, for data-free methods, that...*
> > >
> > > The state-of-system is not just the stored samples but everything stored in memory by the unlearning algorithm. At the very least, data-free methods store the model in the system; thus, the model itself is a part of the state-of-system. The model stored in the system by these data-free methods relies on all of the samples in S, even if the samples are not explicitly stored. Thus, the overall state-of-system (which contains the model) relies on the entirety of S. Furthermore, the data-free method from [2] actually requires access to some retain set samples in order to unlearn; these samples would have to be stored in the system.
> > >
> > > [2] Tarun, A. K., Chundawat, V. S., Mandal, M., and Kankanhalli, M. Fast yet effective machine unlearning. IEEE Transactions on Neural Networks and Learning Systems, pp. 1–10, 2023.
> > >
> > > > *The paper also claims that, S’ \ U can not reveal additional information about U than...*
> > >
> > > It is true that S’ \ U may leak some information about U. However, Theorem 2.4 shows that any information leakage between S’ \ U and U must also be present between S \ U and U. Since traditional unlearning attempts to recover retraining-from-scratch on S \ U, this privacy leakage about U must have also been present under the traditional definition of unlearning, which is considered the gold standard. In particular, we only provide a relative guarantee of privacy. The privacy guarantee of system-aware unlearning is at least as strong as traditional unlearning. In fact, if no assumptions are placed on the data generation process, absolute privacy guarantees or unlearning guarantees are impossible.
> > >
> > > > *I am currently kind of confused about the definition, it seems to only introduce the...*
> > >
> > > We think the confusion is perhaps stemming from the reviewer not thinking of $\mathsf{I}_A$ as a functional. For an unlearning algorithm without access to training samples (either in direct or indirect form), we must have $\mathsf{I}_A(S, U) = \mathsf{I}_A(S, \emptyset)$ for all deletion requests $U$ (or $\mathsf{I}_A(S’, U) = \mathsf{I}_A(S’, \emptyset)$ for all $U$). With this condition, the only unlearning algorithm would be one which has a constant state. (For a quick explanation of why this is true, consider the following. Unlearning requires $\mathsf{I}_A(S, U) = \mathsf{I}_A(S \setminus U, \emptyset)$ for all $U$, and we also know that $\mathsf{I}_A(S, U) = \mathsf{I}_A(S, \emptyset)$ for all $U$. Consider when $U=S$, we have $\mathsf{I}_A(S, U) = \mathsf{I}_A(\emptyset, \emptyset)$, thus we must also have $\mathsf{I}_A(S, \emptyset) = \mathsf{I}_A(S, U) = \mathsf{I}_A(\emptyset, \emptyset)$. Thus, the state after unlearning (which includes the model) cannot depend on the dataset $S$, even if the set of deletion requests is the empty set, $U = \emptyset$.) This concurs with the traditional belief that an unlearning algorithm that throws away the trained model and outputs a constant function (independent of the dataset) is a valid unlearning algorithm. However, while unlearning is preserved in this case, one cannot expect non-trivial performance.
> > >
> > > We show that the privacy guarantee of system-aware unlearning is at least as strong as traditional unlearning. However, through the flexibility of S’ (again it is very important that S’ is chosen only based on S before any unlearning request, and hence S’ captures the essence of the sample set as viewed by the learning algorithm even before any unlearning requests), we can design system-aware unlearning algorithms that are more efficient than traditional unlearning algorithms. We can achieve the same privacy protections using less memory and computation resources. The fact that we can achieve the same privacy protections while gaining additional flexibility in the design of unlearning algorithms is an advantage of system-aware unlearning.
> > >
> > > We thank the reviewer for their valuable feedback and discussion to help improve our paper.

---

### Official Review · Reviewer_C9YU · 2025-03-13

**Overall Recommendation:** 1

**Summary:**

The authors propose system-aware machine unlearning, which constitutes unlearning against an attacker who can observe the entire state of the system (including whatever the learning system uses internally). If the system does store the entire remaining dataset, then system-aware unlearning definition becomes as stringent as traditional unlearning, but otherwise is a relaxation constrained only for the data stored by the system. The paper only provides theoretical insights about the proposed system-aware machine unlearning setting.

___

## update after rebuttal

I would like to thank the authors for their response. However, my fundamental concerns still remain -- despite the claim that the work aims to make theoretical contributions in defining system-aware unlearning for linear classifiers, I believe an empirical validation is necessary given that unlearning is a practical problem and existing baselines can be adapted to the system-aware definition. There are also issues in the current version regarding inconsistent phrasing/statements of non-convexity (which the authors mentioned they will rewrite) and the fact that SISA outperforms their method for a significant duration of the deletion phase. While there are minimalistic experiments in the paper, I will keep my score largely given the non-existent evaluation against relevant baselines.

**Claims And Evidence:**

- Under the definitions/assumptions made by the authors in Section 2 for linear classification, the theoretical claims are well justified. But the overall motivation of this work and the claims in the context of existing work in the machine unlearning domain are unclear, significantly limiting its usefulness.
- To fully verify the claims being made (as unlearning is a task with practical real-world use-cases), the paper needs extensive evaluation against existing methods and approaches for unlearning. Machine unlearning methods need to be evaluated using both the unlearning efficiency and the utility of the model post unlearning across a number of existing methods. Currently, the few experimental results present are deferred to the appendix and only consider the SISA approach and exact retraining (both over 4 years old). Moreover, SISA seems to perform better than the proposed algorithm for a significant duration of the deletion phase. The paper is basically lacking a Results and Experimental Evaluation section currently.
- Given the authors' statement in the Introduction (line 50, right hand column): _"For large datasets, this makes unlearning under
the traditional definitions impractical"_. Yet, there exist approximate methods that work with large datasets and a large number of parameters (e.g. LLMs) that work well while working in the limited data access regime (e.g. [1,2]). The paper focuses on linear classifiers limiting its applicability to models that actually require a very large number of data samples. It would also be good to compare empirically with more recent approximate existing methods along the aforementioned metrics (authors can refer to [3] for more details) and on large datasets. Without these necessary experiments and analyses, the proposed method cannot be fully evaluated from an empirical perspective.
- The authors state in the Introduction (line 43, right hand column): _"This is evidenced by a dire lack of efficient exact or approximate unlearning algorithms beyond the simple case of convex loss functions."_ I am not sure if this claim is correct. Is the SISA method the authors compare with itself not designed for non-convex losses? And given the large amount of approximate approaches [1-2,4-5] that work with LLMs and deep neural networks (inherently non-convex), I believe the authors need to rewrite this statement to reflect a correct and nuanced perspective.

References:
1. Huang, James Y., et al. "Offset unlearning for large language models." arXiv preprint arXiv:2404.11045 (2024).
2. Ji, Jiabao, et al. "Reversing the forget-retain objectives: An efficient llm unlearning framework from logit difference." NeurIPS (2024).
3. Wang, Weiqi, et al. "Machine unlearning: A comprehensive survey." arXiv preprint arXiv:2405.07406 (2024).
4. Liu, Sijia, et al. "Rethinking machine unlearning for large language models." Nature Machine Intelligence (2025): 1-14.
5. Liu, Zheyuan, et al. "Machine unlearning in generative ai: A survey." arXiv preprint arXiv:2407.20516 (2024).

**Essential References Not Discussed:**

There are many references that are missing; primarily because the paper is lacking a Related Works section in the main text. There is some discussion of related papers in the appendix, but these need to be moved to the main paper and more works should be discussed. This is not an exhaustive list, but some references (as mentioned in above sections), include:

1. Huang, James Y., et al. "Offset unlearning for large language models." arXiv preprint arXiv:2404.11045 (2024).
2. Ji, Jiabao, et al. "Reversing the forget-retain objectives: An efficient llm unlearning framework from logit difference." NeurIPS (2024).
3. Wang, Weiqi, et al. "Machine unlearning: A comprehensive survey." arXiv preprint arXiv:2405.07406 (2024).
4. Liu, Sijia, et al. "Rethinking machine unlearning for large language models." Nature Machine Intelligence (2025): 1-14.
5. Liu, Zheyuan, et al. "Machine unlearning in generative ai: A survey." arXiv preprint arXiv:2407.20516 (2024).

**Experimental Designs Or Analyses:**

As mentioned in the Methods And Evaluation Criteria section, there is very limited (and almost non-existent) evaluation. Please see above (Claims and Evidence) for more issues regarding the claims made in the context of methods and experiments. At the moment, there are very few experimental results that are present in the appendix and only consider the SISA approach and exact retraining (both over 4 years old). Moreover, SISA seems to perform better than the proposed algorithm for a significant duration of the deletion phase. The paper needs an extensive Evaluation section for a clear evaluation.

**Methods And Evaluation Criteria:**

There is very limited (and almost non-existent) evaluation. Please see above (Claims and Evidence) for more issues regarding the claims made in the context of methods and evaluation criteria. At the moment, there are very few experimental results that are present in the appendix and only consider the SISA approach and exact retraining (both over 4 years old). Moreover, SISA seems to perform better than the proposed algorithm for a significant duration of the deletion phase. The paper needs an extensive Evaluation section for a clear evaluation.

**Other Comments Or Suggestions:**

N/A.

**Other Strengths And Weaknesses:**

While the ideas in this paper have merit, the limited evaluation makes the work somewhat incomplete. For more details, please see the Methods And Evaluation Criteria and Claims and Evidence sections above.

**Questions For Authors:**

Please see the Methods And Evaluation Criteria and Claims and Evidence sections above. Each of the points can be considered as questions and points of concern.

**Relation To Broader Scientific Literature:**

The paper can be of interest to the machine unlearning community. However, the limited evaluation limits its effectiveness and scope, and could lead to a reduced impact in the field.

**Theoretical Claims:**

Yes, from a cursory verification of the theoretical proofs, I did not find any errors.

---

> ### Author Rebuttal · Authors · 2025-03-31
>
> > CE 2 - *To fully verify the claims being made (as unlearning is a task with practical real-world ...*
>
> > MAEC 1 - *There is very limited (and almost non-existent) evaluation...*
>
> We emphasize that our primary contribution is theoretical. We focus on unlearning algorithms with provable unlearning guarantees. Recent work [1, 2] has demonstrated that many empirical methods fail to properly unlearn. This shows a dire need for algorithms that provide theoretical guarantees of unlearning; thus, we focus on specifically unlearning algorithms that meet theoretical guarantees of certified unlearning. Our paper is a theoretical paper with a focus on developing a new definition for unlearning that is not as pessimistic as the traditional unlearning definition but is still principled. We provably demonstrate that system-aware unlearning algorithms for linear classification are more memory and computation efficient than traditional unlearning algorithms.
>
> > CE 3 - *Given the authors' statement in the Introduction (line 50, right hand column): "For large ...*
>
> In this work, we demonstrate that for linear classification, system-aware unlearning leads to algorithms that are provably more efficient. Furthermore, even under the traditional unlearning definition, it is still unclear how to perform theoretically rigorous and efficient unlearning in simple models like regression. Although the exact methods may not translate directly, the flexibility of system-aware unlearning could also lead to significantly more efficient algorithms for more complex model classes. An empirical evaluation of unlearning with larger models and datasets is beyond the scope of this work, as we focus on system-aware unlearning and exact unlearning algorithms for linear classification with rigorous theoretical guarantees.
>
> > CE 4 - *The authors state in the Introduction (line 43, right hand column): "This is evidenced by ...*
>
> SISA requires the storage of many intermediate models and the entirety of the dataset to facilitate unlearning, which is extremely memory inefficient. We wanted to draw attention to the memory and computation inefficiencies of current unlearning algorithms, particularly those for nonconvex function classes. We will rewrite this statement for clarity.
>
> > ERND 1 - *There are many references that are missing; primarily because the paper is lacking a ...*
>
> In the final version, we will move the Related Works section to the main text and expand the references and discussions.
>
> Please let us know if you have any additional questions or concerns.
>
> [1] Machine Unlearning Fails to Remove Data Poisoning Attacks. Martin Pawelczyk, Jimmy Z. Di, Yiwei Lu, Gautam Kamath, Ayush Sekhari, Seth Neel. ICLR 2025.
>
> [2] Inexact Unlearning Needs More Careful Evaluations to Avoid a False Sense of Privacy. Jamie Hayes, Ilia Shumailov, Eleni Triantafillou, Amr Khalifa, Nicolas Papernot. SatML 2025.

---

> > ### Comment · Reviewer_C9YU · 2025-04-07
> >
> > I would like to thank the authors for their response. However, my fundamental concerns still remain -- despite the claim that the work aims to make theoretical contributions in defining system-aware unlearning for linear classifiers, I believe an empirical validation is necessary given that unlearning is a practical problem and existing baselines can be adapted to the system-aware definition. There are also issues in the current version regarding inconsistent phrasing/statements of non-convexity (which the authors mentioned they will rewrite) and the fact that SISA outperforms their method for a significant duration of the deletion phase. While there are minimalistic experiments in the paper, I will keep my score largely given the non-existent evaluation against relevant baselines.

---

### Official Review · Reviewer_ZP3T · 2025-03-14

**Overall Recommendation:** 3

**Summary:**

The authors propose a new definition for unlearning that they refer to as “system-aware unlearning” where the aim for the unlearned model is to be indistinguishable from a model that was trained on *any* subset of the training data excluding the forget set (rather than a model specifically trained on exactly the retain set), and where indistinguishability here is with respect to the “internal state” required by the model (e.g. any data that must be stored, any intermediate checkpoints that are required, the unlearned model for that request, or the ingredients required to produce it). The idea is that this definition enables the model developer to control what information is placed in the “internal state” (and that consequently an “attacker” would have access to). One could then design systems that support system-aware unlearning by requiring less information to be stored in the first place.
The authors then discuss core set algorithms, and how they facilitate connections between the standard definition of unlearning and system-aware unlearning. Namely, running a standard unlearning algorithm on a coreset S’ of the original dataset S enables system-aware unlearning (assuming only the coreset needs to be stored and not S itself) where for a large number of examples (those not in S’) unlearning is a no-op.

The authors propose an exact learning / unlearning algorithm for linear classification that uses a particular selective sampling strategy called BBQSampler from prior work to effectively get a coreset. Then, to handle a deletion, interventions are only required if the examples to be deleted is part of the “coreset”, i.e. a set of “queried” examples. These interventions required to unlearn a point are efficient in terms of computing the “new coreset” due to a monotonicity property. They prove bounds on deletion capacity, memory requirements, computation time.


########## update after the rebuttal

The authors have clarified several concerns of mine through extensive discussions (see thread below) and I'm confident that reflecting these discussions in the updated manuscript will improve the paper (I read the latest response, too, and while I didn't have time to reply to it directly earlier, I note here that the clarification w.r.t how the divergences are computed is helpful, thanks again). I maintain my recommendation of weak accept. The reason I don't recommend acceptance more strongly is due to the limitation of being applicable only to linear classifiers. I also agree with other reviewers that adding/strengthening the existing empirical investigation would also lead to a stronger contribution even though I understand the nature of the contribution here is theoretical.

**Claims And Evidence:**

- “a dire lack of efficient exact or approximate unlearning algorithms beyond the simple
case of convex loss functions” – while indeed there is a dire lack of certified unlearning algorithms for nonconvex models that are well performing, there are various (uncertified) ones that are efficient, and there is work on empirically auditing them, see e.g. Hayes et al. in the references below.

- “only the privacy of the stored points are at jeopardy as long as the learnt model does not reveal much about points that were not used by the model” – this doesn’t seem to be true in general, since not all data selection mechanisms preserve privacy. Even if only a subset of the training dataset is accessible, this may leak privacy about other training data points (that were used, for instance, for selecting which subset to store).

- “Even if an observer or attacker has access to larger public data sets that might include parts of the data the system was trained on, in such a system, we could expect privacy for data that the system does not use directly for building the model to be preserved.” – similarly here, why is this the case? In general, an example can influence the final computation even if it isn’t “directly” used for training. And this influence can translate to privacy vulnerability in general, right?

- “Traditional unlearning definitions that require the unlearned hypothesis to approximately match A(S ∖ U, ∅) implicitly assume that the information between the deleted individuals U and the remaining dataset S ∖ U is small.” I don’t understand why the authors claim that this assumption is needed. The traditional DP-like unlearning definitions are about measuring the difference between the allowed knowledge one would have on U even if having trained just on S ∖ U compared to the *additional* knowledge about U that stems from having trained on U (and not perfectly removed its influence).  But this doesn’t require that the MI between U and S ∖ U is small?

- In the definition of system-aware unlearning, how should I interpret the probability Pr(IA(S, U ) ∈ F )? In the standard definition, the probabilities are over the output of running either the “retraining” or the “unlearning” recipe and in either case, that output is a set of model weights. When we replace this with the  “state of the system” (which can contain a variety of things like model checkpoints, but also data points, etc), it’s less clear to me how to interpret this probability distribution, or what this divergence exactly means in this case.

- “We point out that the traditional definition of unlearning does not require indistinguishability for auxiliary information stored in the system outside of the unlearned model; thus, the traditional definition does not account for system-awarenes” – I agree in general, but see the distinction introduced in Neel et al. about “perfect unlearning” relating to whether a secret state (that the attacker can’t access) can be kept or not which is relevant to discuss here.

**Essential References Not Discussed:**

- “When machine unlearning jeopardizes privacy” (see references below) shows that, with access to intermediate checkpoints of the model (what can be considered as being part of an “internal state” here), an attacker can even succeed against exact unlearning! This would be interesting to incorporate in the discussion of this paper (since it also relates to the relationship between auxiliary information that the attacker can access vis-a-vis privacy claims or guarantees; even in the case of “exact unlearning”).

- Definition 1 in Golatkar et al. (see references below) has a similar flavour of requiring the “existence of a certificate” such that closeness to that verifies an algorithm as successful unlearning (rather than a fixed recipe as a certificate). This is reminiscent of the “there exists an S’ …” modification in the proposed definition. There are various differences between that definition and the one proposed here, but this similarity is interesting to acknowledge.


References
==========

- Inexact Unlearning Needs More Careful Evaluations to Avoid a False Sense of Privacy. Hayes et al. SatML 2025.

- Descent-to-Delete: Gradient-Based Methods for Machine Unlearning. Neel et al. 2020.

- When machine unlearning jeopardizes privacy. Chen et al. ACM SIGSAC 2021.

- Eternal Sunshine of the Spotless Net: Selective Forgetting in Deep Networks. Golatkar et al. CVPR 2020.

**Experimental Designs Or Analyses:**

N/A

**Methods And Evaluation Criteria:**

N/A

**Other Comments Or Suggestions:**

Minor:

- Line 196 (right column): “S′ ∖ U should leak any more information about U as compared to S ∖U” – I think the authors here mean “should *not* leak any more information [...]”.

- Initially, e.g. in Definition 2.1, the authors use A to denote a learning algorithm and \bar A to denote an unlearning algorithm. But later, in various places, A denotes the unlearning algorithm (and A(S,U) the unlearned model). This discrepancy in notation can cause confusion.

- Many new symbols introduced in Algorithm 1, it would be helpful to describe them intuitively.

- In Theorem 4.2, define A (also, the symbol A is overloaded – elsewhere in the paper it denotes a learning or unlearning algorithm, here it appears to be some matrix).

**Other Strengths And Weaknesses:**

Strengths

- The paper is thought-provoking, proposes an interesting alternative definition of unlearning.

- The authors propose an efficient algorithm for unlearning in linear classification that they prove satisfies exact system-aware unlearning. They prove bounds for how many unlearning requests their algorithm can handle without too big a drop in performance (deletion capacity), memory requirements and amount of computation required.

- The paper is for the most part well-written, giving intuition behind various theoretical results that makes it easier to follow.


Weaknesses

- It’s unclear to me when is system aware unlearning useful in practice, what application is it most appropriate for? I have concerns (discussed above) about its privacy protection, but maybe the authors can convince me otherwise in the rebuttal. For example, for the compression, if we use all of S as input to a “compression function” that outputs a representative subset S’ on top of which we do further processing, it does not mean that only S’ influenced the final model. S influenced the final model here too and that influence depends on the compression function and whether it is privacy preserving.

- The proposed algorithm is only applicable to linear classification. Can it be extended to neural networks?

- “Assume that deletions are drawn without replacement [...]” – often not a realistic assumption as in practice unlearning requests may be correlated (e.g. groups of users that are unhappy with the model and share various characteristics with each other, would be more likely to request data deletion).

**Questions For Authors:**

- “Thus, S′ ∖ U leaks no more information about U than S ∖ U” – what exactly is the privacy guarantee that this implies? Can you help me understand how this statement translates to a privacy guarantee for U which is what subsection 2.1 is trying to argue I think?

- “under traditional unlearning, exact unlearning requires storing the entirety of the dataset” – can’t we be satisfied though with approximate unlearning that stores a random (or not random) subset of the dataset? I don’t quite see why this is impossible under traditional unlearning and we must resort to system aware unlearning here. (I can see it being impossible under *exact* traditional unlearning, but approximate might be okay in practice). What is the benefit of applying a corset algorithm and viewing it as exact system-aware unlearning as opposed to approximate “standard” unlearning?

**Relation To Broader Scientific Literature:**

See below section.

**Theoretical Claims:**

I did not check the proofs in the Appendix, but followed the proof sketches in the main paper.

---

> ### Author Rebuttal · Authors · 2025-03-31
>
> > CE 1
>
> We acknowledge that there are empirical unlearning algorithms that have demonstrated good performance on nonconvex models; however, recent work [1] has demonstrated that many such empirical methods fail to properly unlearn; thus, we focus on algorithms that meet theoretical guarantees of certified unlearning.
>
> > CE 2, CE 3
>
> Our definition (Definition 2.3) requires that for any S, there exists S’ such that *for any U*, $\textsf{I}_A(S, U)$ is indistinguishable from $\textsf{I}_A(S’ \setminus U, \emptyset)$. The order of the quantifiers ensures that S’ is chosen before U is selected, and thus the data selection mechanism is independent of U. Thus, S’ \ U can not reveal additional information about U than S \ U.
>
> > CE 4
>
> We remark that this is not a technical assumption used in any of our results. We only mention this to help interpret our results. Indeed, the mutual information (MI) between U and S ∖ U does not need to be small. However, such an assumption is implicitly necessary when considering unlearning for privacy purposes. One can easily construct scenarios where the MI between U and S ∖ U is large; thus, even retraining from scratch (the gold standard) may not preserve the privacy of deleted individuals U (as S \ U reveals U). Unlearning definitions that compete with retaining-from-scratch only provide meaningful privacy for the deleted individuals when the MI between U and S ∖ U is small. To motivate our definition, we show that the MI between U and S’ ∖ U never exceeds the MI between U and S ∖ U (Theorem 2.4).
>
> > CE 5
>
> The probabilities are over the outcome space of all possible system-states (model weights, saved gradients, stored samples, etc).
>
> > CE 6
>
> We thank the reviewer for this reference. We note that perfect unlearning is equivalent to system-aware unlearning when S=S’. We will add this reference.
>
> > ERND 1
>
> Unlearning definitions do not guarantee privacy against adversaries with continuous model observations. One could incorporate the intermediate checkpoints of the model into the state-of-system definition, resulting in a system-aware unlearning definition that would be robust to such adversaries. For Algorithm 1 specifically, since the core set is stored in memory, observing the system state before unlearning will reveal the deleted individuals in the core set. However, we can always provide privacy guarantees for individuals outside of the core set against such adversaries because they are never stored in the system at any point.
>
> > ERND 2
>
> We thank the reviewer for this reference. We agree that the similarity of the “existence of a certificate of forgetting” is interesting and will be sure to cite Golatkar et al. However, it is important to note that the unlearning algorithm in Golatkar et al. (OQSA) uses a certificate that computes the learning algorithm trained on S \ U, equivalent to traditional unlearning. Golatkar et al. did not capitalize on the flexibility of such a certificate. Our Algorithm 1 leverages the flexibility in S’ for more efficient unlearning.
>
> > OSAW W1
>
> System-aware unlearning is most useful when an accurate model can be learned with a small number of samples, such as models with small eluder dimension or disagreement coefficient [2].
>
> > OSAW W2
>
> Even under the traditional unlearning definition, it is still unclear how to perform theoretically rigorous and efficient unlearning in simple models like regression. We hope to extend our framework to more complex models, but this seems to be a significant challenge in traditional and system-aware unlearning.
>
> > OSAW W3
>
> This model of deletions can capture correlated deletions. For example, the deletion distribution may be such that users sharing certain characteristics have a higher probability of deletion than other users. We do assume that the deletion requests are not adaptive to system updates.
>
> > QFA 1
>
> Theorem 2.4 states that the information leakage between U and S’ ∖ U never exceeds the information between U and S ∖ U. This gives a relative information leakage guarantee that system-aware unlearning leaks no more information about the deleted individuals than traditional unlearning.
>
> > QFA 2
>
> So far, we do not know of any practical unlearning algorithms that can do approximate unlearning without storing the entire dataset in memory. In fact, [3] already shows a lower bound that even approximate unlearning (in the traditional definition) needs to store the entire dataset (while they focus on the realizability testing problem, we strongly suspect that their result can be extended to unlearning).
>
> We will clarify these discussions in the final paper. Please let us know if you have further questions.
>
> [1] Machine Unlearning Fails to Remove Data Poisoning Attacks. Pawelczyk et al. ICLR 2025.
>
> [2] Instance-Dependent Complexity of Contextual Bandits and Reinforcement Learning: A Disagreement-Based Perspective. Foster et al. COLT 2021.
>
> [3] On the unlearnability of the learnable. Cherapanamjeri et al. 2025.

---

> > ### Comment · Reviewer_ZP3T · 2025-04-06
> >
> > Hi authors, thanks a lot for the responses and additional discussion. Some follow-up thoughts:
> >
> > - CE 1: yes, I am aware of works showing limitations of existing uncertified methods in certain cases. However, methods that have guarantees instead have the drawback of requiring simplifying assumptions like convexity or smoothness. So it seems to me that neither certified nor uncertified approaches have yet "solved" the problem for non-convex models but both are worth discussing, and there has been a lot of work in both directions. So the claim “a dire lack of efficient exact or approximate unlearning algorithms beyond the simple case of convex loss functions” is not fully correct. I would prefer instead mentioning recent "sota" of each category (certified vs non-certified), and pointing out their respective weaknesses (such as those uncovered in Pawelczyk et al and Hayes et al for the case of non-certified methods).
> >
> > - CE 2/3: OK, I understand now about the order of picking S' before U (and it must work for any U) means that S’ \ U can not reveal additional information about U than S \ U. Thanks for clarifying.
> >
> > - CE 4: Thanks for clarifying that the MI assumption isn't required for any of your results. I actually disagree that it is implicitly necessary when considering unlearning for privacy purposes. It can be in some cases, but it depends on your definition of privacy. For Differential Privacy, we are simply concerned with the *additional* knowledge that the model has on a data point due to having trained on it (we are not concerned with knowledge that it might be able to infer about it due to having trained on other, similar, examples). And analogously for the usual definition of unlearning that required indistinguishability between the unlearned and retrained distributions: we are only concerned about the *additional* knowledge on the forget set that is due to having trained on it. Any knowledge that would have been obtained on it even when "retraining from scratch" is allowed, and doesn't deduct points from the estimated "unlearning quality" of the algorithm in question. This is the reason that I found these MI arguments confusing. I suggest the authors clarify in the revised text, at least to state, as they did in my response, that the results of the paper don't require an MI assumption.
> >
> > - CE 5: Thanks. Right, I understand conceptually that the probabilities are over the outcome space of all possible system-states. But how would one instantiate this / operationalize computing divergence between such distributions? It sounds practically much harder than when the probabilities are only in terms of model weights. Perhaps adding some discussion for this is helpful.
> >
> > - ERND 2: Agreed w.r.t that difference. It's only partially related due to the "certificate" aspect, and thought that was an interesting connection to discuss. But of course agree about the other key axis of difference there.
> >
> > - OSAW W1: OK. Can you think of a specific practical application where this condition would be met?
> >
> > - QFA 2: "So far, we do not know of any practical unlearning algorithms that can do approximate unlearning without storing the entire dataset in memory." Again, I think here you are referring to algorithms with guarantees specifically. In general, for the non-certified case, it is possible to do approximate unlearning without storing the entire dataset. Of course, the "unlearning quality" would need to be measured separately, empirically, if there is no certification. But I do believe this is an important direction too, and one that should not be discarded from the narrative.
> >
> > Overall, I maintain my opinion that this work is thought-provoking and a nice contribution and I thank the authors for the interesting discussion. I am inclined to keep my score of weak accept and to not raise it further because of concerns of limited (or unclear) practical applicability over other definitions (which applications is this definition most appropriate for compared to the standard one? how can one audit for this in practice if we don't have certification (which we don't for non-convex)?) as well as some clarification issues and the need for additional more well-rounded discussion, though I recognize some of that is personal preference. But I will read and take into consideration any additional follow-up comment from the authors.

---

> > > ### Author Response · Authors · 2025-04-08
> > >
> > > > *CE 1: Yes, I am aware of works showing limitations of existing uncertified methods in...*
> > >
> > > We will update this discussion in the paper to discuss both the strengths and weaknesses of certified and uncertified unlearning methods and add a comprehensive discussion about both the performance (or lack thereof in certified methods) and vulnerabilities of uncertified methods with appropriate citations. We agree that both directions are interesting to explore.
> > >
> > > > *CE 4: Thanks for clarifying that the MI assumption isn't required for any of your results...*
> > >
> > > We will update the paper to state and clarify that the results of the paper do not require a MI assumption. We will emphasize that system-aware unlearning provides a relative privacy guarantee, as is the case with differential privacy and traditional unlearning definitions. We thank the reviewer for pointing out this confusion.
> > >
> > > > *CE 5: Thanks. Right, I understand conceptually that the probabilities are over the...*
> > >
> > > One can compute divergences between system-states similarly to how one computes divergences across model weights. Many unlearning algorithms store “sufficient statistics” of the sample, approximately update these statistics when unlearning, and then compute an unlearned model as a function of these statistics [1, 2]. In fact, we point out that the way these works argue closeness after unlearning is by actually arguing the closeness of these approximate “sufficient statistics” to the true statistic of S \ U, rather than directly arguing closeness of the model weights. We can think of these stored statistics as exactly the state of the system. For example, [1] stores an approximate Hessian matrix of S \ U, $\hat{H}$, to assist with unlearning along with the model weight vector $w$. We can think of $\hat{H}$ and $w$ together as the state of the system. [1] first argues closeness between the approximate Hessian matrix of S \ U, $\hat{H}$ and the true Hessian matrix of S \ U, $H$. Since the unlearned model weight vector $w$ is simply a function of $\hat{H}$ and the retrained-from-scratch model is the same function of $H$, we immediately have closeness between the unlearned model and the retrained-from-scratch model. One can think of this as arguing that the system-states are close in order to conclude that the model weights are close. Thus, we do not believe that computing divergences across system-states poses an additional challenge than computing divergences across model weights. We have included a discussion of this in line 250 of the paper, but we will expand this discussion in the final version of the paper.
> > >
> > > [1] Sekhari, A., Acharya, J., Kamath, G., and Suresh, A. T. Remember what you want to forget: Algorithms for machine unlearning. NeurIPS 2021.
> > >
> > > [2] Guo, C., Goldstein, T., Hannun, A., and Van Der Maaten, L. Certified data removal from machine learning models. ICML 2019.
> > >
> > > > *OSAW W1: OK. Can you think of a specific practical application where this condition...*
> > >
> > > One could use influence functions [3] and data attribution techniques [4] to identify a subset of points that have the largest influence on the model weights and treat these “high-influence” samples as a core set. These techniques have been practically applied to deep learning models [3]. By only updating or retraining when one of these “high-influence” samples, we have fast average deletion time. While it is not clear how to give performance guarantees when one of these “high-influence” points is deleted, this provides a practical framework for system-aware unlearning that can be applied to general model classes.
> > >
> > > [3] Understanding Black-box Predictions via Influence Functions. Pang Wei Koh and Percy Liang. ICML 2017.
> > >
> > > [4] TRAK: Attributing Model Behavior at Scale. Sung Min Park, Kristian Georgiev, Andrew Ilyas, Guillaume Leclerc, Aleksander Madry. ICML 2023.
> > >
> > > > *QFA 2: "So far, we do not know of any practical unlearning algorithms that can do...*
> > >
> > > We do refer to certified algorithms here, as that is the focus of our paper. However, we will add an expanded discussion of uncertified approximate unlearning algorithms in the Related Work, including data-free empirical methods, which do not require the storage of the entire dataset.
> > >
> > > We thank the reviewer for their valuable feedback and perspective. We appreciate the time that the reviewer has taken to engage with our paper. We greatly value the helpful discussions to improve our paper.

---

### Official Review · Reviewer_Hefv · 2025-03-20

**Overall Recommendation:** 3

**Summary:**

The paper introduces a new system-aware unlearning setting, where attackers have access to only a partial dataset stored in the system rather than the entire dataset. The authors argue that this relaxation enables a more efficient and practical unlearning framework for attackers. To address this setting, they propose a selective sampling-based algorithm to identify the core set and provide  theoretical analysis for deletion capacity, risk and etc.

**Claims And Evidence:**

theoretical contributions: The work rigorously proves that system-aware unlearning generalizes traditional unlearning, providing bounds on deletion capacity, memory requirements, and excess risk.

**Essential References Not Discussed:**

There are also related work in [1] and [2] that emphasize the importance of system-aware perspectives in machine unlearning, the authors are encouraged to investigate relevant works to better describe its contribution in the development of machine unlearning methods through the lens of system-awareness


[1] Yuke Hu, Jian Lou, Jiaqi Liu, Feng Lin, Zhan Qin, and Kui Ren. Eraser: Machine unlearning in mlaas via an inference serving-aware
approach. Proceedings of the 2024 ACM SIGSAC Conference on Computer and Communications Security, 2024.
[2] Gaoyang Liu, Xiaoqiang Ma, Yang Yang, Chen Wang, and Jiangchuan Liu. Federaser: Enabling efficient client-level data removal from
federated learning models. In 2021 IEEE/ACM 29th International Symposium on Quality of Service (IWQOS), pages 1–10. IEEE, 2021

**Experimental Designs Or Analyses:**

* The experimental setup is relatively simple and limited. The current study focuses on linear classification, which, while important, represents a rather basic scenario. How would the method perform on non-convex deep models or, at the very least, on SVMs?

* Regarding the experiments, would it be possible to provide further evaluations on how the selection process performs under different distributions and sample size of S'

**Methods And Evaluation Criteria:**

Under partial training setting,  the proposed method use selective sampling-based to select core set points where the label was queried make senses to me and provide interesting perspective to save computation time and capacity.

**Other Comments Or Suggestions:**

see above

**Other Strengths And Weaknesses:**

see above

**Questions For Authors:**

see above

**Relation To Broader Scientific Literature:**

The paper broadens the scope of machine unlearning from systematic perspective by shifting from the traditional unlearning model to a more practical system-aware framework

**Theoretical Claims:**

Lemma 4.7: Let the deletion distribution μ be the uniform distribution. Is it always valid to assume this scenario? Wouldn't there be data-dependent deletion distributions?

---

> ### Author Rebuttal · Authors · 2025-03-31
>
> > TC 1 - *Lemma 4.7: Let the deletion distribution μ be the uniform distribution. Is it always valid to ...*
>
> We agree that assuming the deletion distribution to be uniform is not always valid. Our main theorem (Theorem 4.6) applies to general data-dependent deletion distributions. Lemma 4.7 is an instantiation of Theorem 4.6 with a uniform deletion distribution in order to provide an illustrative example of deletion time savings under system-aware unlearning. For the main Theorem 4.6, the main assumption we make is that deletion requests are not adaptive/adversarial to updates in the system.
>
> > EDA 1 - *The experimental setup is relatively simple and limited. The current study focuses on ...*
>
> We emphasize that our contributions are primarily theoretical. Our main contribution is to provide a new definition of unlearning that is not as pessimistic as the traditional one and show how this definition can help us obtain much more efficient unlearning algorithms in a principled way. Several works that have used empirical metrics for unlearning have been proven to leak information about deleted data points [3, 4], thus illustrating the dire need for a theoretically sound definition for unlearning. Even under traditional unlearning definitions, it is still unclear how to perform theoretically rigorous and efficient unlearning in simple models like regression.
>
> > EDA 2 - *Regarding the experiments, would it be possible to provide further evaluations on how ...*
>
> In Theorem 4.6, the bound on the sample size of $S’$ from the selection process holds under any data distribution. The sample size of $S’$ can be tuned by properly setting $\kappa$ in the BBQSampler. When $\kappa$ is large, the sample size of $S’$ is large, leading to higher accuracy. When $\kappa$ is small, the sample size of $S’$ is small, leading to fast expected deletion times and low memory requirements. The exact tradeoffs are characterized by Theorem 4.4 and Theorem 4.6.
>
> > ERND 1 - *There are also related work in [1] and [2] that emphasize the importance of ...*
>
> These two papers consider unlearning under unique learning architectures, specifically under machine-learning-as-a-service [1] and federated learning [2] settings. Both of these papers work to satisfy the traditional definition of unlearning under these unique learning systems. We agree that it would be interesting to explore how these unique settings could benefit from a system-aware perspective on unlearning.
>
> Please let us know if you have any additional questions or concerns.
>
> [1] Eraser: Machine unlearning in mlaas via an inference serving-aware approach. Yuke Hu, Jian Lou, Jiaqi Liu, Feng Lin, Zhan Qin, and Kui Ren. ACM SIGSAC 2024.
>
> [2] Federaser: Enabling efficient client-level data removal from federated learning models. Gaoyang Liu, Xiaoqiang Ma, Yang Yang, Chen Wang, and Jiangchuan Liu.  IEEE IWQOS 2021
>
> [3] Machine Unlearning Fails to Remove Data Poisoning Attacks. Martin Pawelczyk, Jimmy Z. Di, Yiwei Lu, Gautam Kamath, Ayush Sekhari, Seth Neel. ICLR 2025.
>
> [4] Inexact Unlearning Needs More Careful Evaluations to Avoid a False Sense of Privacy. Jamie Hayes, Ilia Shumailov, Eleni Triantafillou, Amr Khalifa, Nicolas Papernot. SatML 2025.

---

### Official Review · Reviewer_MJw8 · 2025-04-03

**Overall Recommendation:** 3

**Summary:**

This paper introduces system-aware unlearning, a novel framework that generalizes traditional machine unlearning by relaxing privacy guarantees to account for realistic attacker access to the system’s internal data. It proposes a general approach using sample compression or core sets, where reduced algorithmic reliance on stored data inherently limits attack exposure. The authors demonstrate this via the first exact unlearning algorithm for linear classification with sublinear memory in the dataset size, enabled by selective sampling for compression. Theoretical analysis establishes bounds on computation, memory, deletion capacity, and excess risk, balancing efficiency and privacy in practical settings. This framework bridges rigorous unlearning guarantees with feasible resource requirements.

**Claims And Evidence:**

See Strengths and Weaknesses.

**Essential References Not Discussed:**

The paper focuses on the field of certified machine unlearning, but lacks of following papers:

[1] Eli Chien, Chao Pan, and Olgica Milenkovic. “Certified Graph Unlearning”.

[2] Jiaqi Liu, Jian Lou, Zhan Qin, and Kui Ren. “Certified Minimax Unlearning with Generalization Rates and Deletion Capacity”. In NeurIPS 2023.

[3] Binchi Zhang, Yushun Dong, Tianhao Wang, Jundong Li. "Towards Certified Unlearning for Deep Neural Networks." in ICML 2024.

**Experimental Designs Or Analyses:**

There're no experimental designs.

**Methods And Evaluation Criteria:**

See Strengths and Weaknesses.

**Other Comments Or Suggestions:**

1. It would be better to palce Eq.(1) and Eq.(2) after Definition 2.2, which makes the logic more coherent before Definition 2.3.
2. Grammar error. For example, in Definition 2.3, '... there exists a...' should be '... there exists an...'

**Other Strengths And Weaknesses:**

Strengths

1. Proposes a practical relaxation of traditional unlearning definitions by introducing system-aware unlearning, which accounts for realistic attacker capabilities.
2. Generalizes traditional unlearning while enabling more efficient algorithms.
3. Provides the first exact unlearning algorithm for linear classification with sublinear memory, breaking prior lower bounds (Cherapanamjeri et al., 2024) under traditional definitions.
4. Rigorous analysis of deletion capacity, excess risk, and computational complexity, with tradeoffs controlled by the sampling parameter $\kappa$.




Weaknesses

1. The use of mathematical symbols is not standardized. The font size of symbol $\backslash$, $\in$,  $\subseteq$ is different from other letter symbols, which causes difficulties for readers to read.

2. Lack of experimental verification. The paper lacks quantification of the unlearning performance, such as F1 score in [3] and the accuracy of the membership inference attack.

**Questions For Authors:**

1. Why does the input of state-of-system in Definition 3.1 has only one parameters (i.e., $S$) while the input of state-of-system in Eq.(1) & Eq.(2) has 2 input parameters (i.e., $S$ and $U$)? They all use the symbol $\mathrm{I}$, but the definitions before and after are not coherent.

**Relation To Broader Scientific Literature:**

This paper presents work whose goal is to advance the field of machine unlearning, which is specifically oriented to improve the trustworthiness of machine learning.

**Theoretical Claims:**

The reviewer did not conduct a thorough review of the proofs.

---

### Decision · Program_Chairs · 2025-05-01

**Decision:**

Accept (poster)

**Comment:**

This paper proposes a new machine unlearning framework that accounts for realistic attacker access to the system’s internal data. The paper develops an exact learning/unlearning algorithm for linear classification based on coreset construction. It provides theoretical analysis in terms of bounds on deletion capacity, memory requirements, and computation time.

The reviewers appreciate the following strengths: 1) Proposes a new and interesting generalization of the machine unlearning definition that accounts for realistic attacker access to the system’s internal data;
2) Establishes a series of theoretical results in terms of bounds on deletion capacity, memory requirements, and computation time;
3) Mostly well-written, with intuitive explanations that make it easier to follow.

The reviewers also raise several concerns:
1) Mostly limited to simple models such as linear/convex models;
2) Lacks empirical validation;
3) Misses some related work;
4) Provides insufficient discussion about the practicality and assumptions of the new unlearning definition.

During the discussion, the authors provide detailed feedback and addressed most of the concerns. Three of the reviewers give an overall positive evaluation, while two reviewers still lean towards negative. The remaining concern mainly lies in the lack of empirical results. The area chair aligns with the majority of the reviewers and believes that this paper makes novel and interesting theoretical contributions. Regarded as a pure theoretical paper, and in light of its theoretical merits, the area chair recommends weak acceptance, with the final decision left to the SAC and PCs.